# TRIBE: TRI-COMPONENT INFORMATION DECOMPOSITION FOR GRAPH OUT-OF-DISTRIBUTION DETECTION

## ABSTRACT

Graph neural networks are widely used for node classification, but they remain vulnerable to out-of-distribution (OOD) shifts in node features and graph structure. Prior work established that methods trained with standard supervised learning (SL) objectives tend to capture spurious signals from either features and/or structure, leaving the model fragile under distributional changes. To address this, we propose TRIBE, a **novel and effective Tri-Component Information Decomposition** framework that **explicitly decomposes information** into *feature-specific, structure-specific and joint* components. TRIBE aims to preserve only the label-relevant component of the joint information while filtering out spurious feature- and structure-specific information, thereby enhancing the separation between in-distribution (ID) and OOD data. Technically, we develop a novel optimisation pipeline that integrates a graph Information Bottleneck (IB) objective with carefully designed regularisations. Beyond the framework, we provide theoretical and empirical analysis showing the superiority of IB in OOD detection, with higher ID confidence and a greater entropy gap between ID and OOD data compared to the typical SL objective. Extensive experiments across seven datasets confirm the efficacy of TRIBE, achieving up to $34\%$ improvement in FPR95 over strong baselines while maintaining competitive ID accuracy. Code will be released upon acceptance.

## 1 INTRODUCTION

Graph neural networks (GNNs) have become a dominant approach for node classification across real-world domains; however, they remain vulnerable to out-of-distribution (OOD) shifts in node features and/or graph structure, limiting reliable deployment in real-world applications (Li et al., 2022a; Wu et al., 2023b; Gui et al., 2022; Yang et al., 2022; Shen et al., 2024; Giuffrè & Shung, 2023). For instance, in biomedical graphs, node features like biomarkers may correlate with health outcomes in one hospital but shift in another, degrading performance despite stable patient-symptom links (**feature shift**). In e-commerce graphs, co-purchase links may change as shopping patterns evolve, even when product attributes remain stable (**structural shift**). In social networks, new communities can introduce both novel posting behaviours (features) and friendship links (structure), leading to simultaneous changes in features and structure signals (**joint shift**) (Wang et al., 2025b).

Given these challenges, OOD detection has become a key priority for identifying nodes beyond the in-distribution (ID) training data (Lang et al., 2023; Zhou et al., 2022; Bazhenov et al., 2022). Existing graph OOD detectors generally address shifts by enforcing invariant node representations via tailored objectives or augmentations, applying topology-aware metrics to capture node irregularities (Li et al., 2022b; Song & Wang, 2022; Bao et al., 2024), or using post-hoc scoring functions to separate ID from OOD nodes (Wang et al., 2021; Liu et al., 2020; Wu et al., 2023b; Hendrycks & Gimpel, 2017; Hendrycks et al., 2022; Lee et al., 2018b). Others incorporate contrastive or exposure-style objectives when auxiliary OOD data is available (Hendrycks et al., 2019; Yang et al., 2024b).

While effective, existing approaches mainly rely on a **mixed representation of a graph's features and structure**, without adequately addressing individual contributions from features $\mathbf{X}$ or structure $\mathbf{A}$ (Wu et al., 2023b). This mixed representation corresponds to the node embeddings $\mathbf{Z} = f(\mathbf{X}, \mathbf{A})$ learned by a GNN $f$ via a standard supervised learning (SL) objective. As shown in Figure 1, the representation (red dotted circle) mixes label-irrelevant information (shaded area) with three label-

related signals: the feature-specific signal that depends only on $\mathbf{X}$, the structure-specific signal from $\mathbf{A}$, and the joint-input signal from the synergy of features and structure. While all three signals appear predictive in ID classification, prior work has shown that feature-only and structure-only signals often reflect spurious correlations with the label, which become problematic under distribution shifts (Chen et al., 2023; 2024; Fan et al., 2024). For instance, under data having feature shifts but ID-like structures, $(\mathbf{X}_{\text{OOD}}, \mathbf{A}_{\text{ID}})$, a model exploiting spurious structure-label correlations may confidently misclassify OOD data as ID; similarly, under structure shifts but ID-like features, $(\mathbf{X}_{\text{ID}}, \mathbf{A}_{\text{OOD}})$, reliance on feature-only correlations again leads to overconfident class prediction that fails to be detected. In contrast, the joint-input signal reveals distribution shifts more easily via mismatches between features and structure during encoding. Thus, OOD detection can fail under supervised learning when the model abuses spurious correlations from individual inputs (e.g., $\mathbf{X}$ or $\mathbf{A}$) instead of using the more shift-indicative joint-input information.

To ideally handle real-world distribution shifts, **an effective graph OOD detector should prioritise the joint-input label-relevant information as the main predictive signal, while filtering out the feature- and structure-specific spurious correlations**. Figure 1 illustrates this motivation: while SL mixes feature-only, structure-only, and joint signals as well as the label-irrelevant information in the shaded area (red dotted circle), compressing the learned representation toward the triangle-like overlap (red solid triangle) to preserve only the joint-input signals can effectively identify the OOD data in graphs. To achieve this, we propose TRIBE, a Tri-

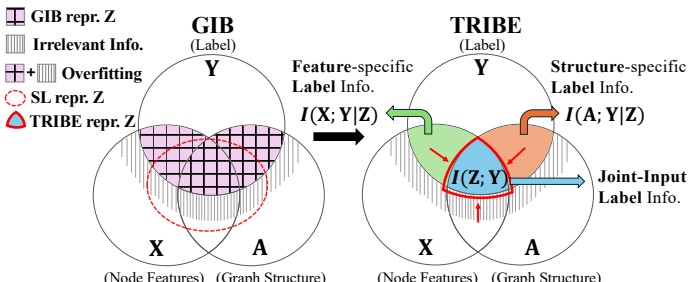

Figure 1: Comparison of the information captured by SL, GIB (Wu et al., 2020), and TRIBE over node features $\mathbf{X}$, graph structure $\mathbf{A}$, labels $\mathbf{Y}$, and the latent representation $\mathbf{Z}$. The dashed red circle illustrates how standard supervised learning mixes feature-, structure-, joint-, and label-irrelevant signals, which are prone to spurious correlations that causes OOD nodes appear ID. The gridded region shows that GIB suppresses label-irrelevant noise but still keeps input-specific spurious cues. In contrast, the solid triangle highlights the joint, label-relevant region that TRIBE captures by separating feature-/structure-specific, and joint information, enabling stronger OOD detection.

Component Information Decomposition framework. Unlike prior OOD detection methods that mix the feature and structure information as a single unified signal, TRIBE guides learning toward the ideal joint-input triangular region by explicitly decomposing label information $I(\mathbf{X}, \mathbf{A}; \mathbf{Y})$ into: feature-specific information $I(\mathbf{X}; \mathbf{Y}|\mathbf{Z})$, structure-specific information $I(\mathbf{A}; \mathbf{Y}|\mathbf{Z})$, and joint-input information $I(\mathbf{Z}; \mathbf{Y})$. We design tailored networks and regularisations to preserve the joint-input signals while suppressing the individual-input components (indicated by the left and right arrows →). Additionally, an information bottleneck (IB) objective is employed to filter out the label-irrelevant noise (indicated by the upward arrow ↑), producing a compact and shift-indicative representation. We further provide theoretical insights showing that, compared to SL, IB promotes higher ID confidence and more reliable separation between ID and OOD samples for logit-based detection. Thus, our **contributions** are threefold:

1. **Methodological:** We propose TRIBE, a novel *tri-component information decomposition framework* for graph OOD detection, that explicitly decomposes predictive information into *feature-specific*, *structure-specific*, and *joint* components, and introduces conditional-independence and pairwise mutual-information regularisations, along with an IB objective to filter out label-irrelevant and spurious correlations.

2. **Theoretical:** We prove that IB increases ID confidence and enlarges the entropy gap between ID and OOD compared to standard SL, thereby improving logit-based OOD detection.

3. **Empirical:** On seven real-world and synthetic graph datasets, TRIBE achieves up to **34% improvement in FPR95** over strong baselines while maintaining competitive ID accuracy.

## 2 RELATED WORK

Graph OOD detection builds on several lines of research. **1) Scoring-based methods** rely on ID data to design OOD scores (Lee et al., 2018a; Koo et al., 2024; Ding & Shi, 2023; Ma et al., 2023; Liu

et al., 2023; Hendrycks & Gimpel, 2017). **2) OOD exposure approaches** incorporate auxiliary OOD data during training for improved detection (Hendrycks et al., 2019; Liu et al., 2020; Park et al., 2023; Zhu et al., 2023; Yang et al., 2024b; Bao et al., 2024). **3) IB principle** has been applied to image and graph representation learning (Wu et al., 2020; Alemi et al., 2017; Tishby et al., 2000), but its role in OOD detection has mainly been studied in Euclidean domains (Hu et al., 2024; Zhao & Cao, 2023; Li et al., 2023b; Wu & Deng, 2024). Prior work analysed "overconfidence" only under the standard supervised objective ($\max I(\mathbf{Z}; \mathbf{Y})$) (Hu et al., 2024), whereas our **theoretical contribution** (Sec. 5) shows how the full IB objective improves both ID confidence and logit-based OOD detection – a gap unaddressed in earlier studies (Hu et al., 2024; Alemi et al., 2018). **4) Graph-specific methods** include GNNSAFE/++ with propagation-based detection (Wu et al., 2023b), NODESAFE/++ with constrained energy scores (Yang et al., 2024b), DeGEM with multi-hop energy-based modelling for heterophilic graphs (Chen et al., 2025), and GOLD with pseudo-OOD embedding synthesis (Wang et al., 2025a). Due to space constraint, a detailed review is provided in Appendix G.

## 3 PRELIMINARY

**Node Classification and Graph Representation.** For node classification, a graph is typically represented as $\mathcal{G} = (\mathbf{X}, \mathbf{A})$. Here, $\mathbf{X} \in \mathbb{R}^{n \times d}$ denotes the node feature matrix, where $n$ is the number of nodes and $d$ is the feature dimension. The adjacency matrix $\mathbf{A} \in \mathbb{R}^{n \times n}$ captures the connections between nodes. Each node $i$ is associated with a label $y_i \in \{1, 2, \ldots, C\}$, where $C$ represents the total number of classes. In this paper, we focus on two tasks:

**Task 1: In-distribution Classification.** Given test nodes from the same distribution as training ($P_{train}(\mathbf{X}, \mathbf{A}) = P_{test}(\mathbf{X}, \mathbf{A})$ and $P_{train}(\mathbf{y}|\mathbf{X}, \mathbf{A}) = P_{test}(\mathbf{y}|\mathbf{X}, \mathbf{A})$), the goal is to train an $L$-layer GNN to predict node labels $\mathbf{y} \in \mathbb{R}^n$ (refer to Appendix C for further details on GNN):

$$\mathbf{y} = \text{Softmax}(\text{GNN}(\mathbf{X}, \mathbf{A})).$$

**Task 2: Out-of-distribution Detection.** Here, the objective is to identify test nodes from a different distribution ($P_{train}(\mathbf{X}, \mathbf{A}) \neq P_{test}(\mathbf{X}, \mathbf{A})$ or $P_{train}(\mathbf{y}|\mathbf{X}, \mathbf{A}) \neq P_{test}(\mathbf{y}|\mathbf{X}, \mathbf{A})$). This is formulated by an OOD detector $F$ with scoring function $S$ and threshold $\tau$:

$$F(\mathbf{x}, \mathbf{A}; \text{GNN}) = \begin{cases} \text{OOD}, & S(\mathbf{x}, \mathbf{A}; \text{GNN}) \geq \tau, \\ \text{ID}, & S(\mathbf{x}, \mathbf{A}; \text{GNN}) < \tau. \end{cases} \tag{1}$$

Extended definitions for structure, feature, joint, and label shifts are in Appendix H (Gui et al., 2022).

**Energy-Based OOD Detection.** The energy score was proposed as an effective scoring function for distinguishing OOD from ID samples (Liu et al., 2020). For a node $i$, the energy score is:

$$S(\mathbf{x}_i, \mathbf{A}; \text{GNN}) = e_i = -\log \sum_{c=0}^{C-1} \exp(\ell_{i,c}), \tag{2}$$

where $e_i \in \mathbb{R}$ indicates the energy score, $\ell_i \in \mathbb{R}^C$ are the logits from GNN$(\mathbf{X}, \mathbf{A})$. GNNSAFE (Wu et al., 2023b) adapts this to graphs via energy propagation:

$$\mathbf{e}^{(k)} = \alpha \mathbf{e}^{(k-1)} + (1 - \alpha)\mathbf{D}^{-1}\mathbf{A}\mathbf{e}^{(k-1)}, \tag{3}$$

where $\alpha$ controls propagation and $\mathbf{D}$ is the degree matrix. For OOD exposure, (Liu et al., 2020) further introduces energy regularisation to separate ID and OOD scores with thresholds $t_{\text{ID}}$ and $t_{\text{OOD}}$:

$$\max \mathcal{L}_{\text{EReg}}, \text{ where } \mathcal{L}_{\text{EReg}} = \mathbb{E}_{i \sim P_{\text{ID}}}[\max(0, t_{\text{ID}} - e_i)]^2 + \mathbb{E}_{j \sim P_{\text{OOD}}}[\max(0, e_j - t_{\text{OOD}})]^2. \tag{4}$$

## 4 METHOD

**Motivation.** Graph data is inherently multi-modal with node features $\mathbf{X}$ and graph structure $\mathbf{A}$. A key challenge, however, is that the typical SL objective does not distinguish whether predictive signals arise jointly from $(\mathbf{X}, \mathbf{A})$ or from individual inputs. As illustrated in Figure 1, this mixture of information (dotted circle) means standard SL-trained models may rely on spurious correlations from an individual input, which can be predictive under ID but mislead detection under shifts (Li et al., 2023a). Thus, our aim is to learn the desired label-relevant joint signal while filtering out the label-irrelevant and spurious feature-/structure-only cues, as in Figure 1's solid triangle region.

Figure 2: Framework of TRIBE: an information decomposition approach that preserves joint label-relevant information in $\mathbf{Z}$ while suppressing spurious feature-only ($\mathbf{V}$) and structure-only ($\mathbf{Q}$) signals.

In light of this, we propose TRIBE, a novel graph-tailored tri-component decomposition that separates label-relevant information into joint, feature-specific, and structure-specific components (Figure 2). The central idea is that the joint-input representation $\mathbf{Z}$ should capture the stable interactions between $\mathbf{X}$ and $\mathbf{A}$, while auxiliary networks isolate feature-only $\mathbf{V}$ and structure-only $\mathbf{Q}$ signals. Training is guided by novel regularisers and an IB objective to disentangle the three components and suppress irrelevant signals, ensuring $\mathbf{Z}$ remains focused on the desired label-relevant joint information. In the following sections, we present our **novel tri-component information decomposition paradigm** (Section 4.1) and propose the TRIBE framework (Section 4.2).

## 4.1 INFORMATION BASED DECOMPOSITION PARADIGM

To begin, we formalise the relationship between inputs, encoded representation, and labels as follows.

**Proposition 4.1. (Information-Preserving Representation Property)**
*Given a representation $\mathbf{Z}$ encoded from inputs $(\mathbf{X}, \mathbf{A})$ by a network $f$ parametrised by $\theta$, and is maximised for predicting label $\mathbf{Y}$, we have $I(\mathbf{Z}; \mathbf{Y}|\mathbf{X}, \mathbf{A}) = 0$ and $I(\mathbf{X}, \mathbf{A}; \mathbf{Y}) = I(\mathbf{X}, \mathbf{A}, \mathbf{Z}; \mathbf{Y})$.*

Here, $I(\mathbf{X}, \mathbf{A}; \mathbf{Y})$ represents the maximum information that we have about the prediction of $\mathbf{Y}$ given inputs $(\mathbf{X}, \mathbf{A})$. Proposition 4.1 thus states that $\mathbf{Z}$ cannot contain more information about $\mathbf{Y}$ than is already present in $(\mathbf{X}, \mathbf{A})$. The proof is provided in Appendix B.1. This leads to the decomposition:

$$I(\mathbf{X}, \mathbf{A}; \mathbf{Y}) = I(\mathbf{X}, \mathbf{A}, \mathbf{Z}; \mathbf{Y}) = I(\mathbf{Z}; \mathbf{Y}) + \underbrace{I(\mathbf{X}, \mathbf{A}; \mathbf{Y}|\mathbf{Z})}_{\text{residual}}, \overset{\text{Chain rule for mutual information}}{} \tag{5}$$

where the last term represents residual label information not captured by $\mathbf{Z}$. To isolate input-specific signals, this residual in Eq. 5 can be further decomposed into features and structure contributions:

$$I(\mathbf{X}, \mathbf{A}; \mathbf{Y}) = I(\mathbf{Z}; \mathbf{Y}) + \underbrace{I(\mathbf{X}; \mathbf{Y}|\mathbf{Z}) + I(\mathbf{A}; \mathbf{Y}|\mathbf{Z}, \mathbf{X})}_{\text{residual}}. \tag{6}$$

To effectively decouple $\mathbf{X}$ and $\mathbf{A}$, we enforce a conditional independence constraint $\mathbf{A} \perp\!\!\!\perp \mathbf{X} \mid \mathbf{Z}$, which is not a restrictive dataset assumption but a modelling objective. More discussions can be found in the framework realisation in Sec 4.2 below. This constraint guides $\mathbf{Z}$ to retain all label-relevant joint information while reducing residual dependence between $\mathbf{X}$ and $\mathbf{A}$:

$$p(\mathbf{X}, \mathbf{A}|\mathbf{Z}) = p(\mathbf{X}|\mathbf{Z})p(\mathbf{A}|\mathbf{Z}). \tag{7}$$

Thus, following Eq. 6, the mutual information between $(\mathbf{X}, \mathbf{A})$ and $\mathbf{Y}$ is:

$$I(\mathbf{X}, \mathbf{A}; \mathbf{Y}) = \underbrace{I(\mathbf{Z}; \mathbf{Y})}_{\text{joint}} + \underbrace{I(\mathbf{X}; \mathbf{Y}|\mathbf{Z})}_{\text{feature}} + \underbrace{I(\mathbf{A}; \mathbf{Y}|\mathbf{Z})}_{\text{structural}}. \tag{8}$$

This illustrates our tri-component information decomposition paradigm in Figure 1. By separating label information into feature, structure, and joint components, we obtain a compact, robust joint representation $\mathbf{Z}$ for predicting $\mathbf{Y}$, free of spurious correlations from individual inputs.

## 4.2 TRIBE FRAMEWORK

The decomposition described in Eq. 8 motivates the design of TRIBE for OOD detection. Rather than relying on a single representation, we explicitly introduce three respective modules to realise the decomposition, as shown in Figure 2: (1) The joint encoder $\mathbf{Z} = f(\mathbf{X}, \mathbf{A})$ serves as **the primary encoder network for both classification and OOD detection**, capturing joint-input label-relevant

information between $\mathbf{X}$, $\mathbf{A}$ and $\mathbf{Y}$. By focusing on the joint-input correlations, the model becomes more sensitive to shifts, making OOD cases (e.g., $(\mathbf{X}_{\text{OOD}}, \mathbf{A}_{\text{ID}})$) more distinguishable from ID. (2) The feature-specific network $\mathbf{V} = g_X(\mathbf{X})$ is used to isolate residual information that remains specific to $\mathbf{X}$, and (3) the structure-specific network $\mathbf{Q} = g_A(\mathbf{A})$ isolates residual information specific to $\mathbf{A}$. These **two auxiliary networks mitigate the spurious individual-input correlations for predicting the label within the input graph data** that drives detection-errors, enabling more effective detection of **structural- and/or feature-driven shifts**. Given the three networks, the information-based decomposition paradigm in Eq. 8 can be realised as:

$$I(\mathbf{X}, \mathbf{A}; \mathbf{Y}) \;=\; I(\mathbf{Z}; \mathbf{Y}) \;+\; I(\mathbf{V}; \mathbf{Y}) \;+\; I(\mathbf{Q}; \mathbf{Y}). \qquad (9)$$

With this construction, $\mathbf{Z}$ carries the joint signal that is beneficial for classification and OOD detection, while $\mathbf{V}$ and $\mathbf{Q}$ act as auxiliary pathways encoding the feature- and structure-specific components, which may contain spurious correlations for making predictions. This design ensures that ID-OOD separation through $\mathbf{Z}$ is more effective under shifts.

**Principled Regularisers.** To ensure a meaningful decomposition that separates the joint-input label-relevant information $\mathbf{Z}$ from spurious correlations between individual inputs and the label, captured respectively by the feature- ($\mathbf{V}$) and structure-specific ($\mathbf{Q}$) representations, TRIBE introduces two principled regularisers. These consist of a **conditional independence** constraint (Eq. 7) and a **pairwise mutual information minimisation** constraint, which together ensure the theoretical decomposition in Eq. 8 under the three-networks formulation in Eq. 9.

**1) Conditional independence regulariser**. Its role is to encourage $\mathbf{Z}$ to capture the meaningful interactions between $\mathbf{X}$ and $\mathbf{A}$, so that the remaining dependence between feature and structure information is treated as spurious. This realises the independence constraint in Eq. 7, and the regularisation is defined as:

$$\min \mathcal{L}_{\text{CInd}_{\mathbf{X}, \mathbf{A}, \mathbf{Z}}}, \text{ where } \mathcal{L}_{\text{CInd}_{\mathbf{X}, \mathbf{A}, \mathbf{Z}}} = I(\mathbf{A}; \mathbf{X} | \mathbf{Z}). \qquad (10)$$

Note that this is not a dataset assumption that features and structure are independent. Rather, it is an optimisation regulariser that acts as an inductive bias: **once $\mathbf{Z}$ has captured the relevant joint-input signal (key for detecting distribution shifts), the residual information between X and A should become independent** (i.e., $I(\mathbf{X}; \mathbf{A} | \mathbf{Z}) \to 0$). In practice, this conditional independence corresponds to approximating $p(\mathbf{X}, \mathbf{A} | \mathbf{Z}) \approx p(\mathbf{X} | \mathbf{Z}) p(\mathbf{A} | \mathbf{Z})$. This design reflects realistic OOD situations: for example in social graphs, connectivity patterns may evolve when user attributes such as demographics remain stable; for financial fraud detection graphs, transaction features (e.g., transaction amount) may change while the underlying network structure is preserved. Thus, once the joint label-relevant semantics are captured, separating the remaining spurious feature- and structure-only information becomes crucial, as they can mislead OOD detection.

**2) Pairwise mutual information minimisation**. Intuitively, since the three networks in Eq. 9 are designed to capture different input information, we need to prevent them from redundantly encoding the same or overlapping information. Thus, we penalise pairwise overlaps among the three learned representations by minimising their mutual information:

$$\min \mathcal{L}_{\text{PMI}_{\mathbf{Z}, \mathbf{V}, \mathbf{Q}}}, \text{ where } \mathcal{L}_{\text{PMI}_{\mathbf{Z}, \mathbf{V}, \mathbf{Q}}} = \alpha_1 I(\mathbf{Z}; \mathbf{V}) + \alpha_2 I(\mathbf{Z}; \mathbf{Q}) + \alpha_3 I(\mathbf{V}; \mathbf{Q}), \qquad (11)$$

where $\alpha_1, \alpha_2, \alpha_3$ are scalar weights. Minimising this loss keeps $\mathbf{Z}$ focused on the desired joint signal, while $\mathbf{V}$ and $\mathbf{Q}$ remain disentangled and feature-/structure-specific respectively.

**Final Objective.** To unify the encoding objective with the proposed regularisation, the training is formulated in an information-bottleneck (IB) style. Each of the joint, feature-specific, and structure-specific encoders is optimised such that **(i) predictive information for the labels is retained**, and **(ii) label-irrelevant noise that obscures the ID-OOD boundary is suppressed** (details on IB are in Appendix D). This aligns with our decomposition (Eq. 8, Eq. 9): $\mathbf{Z}$ captures the joint-input label-relevant information, while $\mathbf{V}$ and $\mathbf{Q}$ represent the remaining feature- and structure-specific components. By reducing spurious individual-input correlations, the IB objective strengthens OOD detection over standard SL objective, with theoretical analysis detailed in Section 5, Proposition 5.3.

$$\text{IB}_{\mathbf{Z}} = \max I(\mathbf{Z}; \mathbf{Y}) - \beta_{\mathbf{Z}} I(\mathbf{X}, \mathbf{A}; \mathbf{Z})$$
$$\text{IB}_{\mathbf{V}} = \max I(\mathbf{V}; \mathbf{Y}) - \beta_{\mathbf{V}} I(\mathbf{X}; \mathbf{V}), \quad \text{IB}_{\mathbf{Q}} = \max I(\mathbf{Q}; \mathbf{Y}) - \beta_{\mathbf{Q}} I(\mathbf{A}; \mathbf{Q}) \qquad (12)$$
$$\max \mathcal{L}_{\text{IB}_{\mathbf{Z}; \mathbf{V}; \mathbf{Q}}}, \text{ where } \mathcal{L}_{\text{IB}_{\mathbf{Z}; \mathbf{V}; \mathbf{Q}}} = \text{IB}_{\mathbf{Z}} + \text{IB}_{\mathbf{V}} + \text{IB}_{\mathbf{Q}},$$

The final TRIBE objective combines these terms with the regularisers to realise our information decomposition objective in Eq. 9:

$$\max_{\theta_{\mathbf{Z}},\theta_{\mathbf{V}},\theta_{\mathbf{Q}}} \mathcal{L}_{\text{TRIBE}_{\mathbf{Z};\mathbf{V};\mathbf{Q}}} = \max_{\theta_{\mathbf{Z}},\theta_{\mathbf{V}},\theta_{\mathbf{Q}}} \mathcal{L}_{\text{IB}_{\mathbf{Z};\mathbf{V};\mathbf{Q}}} - \lambda_{\text{CInd}}\mathcal{L}_{\text{CInd}_{\mathbf{X},\mathbf{A},\mathbf{Z}}} - \mathcal{L}_{\text{PMI}_{\mathbf{Z},\mathbf{V},\mathbf{Q}}}, \tag{13}$$

where $\lambda_{\text{CInd}}$ is a loss coefficient and the scale weight for $\mathcal{L}_{\text{PMI}}$ is handled by $\alpha$ in Eq. 11. This formulation ensures $\mathbf{Z}$ provides a stable classification and OOD detection backbone, while $\mathbf{V}$ and $\mathbf{Q}$ capture component-specific signals that guide disentanglement. By design, ID node classification and OOD detection depend on $\mathbf{Z}$, while $\mathbf{V}$ and $\mathbf{Q}$ are only used to regularise training.

### 4.3 IMPLEMENTATION AND INFERENCE

We note that **direct computation of mutual information in our final objective (Eq. 13) is intractable**, so we follow standard practice and adopt variational approximations. We use VIB (Alemi et al., 2017) to estimate the IB terms, and use reconstruction loss and CLUB (Cheng et al., 2020) loss to estimate conditional-independence and PMI regularisers, respectively. Due to the page limit, and *to emphasise our primary contribution*, the detailed discussion on obtaining the **tractable objective** is instead provided in Appendix E, a **neural network parameterisation** of TRIBE is given in Appendix F, and Algorithm 1 illustrates the optimisation process with detailed description provided in Appendix E. Without loss of generality, in the following sections, we refer to the mutual information and TRIBE objective as their tractable variational forms (i.e., $\mathcal{L}_{\text{VIB}}, \mathcal{L}_{\text{VCInd}}, \mathcal{L}_{\text{VPMI}}$). **At inference time**, only the GNN classifier for $\mathbf{Z}$ will be used for OOD detection with the energy score in Eq. 3 derived from prediction logits (Algorithm. 2).

---

**Algorithm 1** TRIBE Framework

**Input:** ID graph $\mathcal{G} = (\mathbf{X}, \mathbf{A})$, randomly initialised GNN classifier ($\text{GNN}_{\text{CLS-}\mathbf{Z}}$), MLP feature network ($\text{MLP}_{\text{Feat-}\mathbf{V}}$), structure network ($\text{GNN}_{\text{Struct-}\mathbf{Q}}$) with parameters $\theta_{\mathbf{Z}}, \theta_{\mathbf{V}}, \theta_{\mathbf{Q}}$ respectively, loss coefficients $\alpha_1, \alpha_2, \alpha_3, \lambda_{\text{CInd}}$.
**Output:** Optimised $\text{GNN}_{\text{CLS-}\mathbf{Z}}$.
**while** $train$ **do**
  Update:
  **1. Minimising** $-\mathcal{L}_{\text{VIB}_{\mathbf{Z};\mathbf{V};\mathbf{Q}}}$ **Eq. 38**
  $\text{GNN}_{\text{CLS-}\mathbf{Z}} \leftarrow -\text{VIB}_{\mathbf{Z}}$
  $\text{MLP}_{\text{Feat-}\mathbf{V}} \leftarrow -\text{VIB}_{\mathbf{V}}$
  $\text{GNN}_{\text{Struct-}\mathbf{Q}} \leftarrow -\text{VIB}_{\mathbf{Q}}$
  **2. Minimising** $\mathcal{L}_{\text{VCIND}_{\mathbf{X},\mathbf{A},\mathbf{Z}}}$ **Eq. 39**
  $\text{GNN}_{\text{CLS-}\mathbf{Z}} \leftarrow \lambda_{\text{CInd}} \mathcal{L}_{\text{VCIND}_{\mathbf{X},\mathbf{A},\mathbf{Z}}}$
  **3. Minimising** $\mathcal{L}_{\text{VPMI}_{\mathbf{Z};\mathbf{V};\mathbf{Q}}}$ **Eq. 41**
  $\text{GNN}_{\text{CLS-}\mathbf{Z}} \leftarrow \alpha_1 I_{\text{VCLUB}_{\mathbf{Z};\mathbf{V}}} + \alpha_2 I_{\text{VCLUB}_{\mathbf{Z};\mathbf{Q}}}$
  $\text{MLP}_{\text{Feat-}\mathbf{V}} \leftarrow \alpha_1 I_{\text{VCLUB}_{\mathbf{Z};\mathbf{V}}} + \alpha_3 I_{\text{VCLUB}_{\mathbf{V};\mathbf{Q}}}$
  $\text{GNN}_{\text{Struct-}\mathbf{Q}} \leftarrow \alpha_2 I_{\text{VCLUB}_{\mathbf{Z};\mathbf{Q}}} + \alpha_3 I_{\text{VCLUB}_{\mathbf{V};\mathbf{Q}}}$
**end while**

---

**Algorithm 2** TRIBE Inference

1: **Input:** Test graph $\mathcal{G} = (\mathbf{X}, \mathbf{A})$, optimised GNN classifier ($\text{GNN}_{\text{CLS-}\mathbf{Z}}$) with parameter $\theta_{\mathbf{Z}}$.
2: **Output:** Predicted labels and energy scores.
3: **1. Inference: Obtain logits**
4: $\ell \leftarrow \text{GNN}_{\text{CLS-}\mathbf{Z}}(\mathcal{G})$
5: **2. Score calculation**
6: Energy: $\mathbf{e} \leftarrow$ Eq. 2, 3 using $\ell$.
7: Prediction: $\hat{\mathbf{Y}} \leftarrow \text{Softmax}(\ell)$.

---

## 5 THEORETICAL INSIGHTS ON IB FOR GRAPH OOD DETECTION

While the previous section introduced decomposition as the main methodological contribution, we now show why IB is better suited than standard SL as the optimisation backbone. Unlike SL (i.e., $\max I(\mathbf{Z}; \mathbf{Y})$), which rewards any predictive correlation, IB retains only information truly supporting the label. We show that this yields **two key benefits**: (i) sharper ID prediction confidence and (ii) a larger entropy gap between ID and OOD data, which **directly improves logit-based OOD detection**.

**Lemma 5.1. (Target-Irrelevant Information and ID Prediction Confidence)**
*Minimising the conditional mutual information $I(\mathbf{X}, \mathbf{A}; \mathbf{Z} \mid \mathbf{Y})$ reduces the conditional entropy $H(\mathbf{Z} \mid \mathbf{Y})$, leading to a more concentrated posterior distribution $P(\mathbf{Y} \mid \mathbf{Z})$.*

Lemma 5.1 shows that when the representation $\mathbf{Z}$ discards target-irrelevant information, the uncertainty of $\mathbf{Z}$ conditioned on $\mathbf{Y}$ decreases. In other words, the representation becomes more predictable given the label, which sharpens the posterior distribution $P(\mathbf{Y} \mid \mathbf{Z})$. Intuitively, as $H(\mathbf{Z}|\mathbf{Y})$ approaches zero, the peaks in the posterior become sharper, meaning the model assigns higher maximum probability to the correct class. This provides the first link between minimising irrelevant information and achieving higher confidence on ID predictions. A full proof is provided in Appendix B.2.

> **Theorem 5.2. (ID Confidence Improvement for IB over SL)**
> *Let* $\mathbf{Z} = f(\mathbf{X}, \mathbf{A})$ *be an encoded representation of* $(\mathbf{X}, \mathbf{A})$. *With information bottleneck:*
> $$\max I(\mathbf{Z}; \mathbf{Y}) - \beta I(\mathbf{X}, \mathbf{A}; \mathbf{Z}),$$
> *where* $\beta \in (0, 1)$, *the model achieves higher prediction confidence on in-distribution data compared to standard supervised learning (* $\beta = 0$ *), provided* $I(\mathbf{X}, \mathbf{A}; \mathbf{Z} \mid \mathbf{Y})$ *is minimised.*

Theorem 5.2 builds directly on Lemma 5.1. By rewriting the IB objective, we obtain:

$$\max(1 - \beta)I(\mathbf{Z}; \mathbf{Y}) - \beta I(\mathbf{X}, \mathbf{A}; \mathbf{Z} \mid \mathbf{Y}). \tag{14}$$

Theorem 5.2 therefore guarantees that models trained with IB achieve higher prediction confidence on ID data than standard SL (e.g., $\beta = 0$). The proof is provided in Appendix B.3.

> **Proposition 5.3. (IB Objective Increases Entropy Separation between ID and OOD)**
> *Let* $\mathbf{Z}^*$ *be the representation obtained from an optimal network trained with ID data via the IB objective (Eq. 36). Then:*
> 1. $I(\mathbf{X}_{id}, \mathbf{A}_{id}; \mathbf{Z}_{id}^* \mid \mathbf{Y}) \to 0$,
> 2. $I(\mathcal{G}_{OOD}; \mathbf{Z}_{ood}^* \mid \mathbf{Y}) \geq I(\mathbf{X}_{id}, \mathbf{A}_{id}; \mathbf{Z}_{id}^* \mid \mathbf{Y})$.
>
> *This induces entropy separation:*
> $$H(\mathbf{Y} \mid \mathbf{Z}_{id}^*) \ll H(\mathbf{Y} \mid \mathbf{Z}_{ood}^*),$$
> *enabling improved OOD detection via logit-based scores.*

**Proposition 5.3 formalises how the advantages of IB for ID confidence translate into OOD detection benefits.** The proof is provided in Appendix B.4. For ID data, point (1) follows from Theorem 5.2: the conditional mutual information vanishes, which yields low entropy predictions and sharp confidence. For OOD data, point (2) shows that the information compressed into $\mathbf{Z}$ inevitably contains more irrelevant content, resulting in higher conditional entropy and lower confidence. The contrast between these two cases induces a larger entropy gap between ID and OOD data under IB training than under SL training. This entropy separation directly improves logit-based OOD scores such as energy, making IB-trained models naturally better suited for OOD detection. Empirical evaluations in Section 6 validate this theoretical analysis for OOD detection.

## 6 EXPERIMENTS

**Datasets.** Following Wu et al. (2023b); Yang et al. (2024b), we evaluate on seven benchmarks. Six are single-graph datasets: `Cora`, `Citeseer`, `Pubmed`, `Amazon-Photo`, `Coauthor-CS`, and `ogbn-Arxiv`, where OOD nodes are generated via structure manipulation, feature interpolation, label exclusion, or temporal splits. We also use the multi-graph `TwitchGamers-Explicit`, where OOD is defined as graphs from different regions. Dataset splits are in Appendix K.

**Baselines.** To **fairly evaluate our information decomposition and IB approach**, we select **SOTA baselines trained with standard SL**: General OOD methods: MSP (Hendrycks & Gimpel, 2017), ODIN (Liang et al., 2018), Mahalanobis (Lee et al., 2018b), Energy (Liu et al., 2020). Graph-specific methods: GKDE (Zhao et al., 2020), GPN (Stadler et al., 2021), GNNSAFE (Wu et al., 2023b), NODESAFE (Yang et al., 2024b). OOD exposure methods: OE (Hendrycks et al., 2019), Energy FT (Liu et al., 2020), GNNSAFE++ (Wu et al., 2023b), NODESAFE++ (Yang et al., 2024b).

**Metrics.** Following Wu et al. (2023b), OOD detection is measured by **AUROC** ($\uparrow$), **AUPR** ($\uparrow$), and **FPR95** ($\downarrow$). ID classification accuracy (**ID ACC**) ($\uparrow$) is also reported. See Appendix I for details.

**Implementation.** All methods use a two-layer GCN backbone (hidden size 64). The feature and structure networks are implemented as an MLP and a GCN, respectively, with the same architecture. For the structure network, node features are fixed to $\mathbf{1}$, while the feature network uses only raw node attributes. Additional setups, hyperparameters, and sensitivity analyses are provided in Appendix J.

### 6.1 OVERALL PERFORMANCE

**As a non-OOD exposed method, TRIBE markedly outperforms SOTA non-OOD exposure baselines.** Evident in Table 1, TRIBE significantly enhances OOD detection performance across all datasets (blue highlights). On synthetic datasets such as `Citeseer` and `Pubmed`, TRIBE

Table 1: Model performance in Non-OOD exposure. Following Wu et al. (2023b); Yang et al. (2024b), the results are **averaged on multiple OOD test sets with different difficulty level**, therefore with a relatively high variance $\pm$ across subsets. Individual subset results with a much lower variance are in Appendix L. The best and runner-up results are highlighted by **best** and **runner-up**, respectively.

| | Metrics | MSP | ODIN | Maha | Energy | GKDE | GPN | GNNSAFE | NODESAFE | TRIBE |
|---|---|---|---|---|---|---|---|---|---|---|
| Cora | AUROC (↑) | 82.55 | 49.87 | 54.74 | 83.09 | 69.54 | 84.56 | 91.20 ± 3.09 | 93.35 ± 2.41 | 95.58 ± 2.12 |
| | AUPR (↑) | 65.82 | 26.08 | 34.43 | 66.21 | 46.09 | 68.02 | 82.92 ± 4.96 | 84.64 ± 6.40 | 89.34 ± 5.32 |
| | FPR95 (↓) | 62.39 | 100.00 | 96.30 | 65.21 | 80.51 | 58.30 | 50.53 ± 22.04 | 29.23 ± 12.06 | 20.09 ± 10.72 |
| | ID ACC (↑) | 79.91 | 79.61 | 79.57 | 80.34 | 79.86 | 81.65 | 81.56 ± 7.01 | 82.66 ± 6.61 | 82.56 ± 6.99 |
| Citeseer | AUROC (↑) | 77.69 | 50.14 | 49.55 | 78.26 | 72.95 | 78.22 | 84.89 ± 5.47 | 87.80 ± 2.74 | 92.27 ± 1.53 |
| | AUPR (↑) | 51.43 | 21.38 | 29.29 | 51.22 | 47.67 | 52.22 | 64.86 ± 3.39 | 72.35 ± 6.41 | 76.57 ± 10.34 |
| | FPR95 (↓) | 69.42 | 100 | 95.06 | 65.34 | 71.85 | 72.09 | 60.85 ± 18.06 | 53.38 ± 17.47 | 30.79 ± 7.64 |
| | ID ACC (↑) | 73.72 | 73.75 | 62.17 | 73.43 | 72.69 | 72.77 | 76.06 ± 12.05 | 73.12 ± 14.15 | 75.66 ± 11.19 |
| Pubmed | AUROC (↑) | 78.80 | 49.72 | 62.20 | 79.25 | 78.14 | 78.76 | 93.82 ± 1.93 | 91.08 ± 2.95 | 97.35 ± 0.13 |
| | AUPR (↑) | 28.37 | 4.83 | 11.74 | 28.21 | 24.65 | 28.65 | 69.94 ± 6.32 | 68.56 ± 1.06 | 80.73 ± 1.37 |
| | FPR95 (↓) | 76.73 | 100 | 91.26 | 70.69 | 75.04 | 71.06 | 37.71 ± 14.28 | 50.17 ± 0.89 | 12.75 ± 0.31 |
| | ID ACC (↑) | 75.05 | 75.30 | 71.15 | 75.55 | 74.65 | 75.15 | 76.78 ± 0.31 | 77.32 ± 0.02 | 76.17 ± 0.33 |
| Amazon | AUROC (↑) | 96.52 | 80.12 | 73.81 | 96.73 | 66.98 | 92.60 | 97.99 ± 0.93 | 97.84 ± 1.11 | 98.16 ± 1.14 |
| | AUPR (↑) | 95.01 | 77.18 | 72.35 | 95.16 | 71.18 | 90.50 | 98.16 ± 1.56 | 97.77 ± 2.11 | 98.08 ± 1.99 |
| | FPR95 (↓) | 13.83 | 85.22 | 83.44 | 13.15 | 98.47 | 32.64 | 3.24 ± 5.24 | 3.69 ± 6.14 | 2.81 ± 4.61 |
| | ID ACC (↑) | 93.83 | 93.88 | 93.80 | 93.85 | 87.71 | 89.54 | 93.79 ± 1.99 | 92.70 ± 2.16 | 93.90 ± 1.88 |
| Coauthor | AUROC (↑) | 95.74 | 51.71 | 82.02 | 96.64 | 69.24 | 69.89 | 98.98 ± 1.41 | 99.02 ± 1.45 | 99.27 ± 1.17 |
| | AUPR (↑) | 96.43 | 56.37 | 87.05 | 97.09 | 80.17 | 72.77 | 99.55 ± 0.44 | 99.57 ± 0.47 | 99.70 ± 0.40 |
| | FPR95 (↓) | 21.37 | 99.97 | 48.09 | 15.49 | 97.04 | 69.60 | 4.29 ± 6.87 | 4.33 ± 7.04 | 3.31 ± 5.45 |
| | ID ACC (↑) | 93.37 | 93.29 | 93.29 | 93.57 | 87.74 | 89.39 | 93.21 ± 1.50 | 93.21 ± 1.86 | 93.48 ± 1.74 |
| Twitch | AUROC (↑) | 33.59 | 58.16 | 55.68 | 51.24 | 46.48 | 51.73 | 66.33 ± 15.32 | 66.48 ± 15.44 | 89.72 ± 5.42 |
| | AUPR (↑) | 49.14 | 72.12 | 66.42 | 60.81 | 62.11 | 66.36 | 72.59 ± 13.44 | 72.71 ± 13.43 | 91.92 ± 3.85 |
| | FPR95 (↓) | 97.45 | 93.96 | 90.13 | 91.61 | 95.62 | 95.51 | 81.18 ± 19.87 | 80.62 ± 20.03 | 46.60 ± 26.47 |
| | ID ACC (↑) | 68.72 | 70.79 | 70.51 | 70.40 | 67.44 | 68.09 | 70.75 ± 0.30 | 70.75 ± 0.33 | 68.63 ± 1.41 |
| Arxiv | AUROC (↑) | 63.91 | 55.07 | 56.92 | 64.20 | 58.32 | OOM | 70.58 ± 6.41 | 71.36 ± 6.35 | 72.72 ± 6.48 |
| | AUPR (↑) | 75.85 | 68.85 | 69.63 | 75.78 | 72.62 | OOM | 79.99 ± 12.80 | 80.67 ± 12.37 | 81.62 ± 11.86 |
| | FPR95 (↓) | 90.59 | 100.0 | 94.24 | 90.80 | 93.84 | OOM | 87.90 ± 2.57 | 86.45 ± 2.97 | 83.65 ± 4.14 |
| | ID ACC (↑) | 53.78 | 51.39 | 51.59 | 53.36 | 50.76 | OOM | 53.21 ± 0.25 | 53.10 ± 0.21 | 52.44 ± 0.24 |

reduces FPR95 by an average of 24%. While datasets like Amazon and Coauthor exhibit strong performance with existing SOTA baselines, driven by high classification accuracy, which yield more discriminative energy scores for OOD detection - TRIBE further improves detection performance. This shows the efficacy of our dedicated information learning and IB over the standard SL in GNNSAFE. For real-world datasets, TRIBE delivers remarkable improvements, increasing the AUROC score by over 23% and reducing the FPR95 by 34% on the Twitch dataset. On the more challenging Arxiv dataset, where performance is constrained by limited classification accuracy, TRIBE still outperforms baseline methods. This highlights TRIBE's superiority in OOD detection while maintaining strong ID classification accuracy. Extended results are in Appendix L. Moreover, TRIBE achieves **similar efficiency in inference speed and memory usage** as SOTA methods with discussions in Appendix M.

## 6.2 IB VS. SL IN OOD DETECTION

**The empirical results strongly validate IB's superiority over SL for OOD detection.** Table 2 compares SL-based and IB-based classifiers, with energy as the OOD score. Even in its simplest form, IB-based models consistently outperform their SL counterparts (SL < IB), highlighting IB's fundamental advantages for OOD detection. When enhanced with effective energy propagation techniques (E. Prop.), the performance gap widens further, with IB models showing significantly greater improvements.

Table 2: **Detection comparison between IB vs. SL.** E. Prop. denotes using energy propagation.

| | Metrics | w/o E. Prop. | | w/ E. Prop. | |
|---|---|---|---|---|---|
| | | SL | IB | SL | IB |
| Cora | AUROC (↑) | 83.09 | 86.90 | 91.20 | 95.18 |
| | AUPR (↑) | 66.21 | 69.89 | 82.92 | 88.49 |
| | FPR95 (↓) | 65.21 | 46.26 | 50.53 | 22.20 |
| Pubmed | AUROC (↑) | 79.25 | 87.04 | 84.49 | 91.50 |
| | AUPR (↑) | 29.21 | 45.87 | 64.86 | 75.22 |
| | FPR95 (↓) | 70.69 | 52.73 | 60.85 | 37.42 |
| Twitch | AUROC (↑) | 51.24 | 70.59 | 66.33 | 73.50 |
| | AUPR (↑) | 60.81 | 73.26 | 72.59 | 76.67 |
| | FPR95 (↓) | 91.61 | 82.46 | 81.18 | 67.26 |

## 6.3 EXTENDED OOD EXPOSURE STUDY

**TRIBE, when equipped with OOD exposure regularisation via Eq. 4, achieves superior or competitive performance against SOTA baselines.** Shown in Table 3, TRIBE outperforms GNNSAFE++ and NODESAFE++ across multiple datasets, as highlighted in bold. For example, on Cora, TRIBE achieves an AUROC of 95.76 (vs. 93.13 & 91.32) and significantly reduces the FPR95 from 42.48 down to 19.34. On larger Pubmed, TRIBE improves AUROC to 98.26 (vs. 93.77 &

Table 4: Comparison of TRIBE with advanced graph OOD detection. "+TRIBE" denotes incorporating our TRIBE objective into the compared methods.

| Model | Cora – Label | | | | Pubmed – S | | | | Pubmed – F | | | | Amazon – F | | | | Citeseer – S | | | |
|---|---|---|---|---|---|---|---|---|---|---|---|---|---|---|---|---|---|---|---|---|
| | AUROC | AUPR | FPR95 | ID Acc | AUROC | AUPR | FPR95 | ID Acc | AUROC | AUPR | FPR95 | ID Acc | AUROC | AUPR | FPR95 | ID Acc | AUROC | AUPR | FPR95 | ID Acc |
| GNNSAFE | 93.19 | 82.99 | 29.55 | 89.66 | 92.45 | 65.47 | 47.81 | 77.00 | 95.18 | 74.41 | 27.62 | 76.57 | 98.46 | 98.90 | 0.44 | 92.65 | 79.87 | 61.44 | 74.45 | 65.20 |
| TRIBE | 93.70 | 84.03 | 28.84 | 90.61 | 97.25 | 79.76 | 12.97 | 75.93 | 97.44 | 81.70 | 12.53 | 76.40 | 98.65 | 99.04 | 0.30 | 92.70 | 91.89 | 80.11 | 38.41 | 70.03 |
| DeGEM | 92.24 | 78.80 | 31.34 | 91.77 | 95.37 | 51.55 | 20.07 | 73.10 | 99.63 | 93.40 | 1.70 | 79.00 | 97.34 | 96.70 | 3.16 | 91.65 | 94.93 | 84.63 | 17.49 | 70.90 |
| + TRIBE | 95.85 | 89.24 | 20.59 | 92.41 | 97.20 | 55.60 | 8.44 | 78.40 | 99.67 | 91.73 | 0.59 | 78.70 | 97.71 | 96.49 | 1.84 | 92.32 | 96.92 | 85.63 | 7.24 | 69.70 |
| GOLD | 95.36 | 85.33 | 21.20 | 89.56 | 88.01 | 97.84 | 11.57 | 75.30 | 87.28 | 98.49 | 6.98 | 73.20 | 99.52 | 99.63 | 0.24 | 92.48 | 78.09 | 82.12 | 65.98 | 69.10 |
| + TRIBE | 94.85 | 83.96 | 18.86 | 89.56 | 91.13 | 98.72 | 5.52 | 74.60 | 93.15 | 99.00 | 3.80 | 74.70 | 99.61 | 99.70 | 0.10 | 91.98 | 78.61 | 82.90 | 44.97 | 68.70 |

95.14) and decreases FPR95 to 8.84 (vs. 39.58 & 24.87). On more challenging real-world OOD datasets, TRIBE remains competitive on the `Twitch` and `Arxiv` datasets. This highlights TRIBE's adaptability to incorporate OOD exposure strategies, enabling it to achieve superior performance.

## 6.4 EXTENDED COMPARISONS WITH ADVANCED GRAPH OOD DETECTORS

To further demonstrate the effectiveness of TRIBE and the role of the IB objective, we compare against two advanced graph OOD detection baselines, DeGEM (Chen et al., 2025) and GOLD (Wang et al., 2025a), on Cora and Pubmed under structure shift, feature shift, and label shift settings in Table 4. All results were reproduced to the best of our ability; deviations from the original reports may arise from differences in data splits, random seeds, or unavailable hyperparameters. Both baselines exhibit strong performance across several settings, reflecting the strength of their respective designs. To enable a fairer comparison with our IB-driven analysis, we additionally evaluate "+TRIBE" variants of GOLD and DeGEM by incorporating our TRIBE objective during training. As shown

Table 3: Comparisons of OOD exposure with **best** and runner-up highlighted. GS++ and NS++ are short for GNNSafe++ and NODESAFE++.

| | Metrics | OOD Exposure | | | | TRIBE w/ OE |
|---|---|---|---|---|---|---|
| | | OE | Energy FT | GS++ | NS++ | |
| Cora | AUROC (↑) | 79.76 | 85.13 | 93.13 | 91.32 | 95.76 |
| | AUPR (↑) | 64.93 | 67.89 | 85.28 | 82.74 | 89.68 |
| | FPR95 (↓) | 75.22 | 51.03 | 37.28 | 42.48 | 19.34 |
| | ID ACC (↑) | 77.69 | 80.44 | 82.16 | 74.94 | 82.20 |
| Citeseer | AUROC (↑) | 63.75 | 79.81 | 85.69 | 84.29 | 92.16 |
| | AUPR (↑) | 47.20 | 52.79 | 65.89 | 64.86 | 75.90 |
| | FPR95 (↓) | 74.15 | 57.37 | 54.76 | 56.12 | 29.22 |
| | ID ACC (↑) | 62.24 | 72.66 | 72.74 | 68.60 | 75.81 |
| Pubmed | AUROC (↑) | 78.38 | 76.25 | 93.77 | 95.14 | 98.26 |
| | AUPR (↑) | 27.67 | 27.61 | 74.06 | 72.20 | 85.36 |
| | FPR95 (↓) | 79.05 | 91.02 | 39.58 | 24.87 | 8.84 |
| | ID ACC (↑) | 73.00 | 75.80 | 77.88 | 74.17 | 76.67 |
| Twitch | AUROC (↑) | 55.72 | 84.50 | 95.76 | 76.79 | 91.52 |
| | AUPR (↑) | 70.18 | 88.04 | 97.45 | 83.97 | 93.59 |
| | FPR95 (↓) | 95.07 | 61.29 | 29.81 | 34.46 | 33.19 |
| | ID ACC (↑) | 70.73 | 70.52 | 70.36 | 70.07 | 68.74 |
| Arxiv | AUROC (↑) | 69.80 | 71.56 | 73.98 | 74.50 | 74.89 |
| | AUPR (↑) | 80.15 | 80.47 | 82.50 | 81.55 | 83.25 |
| | FPR95 (↓) | 85.16 | 80.59 | 79.99 | 77.53 | 77.37 |
| | ID ACC (↑) | 52.39 | 53.26 | 53.28 | 51.32 | 52.41 |

in the table, these TRIBE-augmented versions often improve over their base models in AUROC, AUPR, or FPR95, supporting our theoretical finding that IB increases ID-OOD separation. Across all datasets and shift types, TRIBE consistently matches or surpasses the competitive methods, highlighting the benefit of explicitly decomposing joint, feature-specific, and structure-specific information for stable graph OOD detection.

## 6.5 ABLATION STUDY

We conduct an ablation study on the IB backbone, a conditional independence constraint, and the full TRIBE framework, which includes a pairwise mutual information minimisation loss to reduce spurious correlations. Compared to GNNSAFE (standard supervised learning, $\max I(\mathbf{Z}; \mathbf{Y})$), the IB backbone lowers FPR95 by 20% on average, showing the value of compressing irrelevant features. The independence

Table 5: Ablation study.

| | Metrics | Cora | Citeseer | Pudmed | Twitch | Arxiv |
|---|---|---|---|---|---|---|
| GNNSAFE ($\mathcal{L}_{sup}$) | AUROC (↑) | 91.20 | 84.89 | 93.82 | 66.33 | 70.58 |
| | AUPR (↑) | 82.92 | 64.86 | 69.94 | 72.59 | 79.99 |
| | FPR95 (↓) | 50.53 | 60.85 | 37.71 | 81.18 | 87.90 |
| | ID ACC (↑) | 81.56 | 76.06 | 76.78 | 70.75 | 53.21 |
| $\mathcal{L}_{VIB}$ | AUROC (↑) | 95.18 | 91.50 | 96.58 | 73.50 | 72.15 |
| | AUPR (↑) | 88.49 | 75.22 | 77.34 | 76.67 | 81.13 |
| | FPR95 (↓) | 22.20 | 37.42 | 16.85 | 67.26 | 84.93 |
| | ID ACC (↑) | 82.15 | 75.39 | 74.95 | 70.36 | 52.46 |
| $\mathcal{L}_{VIB}$ & $\mathcal{L}_{VCInd}$ | AUROC (↑) | 95.23 | 91.99 | 96.94 | 87.61 | 72.68 |
| | AUPR (↑) | 88.65 | 76.01 | 78.77 | 90.61 | 81.57 |
| | FPR95 (↓) | 21.56 | 32.57 | 14.37 | 49.91 | 83.86 |
| | ID ACC (↑) | 82.12 | 75.24 | 75.18 | 70.33 | 52.43 |
| TRIBE | AUROC (↑) | 95.58 | 92.27 | 97.35 | 89.72 | 72.72 |
| | AUPR (↑) | 89.34 | 76.57 | 80.73 | 91.92 | 81.62 |
| | FPR95 (↓) | 20.09 | 30.79 | 12.75 | 46.60 | 83.65 |
| | ID ACC (↑) | 82.56 | 75.66 | 76.17 | 68.63 | 52.44 |

constraint further improves performance, reducing FPR95 by 17% on the diverse `Twitch` dataset, by better modeling joint inputs. With all components, TRIBE achieves the best performing results, learning robust representations for both ID classification and OOD detection. Though gains over the IB backbone are modest, the added constraints improve stability, supporting our claim that the IB objective is better suited for OOD detection. Detailed results are provided in Appendix L.

## 6.6 VISUALISATIONS

**Energy Gap.** To further validate the effectiveness of our information-decomposed framework and highlight the advantage of IB over the conventional SL baseline for OOD detection, we visualise

the **prediction confidence and energy distributions** of TRIBE and GNNSAFE in Figure 3. The first subplot supports the theoretical insight that the IB objective enhances ID prediction confidence. Specifically, the blue distribution of TRIBE is skewed toward higher confidence values compared to the green distribution of GNNSAFE. Additionally, the third subplot shows that **TRIBE achieves a greater separation between ID and OOD energy scores than GNNSAFE**, as seen in the second subplot. This increased energy margin highlights TRIBE's effectiveness in distinguishing ID from OOD data. Furthermore, the dashed lines denote the FPR95 threshold, where TRIBE reduces the overlap between ID and OOD energy scores to the left of the threshold, reflecting the effect of increased ID prediction confidence on OOD scores. These findings validate TRIBE's improved OOD detection ability while preserving high ID accuracy.

Connecting this visualisation to our theory, recall from Proposition 5.3 that the IB objective increases ID prediction confidence and enlarges the **entropy gap** between ID and OOD compared to standard supervised learning. This analysis is stated in terms of the predictive distribution $p_\theta(\mathbf{y} \mid \mathbf{z}) = \text{Softmax}(\ell)$ and its entropy $H(\mathbf{Y} \mid \mathbf{Z})$, where $\ell$ are the nodes logits. The energy score we use for OOD detection, $S(\mathbf{x}, \mathbf{A}; \text{GNN}) = \mathbf{e} = -\log \sum_c \exp(\ell_c)$ (Eq. 2), is computed from the same logits and depends on the same log-partition term that appears in the predictive entropy. Consequently, the IB-induced change in logit geometry that sharpens ID posteriors (higher confidence, lower entropy) and makes OOD posteriors more diffuse (higher entropy) naturally manifests as **lower energies for ID nodes and higher energies for OOD nodes**, as demonstrated in Figure 3.

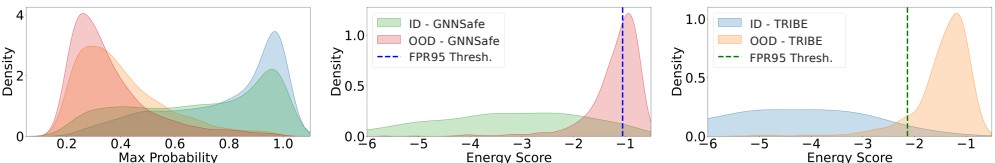

Figure 3: Comparison of **(1) Prediction Confidence** and **Energy Score distribution** of **(2) GNNSAFE** and **(3) TRIBE** on `Cora-Feature`. The dashed line is the FPR95 threshold.

**Representation Distribution.** With the tri-component information decomposition framework, **TRIBE learns more shift-indicative representations through its optimised joint-input network.** In Figure 4, we compare ID and OOD representation distributions from **(1) a standard SL-trained GCN** and from the **(2) joint**, **(3) structure**, and **(4) feature** networks of TRIBE. Notably, the joint network achieves clearer ID-OOD separation than the baseline GCN. On the structure-shifted dataset (Cora-S), the feature network shows minimal separation (as expected), while the structure network captures a distinct, separable topology pattern.

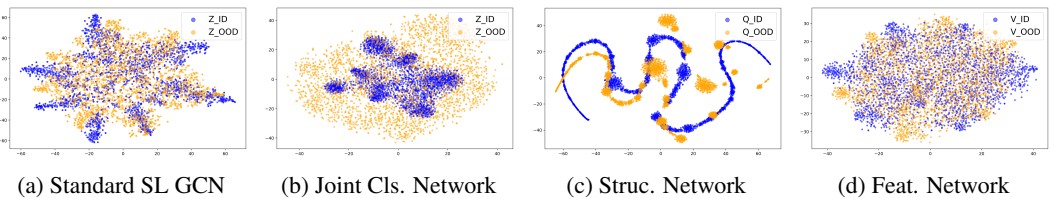

(a) Standard SL GCN     (b) Joint Cls. Network     (c) Struc. Network     (d) Feat. Network

Figure 4: Distribution visualisation comparing ID vs. OOD representations between a standard **SL trained GCN** and **TRIBE's different networks** on **Cora-Structure** with structural shift.

# 7 CONCLUSION

We propose TRIBE, a tri-component information decomposition framework designed to effectively detect OOD instances of graph-structured data. By decomposing information into structural, feature, and joint components with an IB objective, TRIBE effectively retains shift-indicative joint-input label-specific information while mitigating label-irrelevant and spurious correlations within individual input components. Additionally, we provide theoretical insights into the advantages of the IB principle over the standard supervised learning objective for in-distribution classification and detection, particularly emphasising its ability to enhance prediction confidence and its implications for improving OOD detection. Extensive experiments validate the efficacy of TRIBE, which outperforms SOTA SL-based OOD detection methods, including both non-OOD-exposed and OOD-exposed approaches.

## 8 Impact and Ethics Statement

We hope our work inspires future research on node-level graph OOD detection in real-world settings. As a foundational study on OOD detection for graph-structured data, we do not identify any direct negative societal impacts. Our research relies on publicly available datasets, algorithms, and models, all of which are properly acknowledged and pose no risks requiring safeguards. While potential societal implications may exist, none warrant specific emphasis in this study.

**LLM Usage.** LLMs contributed only to polishing the writing.

## 9 Reproducibility Statement

To support reproducible research, we summarise our efforts as below:

1. **Baselines & Datasets.** We follow the baseline from (Wu et al., 2023b) and utilise publicly available datasets. The details are described in Section 6 and Appendix K.

2. **Model training.** Detailed implementation setting is provided in Section 6 and Appendix J.

3. **Evaluation Metrics.** We discuss the evaluation metrics used in Section 6 and Appendix I.

4. **Methodology.** Our TRIBE framework is fully documented in Section 4, with implementation notes provided in Section 4.3. To support implementation and reproducibility we provide a detailed discussion on obtaining the **tractable objective** in Appendix E, a **neural network parameterisation** of TRIBE is given in Appendix F. In addition, we provide a detailed pseudo code of the optimisation process in Algorithm 1 and the inference process in Algorithm 2. We provide theoretical analysis and corresponding proofs in Section 5 and Appendix B.

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

# APPENDIX

## A   POTENTIAL LIMITATIONS

In Section 4.1, we illustrate that TRIBE employs two additional networks on top of the GNN classifier to effectively capture structural and feature information: a GNN for modelling structural relationships and an MLP for processing feature-based information. This multi-network architecture inevitably incurs additional computational and memory overhead during training. However, during inference, our model achieves comparable inference speeds while delivering significant performance improvements across all evaluated datasets. We argue that the increased training cost is a justifiable trade-off given the substantial gains in detection performance. The detailed computational cost is discussed in Appendix M. Described in Appendix J, a constant $\alpha$ and $\beta$ values were applied uniformly in the current experiments, future work will explore fine-grained control of individual terms. Furthermore, currently implementation only considers a simple structure representation learning strategy (i.e., using a constant feature vector), we will investigate the use of more advanced position encoding and structure representation learning techniques in future work. This may include methods like spectral embeddings via Laplacian eigenmaps, random walk-based encodings, and learnable position encodings etc. Since our TRIBE focuses on exploring the advantages of IB over standard supervised learning, we have left the comparison and integration with more advanced graph OOD detection methods and novel training frameworks like DeGEM and GOLD as future work (Chen et al., 2025; Wang et al., 2025a). Moreover, our proposed method and theoretical analysis are currently focused on node-level classification and we do not study the anomaly detection task. However, we believe the framework can be extended to graph-level OOD detection. Potential directions include applying the IB principle to the representation of entire graphs or learning compact and informative subgraphs, as explored in prior works (Dai et al., 2023; Sun et al., 2022). We also aim to investigate the impact of this framework in the context of heterophilic graphs. These extensions present promising avenues for future research.

## B   PROOF OF TECHNICAL INSIGHTS

### B.1   PROOF OF PROPOSITION 4.1

*Proof.* Notice that given a sample $(\mathbf{x}_i, \mathbf{a}_i, \mathbf{y}_i) \in \mathbf{X}, \mathbf{A}, \mathbf{Y}$, the associated distribution $\mathbf{z}$ is obtained by $f_\theta(\mathbf{z}, \mathbf{x}_i, \mathbf{a}_i) = P(\mathbf{z}|\mathbf{x}_i, \mathbf{a}_i, \mathbf{y}_i; \theta)$, where $f_\theta$ is neural network parametrised by $\theta$. Using the empirical data distribution, we can approximate the joint distribution $P(\mathbf{x}, \mathbf{a}, \mathbf{y}, \mathbf{z}; \theta)$ as:

$$P(\mathbf{x}, \mathbf{a}, \mathbf{y}, \mathbf{z}; \theta) = \frac{1}{N} \sum_{i=1}^{N} \delta_{\mathbf{x}_i}(\mathbf{x}) \delta_{\mathbf{a}_i}(\mathbf{a}) \delta_{\mathbf{y}_i}(\mathbf{y}) f_\theta(\mathbf{z}, \mathbf{x}_i, \mathbf{a}_i),$$

where $\delta(\cdot)$ is the Dirac delta function.

Furthermore, we can also derive the marginal distributions as:

$$P(\mathbf{x}, \mathbf{a}, \mathbf{z}; \theta) = \int P(\mathbf{x}, \mathbf{a}, \mathbf{y}, \mathbf{z}; \theta) \, d\mathbf{y} = \frac{1}{N} \sum_{i=1}^{N} \delta_{\mathbf{x}_i}(\mathbf{x}) \delta_{\mathbf{a}_i}(\mathbf{a}) f_\theta(\mathbf{z}, \mathbf{x}_i, \mathbf{a}_i)$$

$$P(\mathbf{x}, \mathbf{a}, \mathbf{y}; \theta) = \int P(\mathbf{x}, \mathbf{a}, \mathbf{y}, \mathbf{z}; \theta) \, d\mathbf{z} = \frac{1}{N} \sum_{i=1}^{N} \delta_{\mathbf{x}_i}(\mathbf{x}) \delta_{\mathbf{a}_i}(\mathbf{a}) \delta_{\mathbf{y}_i}(\mathbf{y})$$

$$P(\mathbf{x}, \mathbf{a}; \theta) = \int \int P(\mathbf{x}, \mathbf{a}, \mathbf{y}, \mathbf{z}; \theta) \, d\mathbf{y} \, d\mathbf{z} = \frac{1}{N} \sum_{i=1}^{N} \delta_{\mathbf{x}_i}(\mathbf{x}) \delta_{\mathbf{a}_i}(\mathbf{a})$$

The conditional mutual information $I(\mathbf{z}; \mathbf{y}|\mathbf{x}, \mathbf{a})$ is expressed as:

$$I(\mathbf{z}; \mathbf{y}|\mathbf{x}, \mathbf{a}) = \int \int \int \int P(\mathbf{x}, \mathbf{a}, \mathbf{y}, \mathbf{z}; \theta) \log \left( \frac{P(\mathbf{x}, \mathbf{a}; \theta) P(\mathbf{x}, \mathbf{a}, \mathbf{y}, \mathbf{z}; \theta)}{P(\mathbf{x}, \mathbf{a}, \mathbf{y}; \theta) P(\mathbf{x}, \mathbf{a}, \mathbf{z}; \theta)} \right) d\mathbf{x} \, d\mathbf{a} \, d\mathbf{y} \, d\mathbf{z}.$$

Substituting the empirical distributions we have:

$$= \frac{1}{N} \int \int \int \int \left( \sum_{i=1}^{N} \delta_{\mathbf{x}_i}(\mathbf{x}) \delta_{\mathbf{a}_i}(\mathbf{a}) \delta_{\mathbf{y}_i}(\mathbf{y}) f_\theta(\mathbf{z}, \mathbf{x}_i, \mathbf{a}_i) \right) \times$$

$$\log \left( \frac{\left( \sum_{i=1}^{N} \delta_{\mathbf{x}_i}(\mathbf{x}) \delta_{\mathbf{a}_i}(\mathbf{a}) \right) \left( \sum_{i=1}^{N} \delta_{\mathbf{x}_i}(\mathbf{x}) \delta_{\mathbf{a}_i}(\mathbf{a}) \delta_{\mathbf{y}_i}(\mathbf{y}) f_\theta(\mathbf{z}, \mathbf{x}_i, \mathbf{a}_i) \right)}{\left( \sum_{i=1}^{N} \delta_{\mathbf{x}_i}(\mathbf{x}) \delta_{\mathbf{a}_i}(\mathbf{a}) \delta_{\mathbf{y}_i}(\mathbf{y}) \right) \left( \sum_{i=1}^{N} \delta_{\mathbf{x}_i}(\mathbf{x}) \delta_{\mathbf{a}_i}(\mathbf{a}) f_\theta(\mathbf{z}, \mathbf{x}_i, \mathbf{a}_i) \right)} \right) d\mathbf{x} \, d\mathbf{a} \, d\mathbf{y} \, d\mathbf{z}.$$

Simplify using the sifting property of Dirac delta functions, the logarithm term becomes:

$$\log \left( \frac{f_\theta(\mathbf{z}, \mathbf{x}_i, \mathbf{a}_i)}{f_\theta(\mathbf{z}, \mathbf{x}_i, \mathbf{a}_i)} \right) = \log(1) = 0.$$

Thus, we have $I(\mathbf{z}; \mathbf{y}|\mathbf{x}, \mathbf{a}) = 0$, regardless of the parameters $\theta$. This indicates that the information of $\mathbf{Z}$ is contained by the information of $(\mathbf{X}, \mathbf{A})$. This result holds because $\mathbf{Z}$ is a obtained from $(\mathbf{X}, \mathbf{A})$, and $\mathbf{Z}$ does not provide any additional information about $\mathbf{Y}$ beyond $(\mathbf{X}, \mathbf{A})$. $\qquad\square$

## B.2 PROOF OF LEMMA 5.1

*Proof.* Consider a deterministic encoder with encoded representation $\mathbf{Z} = f(\mathbf{X}, \mathbf{A})$, the conditional mutual information $I(\mathbf{X}, \mathbf{A}; \mathbf{Z} \mid \mathbf{Y})$ can be expanded as:

$$\begin{aligned} I(\mathbf{X}, \mathbf{A}; \mathbf{Z} \mid \mathbf{Y}) &= H(\mathbf{X}, \mathbf{A} \mid \mathbf{Y}) + H(\mathbf{Z} \mid \mathbf{Y}) - H(\mathbf{X}, \mathbf{A}, \mathbf{Z} \mid \mathbf{Y}) \\ &= H(\mathbf{X}, \mathbf{A} \mid \mathbf{Y}) + H(\mathbf{Z} \mid \mathbf{Y}) - \big(H(\mathbf{X}, \mathbf{A} \mid \mathbf{Y}) + H(\mathbf{Z} \mid \mathbf{X}, \mathbf{A}, \mathbf{Y})\big) \quad (15) \\ &= H(\mathbf{Z} \mid \mathbf{Y}) - H(\mathbf{Z} \mid \mathbf{X}, \mathbf{A}, \mathbf{Y}). \end{aligned}$$

Since $\mathbf{Z}$ is a determined by $\mathbf{X}$ and $\mathbf{A}$, the second term vanishes:

$$H(\mathbf{Z} \mid \mathbf{X}, \mathbf{A}, \mathbf{Y}) = 0 \quad \Rightarrow \quad I(\mathbf{X}, \mathbf{A}; \mathbf{Z} \mid \mathbf{Y}) = H(\mathbf{Z} \mid \mathbf{Y}). \quad (16)$$

Thus, minimising $I(\mathbf{X}, \mathbf{A}; \mathbf{Z} \mid \mathbf{Y})$ is equivalent to minimising $H(\mathbf{Z} \mid \mathbf{Y})$.

Notably, reducing $H(\mathbf{Z} \mid \mathbf{Y})$ implies the conditional distribution $P(\mathbf{Z} \mid \mathbf{Y})$ becomes more concentrated (i.e., $\mathbf{Z}$ is more predictable given $\mathbf{Y}$ (reduced uncertainty)). Applying Bayes' theorem, we have:

$$P(\mathbf{Y} \mid \mathbf{Z}) = \frac{P(\mathbf{Z} \mid \mathbf{Y}) P(\mathbf{Y})}{P(\mathbf{Z})}. \quad (17)$$

Here, a more concentrated $P(\mathbf{Z} \mid \mathbf{Y})$ would amplify the likelihood term $P(\mathbf{Z} \mid \mathbf{Y})$ relative to the marginal $P(\mathbf{Z}) = \mathbb{E}_{\mathbf{Y}'}[P(\mathbf{Z} \mid \mathbf{Y}')]$. As a result, it produces sharper peaks in $P(\mathbf{Y} \mid \mathbf{Z})$, increasing the maximum probability $\max_{\mathbf{y}} P(\mathbf{Y} = \mathbf{y} \mid \mathbf{Z})$ and thus the prediction confidence of ID data, where:

$$H(\mathbf{Y} \mid \mathbf{Z}) = -\mathbb{E}_{\mathbf{Z}} \left[ \sum_y P(y \mid \mathbf{Z}) \log P(y \mid \mathbf{Z}) \right] \to 0. \quad (18)$$

$$\square$$

## B.3 PROOF OF THEOREM 5.2

*Proof.* To begin, using the chain rule of mutual information, we can decompose the second mutual information term in the IB objective as:

$$I(\mathbf{X}, \mathbf{A}; \mathbf{Z}) = I(\mathbf{Z}; \mathbf{Y}) + I(\mathbf{X}, \mathbf{A}; \mathbf{Z} \mid \mathbf{Y}). \quad (19)$$

Substituting into the IB objective yields:

$$\max(1 - \beta) I(\mathbf{Z}; \mathbf{Y}) - \beta I(\mathbf{X}, \mathbf{A}; \mathbf{Z} \mid \mathbf{Y}). \quad (20)$$

Notably, when $\beta = 0$, the objective reduces to the standard supervised learning goal of maximising $I(\mathbf{Z}; \mathbf{Y})$. However, this indicates that the superfluous information $I(\mathbf{X}, \mathbf{A}; \mathbf{Z} \mid \mathbf{Y})$ is preserved, meaning the encoded representation $\mathbf{Z}$ may contain information from $\mathbf{X}$ and $\mathbf{A}$ that is irrelevant to the label $\mathbf{Y}$. This can negatively impact predictive performance, as $\mathbf{Z}$ is not optimised to focus solely on label-relevant information.

In contrast, for $\beta \in (0, 1)$, the IB objective simultaneously increases predictive information $I(\mathbf{Z}; \mathbf{Y})$ and reduces superfluous information by minimising $I(\mathbf{X}, \mathbf{A}; \mathbf{Z} \mid \mathbf{Y})$. This ensures $\mathbf{Z}$ can capture useful information about $\mathbf{Y}$, while compressing out input information irrelevant to $\mathbf{Y}$. From Lemma 5.1, minimising $I(\mathbf{X}, \mathbf{A}; \mathbf{Z} \mid \mathbf{Y})$ reduces $H(\mathbf{Z} \mid \mathbf{Y})$. This reduction forces $\mathbf{Z}$ to discard information in $\mathbf{X}$ that is irrelevant to $\mathbf{Y}$, leading to a sharper conditional distribution $P(\mathbf{Y} \mid \mathbf{Z})$. Thus, when $P(\mathbf{Y} \mid \mathbf{Z})$ becomes sharper, $\max_{\mathbf{y}} P(\mathbf{Y} = \mathbf{y})$ increases, as the probability mass is concentrated on the most likely value of $\mathbf{Y}$. Consequently, the conditional entropy decreases:

$$H(\mathbf{Y} \mid \mathbf{Z}) = -\mathbb{E}_{\mathbf{Z}}[\max_{\mathbf{y}} P(\mathbf{Y} = \mathbf{y} \mid \mathbf{Z})] + \text{cross-entropy terms}, \tag{21}$$

thereby increasing prediction confidence. In other words, the model becomes more certain about its predictions, as $\mathbf{Z}$ is optimised to focus on the most relevant information for predicting $\mathbf{Y}$.

Additionally, an optimal trade-off occurs when $\beta$ balances information compression-to-prediction. Let $\Delta_Z = \text{Var}(I(\mathbf{Z}; \mathbf{Y}))$ and $\Delta_{Z|Y} = \text{Var}(I(\mathbf{X}, \mathbf{A}; \mathbf{Z} \mid \mathbf{Y}))$ represent parameter sensitivity. The critical ratio:

$$\beta < \frac{\Delta_Z}{\Delta_Z + \Delta_{Z|Y}} \tag{22}$$

ensures sufficient weight on predictive information $I(\mathbf{Z}; \mathbf{Y})$. Under this condition, IB produces representations $\mathbf{Z}$ with minimised superfluous information and maximised prediction confidence than standard supervised learning. $\square$

**Derivation of** $I(\mathbf{X}, \mathbf{A}; \mathbf{Z} \mid \mathbf{Y}) = I(\mathbf{X}; \mathbf{Z} \mid \mathbf{Y}) + I(\mathbf{A}; \mathbf{Z} \mid \mathbf{Y}, \mathbf{X})$. The decomposition follows from the chain rule of mutual information:

$$\begin{aligned} I(\mathbf{X}, \mathbf{A}; \mathbf{Z} \mid \mathbf{Y}) &= I(\mathbf{X}; \mathbf{Z} \mid \mathbf{Y}) + I(\mathbf{A}; \mathbf{Z} \mid \mathbf{Y}, \mathbf{X}) \\ &= H(\mathbf{Z} \mid \mathbf{Y}) - H(\mathbf{Z} \mid \mathbf{Y}, \mathbf{X}) \\ &\quad + H(\mathbf{Z} \mid \mathbf{Y}, \mathbf{X}) - H(\mathbf{Z} \mid \mathbf{Y}, \mathbf{X}, \mathbf{A}) \\ &= H(\mathbf{Z} \mid \mathbf{Y}) - H(\mathbf{Z} \mid \mathbf{Y}, \mathbf{X}, \mathbf{A}). \end{aligned} \tag{23}$$

For deterministic encoders where $\mathbf{Z} = f(\mathbf{X}, \mathbf{A})$, we have $H(\mathbf{Z} \mid \mathbf{X}, \mathbf{A}, \mathbf{Y}) = 0$, by Eq. 16, we complete the derivation.

### B.4    PROOF OF PROPOSITION 5.3

*Proof.* Given an optmised network $f^*$ trained with ID data $(\mathbf{X}_{\text{id}}, \mathbf{A}_{\text{id}})$ via the IB objective. Let $\mathbf{Z}^* = f^*(\mathbf{X}, \mathbf{A})$ be the encoded representation obtained from the network. We can derive the following:

**Part 1: ID Data Compression** From the derivation in Eq. 23, given ID inputs $(\mathbf{X}_{\text{id}}, \mathbf{A}_{\text{id}})$ and optimal encoded ID representation $\mathbf{Z}_{\text{id}}^*$, the second term in the IB objective in Eq. 36 can be expressed as:

$$I(\mathbf{X}_{\text{id}}, \mathbf{A}_{\text{id}}; \mathbf{Z}_{\text{id}}^* \mid \mathbf{Y}) \tag{24}$$

Following directly from Lemma 5.1 and Theorem 5.2, this is equivalent to minimising $H(\mathbf{Z}_{\text{id}}^* \mid \mathbf{Y})$, making $P(\mathbf{Y} \mid \mathbf{Z}_{\text{id}}^*)$ sharply concentrated. Thus:

$$H(\mathbf{Y} \mid \mathbf{Z}_{\text{id}}^*) = -\mathbb{E}_{\mathbf{Z}_{\text{id}}^*}\left[\sum_y P(y \mid \mathbf{Z}_{\text{id}}^*) \log P(y \mid \mathbf{Z}_{\text{id}}^*)\right] \to 0. \tag{25}$$

**Part 2: OOD Data Separation** To investigate the effect on OOD data, without loss of generality, we consider the feature shift defined in Section H, where $P_{\text{id}}(\mathbf{X}) \neq P_{\text{ood}}(\mathbf{X})$, $P_{\text{id}}(\mathbf{X}|\mathbf{A}) \neq P_{\text{ood}}(\mathbf{X}|\mathbf{A})$, $P_{\text{id}}(\mathbf{Y}|\mathbf{X}) \neq P_{\text{ood}}(\mathbf{Y}|\mathbf{X})$, and $P_{\text{id}}(\mathbf{Y}|\mathbf{A}, \mathbf{X}) \neq P_{\text{ood}}(\mathbf{Y}|\mathbf{A}, \mathbf{X})$. Such that we have OOD features

$\mathbf{X}_{\text{ood}}$ with ID structure $\mathbf{A}_{\text{id}}$ (i.e., $\mathcal{G}_{\text{feat shift}} = (\mathbf{X}_{\text{ood}}, \mathbf{A}_{\text{id}})$). Let $\mathbf{Z}^*_{\text{ood}} = f^*(\mathbf{X}_{\text{ood}}, \mathbf{A}_{\text{id}})$ denote the representation encoded from the ID trained model for the given OOD data.

From the derivation in Eq. 23, we have:

$$I(\mathbf{X}_{\text{id}}, \mathbf{A}_{\text{id}}; \mathbf{Z}^*_{\text{id}} \mid \mathbf{Y}) = I(\mathbf{X}_{\text{id}}; \mathbf{Z}^*_{\text{id}} \mid \mathbf{Y}) + I(\mathbf{A}_{\text{id}}; \mathbf{Z}^*_{\text{id}} \mid \mathbf{Y}, \mathbf{X}_{\text{id}}). \tag{26}$$

$$I(\mathbf{X}_{\text{ood}}, \mathbf{A}_{\text{id}}; \mathbf{Z}^*_{\text{ood}} \mid \mathbf{Y}) = I(\mathbf{X}_{\text{ood}}; \mathbf{Z}^*_{\text{ood}} \mid \mathbf{Y}) + I(\mathbf{A}_{\text{id}}; \mathbf{Z}^*_{\text{ood}} \mid \mathbf{Y}, \mathbf{X}_{\text{ood}}). \tag{27}$$

Suppose for contradiction that:

$$I(\mathbf{X}_{\text{ood}}; \mathbf{Z}^*_{\text{ood}} \mid \mathbf{Y}) < I(\mathbf{X}_{\text{id}}; \mathbf{Z}^*_{\text{id}} \mid \mathbf{Y}). \tag{28}$$

This would imply:

$$I(\mathbf{A}_{\text{id}}; \mathbf{Z}^*_{\text{ood}} \mid \mathbf{Y}, \mathbf{X}_{\text{ood}}) \cong I(\mathbf{A}_{\text{id}}; \mathbf{Z}^*_{\text{id}} \mid \mathbf{Y}, \mathbf{X}_{\text{id}}), \tag{29}$$

$$\Rightarrow I(\mathbf{X}_{\text{ood}}, \mathbf{A}_{\text{id}}; \mathbf{Z}^*_{\text{ood}} \mid \mathbf{Y}) \leq I(\mathbf{X}_{\text{id}}, \mathbf{A}_{\text{id}}; \mathbf{Z}^*_{\text{id}} \mid \mathbf{Y}). \tag{30}$$

However, this contradicts the IB optimality condition since the encoder $\mathbf{Z}^*$ was never trained to compress $\mathbf{X}_{\text{ood}}$. Thus, we must have:

$$I(\mathbf{X}_{\text{ood}}; \mathbf{Z}^*_{\text{ood}} \mid \mathbf{Y}) \geq I(\mathbf{X}_{\text{id}}; \mathbf{Z}^*_{\text{id}} \mid \mathbf{Y}). \tag{31}$$

Similarly, since $\mathbf{A}_{\text{id}}$ is fixed but $\mathbf{X}_{\text{ood}}$ is novel:

$$I(\mathbf{A}_{\text{id}}; \mathbf{Z}^*_{\text{ood}} \mid \mathbf{Y}, \mathbf{X}_{\text{ood}}) \geq I(\mathbf{A}_{\text{id}}; \mathbf{Z}^*_{\text{id}} \mid \mathbf{Y}, \mathbf{X}_{\text{id}}). \tag{32}$$

Therefore:

$$I(\mathbf{X}_{\text{ood}}, \mathbf{A}_{\text{id}}; \mathbf{Z}^*_{\text{ood}} \mid \mathbf{Y}) \geq I(\mathbf{X}_{\text{id}}, \mathbf{A}_{\text{id}}; \mathbf{Z}^*_{\text{id}} \mid \mathbf{Y}). \tag{33}$$

From Lemma 5.1, higher conditional mutual information implies higher $H(\mathbf{Z}^*_{\text{ood}} \mid \mathbf{Y})$. By Bayes' theorem:

$$P(\mathbf{Y} \mid \mathbf{Z}^*_{\text{ood}}) = \frac{P(\mathbf{Z}^*_{\text{ood}} \mid \mathbf{Y})P(\mathbf{Y})}{P(\mathbf{Z}^*_{\text{ood}})}. \tag{34}$$

The diffuse $P(\mathbf{Z}^*_{\text{ood}} \mid \mathbf{Y})$ makes $P(\mathbf{Y} \mid \mathbf{Z}^*_{\text{ood}})$ approximately uniform, yielding:

$$H(\mathbf{Y} \mid \mathbf{Z}^*_{\text{ood}}) \approx \log C \gg H(\mathbf{Y} \mid \mathbf{Z}^*_{\text{id}}), \tag{35}$$

where $C$ is the number of classes. This entropy gap enables improved OOD detection through logit-based scoring methods (i.e., energy score (Liu et al., 2020), MaxLogit (Hendrycks et al., 2022)). $\qquad\square$

## C GRAPH NEURAL NETWORK

Graph Neural Networks (GNNs) are inherently well-suited for capturing intricate dependencies between nodes in a graph. Their effectiveness largely stems from the message-passing mechanism, which progressively aggregates information from neighbouring nodes, allowing the model to learn both local and global structural and feature patterns. Let $\mathbf{z}_i^{(l)}$ represent the learned embedding of node $i$ at layer $l$. A standard Graph Convolutional Network (GCN) updates node representations iteratively using the propagation rule:

$$\mathbf{Z}^{(l)} = \sigma\left(\mathbf{D}^{-1/2}\tilde{\mathbf{A}}\mathbf{D}^{-1/2}\mathbf{Z}^{(l-1)}\mathbf{W}^{(l)}\right),$$

where $\mathbf{Z}^{(l-1)} = [\mathbf{z}_i^{(l-1)}]$, and the initial node features are given by $\mathbf{H}^{(0)} = \mathbf{X}$. Here, $\tilde{\mathbf{A}} = \mathbf{A} + \mathbf{I}$ is the adjacency matrix with self-loops, $\mathbf{I}$ denotes the identity matrix, $\mathbf{D}$ is the diagonal degree matrix of $\tilde{\mathbf{A}}$, $\sigma$ is a nonlinear activation function (e.g., ReLU), and $\mathbf{W}^{(l)}$ represents the trainable weight matrix for layer $l$ (Kipf & Welling, 2017).

## D  INFORMATION BOTTLENECK PRINCIPLE

The information bottleneck (IB) principle aims to learn a compressed representation $\mathbf{Z}$ with the maximum relevant information about the label $\mathbf{Y}$ (i.e., $I(\mathbf{Z}; \mathbf{Y})$) while minimising the label-irrelevant information (i.e., $I(\mathbf{X}; \mathbf{Z})$), as constrained by the Markov chain $\mathbf{Y} \rightarrow \mathbf{X} \rightarrow \mathbf{Z}$ (Tishby et al., 2000; Alemi et al., 2017). For graph data, $\mathbf{Z}$ is a function of both node features $\mathbf{X}$ and graph structure $\mathbf{A}$ (i.e., $\mathbf{Z} = \text{GNN}(\mathbf{X}, \mathbf{A})$) The IB objective can be defined as:

$$\max_{\mathbf{Z}} I(\mathbf{Z}; \mathbf{Y}) - \beta I(\mathbf{X}, \mathbf{A}; \mathbf{Z}), \tag{36}$$

where $\beta$ is a Lagrange multiplier controlling the trade-off between compression and prediction. This enables the compression of the joint input information to representation $\mathbf{Z}$ via $I(\mathbf{X}, \mathbf{A}; \mathbf{Z})$ and optimise its classification ability via $I(\mathbf{Z}; \mathbf{Y})$ (Wu et al., 2020; Sun et al., 2022).

## E  TRACTABLE OPTIMISATION OF TRIBE

To optimise the intractable TRIBE objective in Eq. 13, we approximate the IB terms via variational lower bounds (Alemi et al., 2017) and enforce the conditional independence and pairwise mutual information minimisation using reconstruction loss and contrastive loss (Cheng et al., 2020), respectively. **Variational Approximation of $\mathcal{L}_{\text{IB}}$:** Beginning with the first term in objective Eq. 13, without loss of generality, we provide a variational approximation bound for $\text{IB}_{\mathbf{Z}} = I(\mathbf{Z}; \mathbf{Y}) - \beta_{\mathbf{Z}} I(\mathbf{X}, \mathbf{A}; \mathbf{Z})$, and the tractable bounds for $\text{IB}_{\mathbf{V}}$ and $\text{IB}_{\mathbf{Q}}$ can be derived accordingly. Let $q(\mathbf{Y}|\mathbf{Z})$ and $r(\mathbf{Z})$ denote variational approximations to the true conditional distribution $p_{\mathbf{Z}}(\mathbf{Y}|\mathbf{Z})$, and marginal distribution $p(\mathbf{Z})$, respectively. Consider a parametric Gaussian distribution as prior $p(\mathbf{Z}|\mathbf{X}, \mathbf{A})$, we have:

$$p(\mathbf{Z}|\mathbf{X}, \mathbf{A}) = \mathcal{N}(\mathbf{Z}; \mu(f(\mathbf{X}, \mathbf{A})), \Sigma(f(\mathbf{X}, \mathbf{A}))),$$

where $f$ is modelled as a GNN network that encodes the input $(\mathbf{X}, \mathbf{A})$, followed by linear layers to obtain the mean and variance representations respectively. Subsequently, we apply the reparameterisation trick as $\mathbf{Z} = f(\mathbf{X}, \mathbf{A}, \epsilon)$, which ensures it is a deterministic function of $\mathbf{X}, \mathbf{A}$ and the Gaussian random variable $\epsilon \sim p(\epsilon) = \mathcal{N}(\mathbf{0}, \mathbf{1})$. Thus, given training data $\mathbf{X}_{\text{tr}} = (\mathbf{x_1}, \dots, \mathbf{x_N}), \mathbf{Y}_{\text{tr}} = (y_1, \dots, y_N)$ with adjacency matrix $\mathbf{A}_{\text{tr}}$, using the empirical data distribution, we can obtain a tractable variational lower bound for $\text{IB}_{\mathbf{Z}}$:

$$\text{VIB}_{\mathbf{Z}} = \frac{1}{N} \sum_{n=1}^{N} \left[ \mathbb{E}_{\epsilon \sim p(\epsilon)} \left[ \log q(y_n | f(\mathbf{x_n}, \mathbf{A}, \epsilon)) \right] - \beta \text{KL} \left( p(\mathbf{Z}|\mathbf{x_n}, \mathbf{A}) || r(\mathbf{Z}) \right) \right] \leq \text{IB}_{\mathbf{Z}}, \tag{37}$$

where $r(\mathbf{Z}) = \mathcal{N}(\mathbf{Z}, \mathbf{0}, \mathbf{1})$ is fixed to the standard normal. The first term in Eq. 37 represents the log-likelihood of the output $\mathbf{Y}$ given the representation $\mathbf{Z}$, which can be calculated using the cross entropy loss to encourage $\mathbf{Z}$ to be predictive of $\mathbf{Y}$. The second term is the Kullback-Leibler (KL) divergence between the conditional distribution $p(\mathbf{Z} \mid \mathbf{X}, \mathbf{A})$ and the variational prior $r(\mathbf{Z})$, which reduces the irrelevant information compressed from the joint input $(\mathbf{X}, \mathbf{A})$ into the representation $\mathbf{Z}$. Following a similar approach, we can derive the lower bound for $\text{IB}_{\mathbf{V}}$ and $\text{IB}_{\mathbf{Q}}$, and thus obtain the tractable variational lower bound $\mathcal{L}_{\text{VIB}_{\mathbf{Z}; \mathbf{V}; \mathbf{Q}}}$ for the first term in Eq. 13 as:

$$\max \mathcal{L}_{\text{IB}_{\mathbf{Z}; \mathbf{V}; \mathbf{Q}}} \geq \mathcal{L}_{\text{VIB}_{\mathbf{Z}; \mathbf{V}; \mathbf{Q}}} = \text{VIB}_{\mathbf{Z}} + \text{VIB}_{\mathbf{V}} + \text{VIB}_{\mathbf{Q}}. \tag{38}$$

**Conditional Independence of $\mathcal{L}_{\text{CInd}}$:** To make $\mathcal{L}_{\text{CInd}} = I(\mathbf{A}; \mathbf{X}|\mathbf{Z})$ tractable, we can use a variational approximation $q(\mathbf{X}|\mathbf{Z})$ to estimate the true distribution $p(\mathbf{X}|\mathbf{Z})$ and derive a varitional upper bound via:

$$\min \mathcal{L}_{\text{VCInd}_{\mathbf{X}, \mathbf{A}, \mathbf{Z}}} = \mathbb{E}_{p(\mathbf{x}, \mathbf{A}, \mathbf{z})} [\underbrace{\log p(\mathbf{x}|\mathbf{A}, \mathbf{z})}_{(1)} - \underbrace{\log q(\mathbf{x}|\mathbf{z})}_{(2)}] \geq \mathcal{L}_{\text{CInd}_{\mathbf{X}, \mathbf{A}, \mathbf{Z}}}. \tag{39}$$

Minimising this objective encourages $\mathbf{Z}$ to encode all the information of $\mathbf{X}$ that is relevant to $\mathbf{A}$ (via (1)) and reducing the information of $\mathbf{X}$ that is independent of $\mathbf{A}$ (via (2)). This ensures $\mathbf{X}$ and $\mathbf{A}$ is conditionally independent given $\mathbf{Z}$.

**Pairwise Mutual Information Minimisation via Contrastive Learning $\mathcal{L}_{\text{PMI}}$:** Similar to the variational bound for the IB terms, without loss of generality, we provide an upper bound for $I(\mathbf{Z}; \mathbf{V})$,

and the bounds for $I(\mathbf{Z}; \mathbf{Q})$ and $I(\mathbf{V}; \mathbf{Q})$ can be derived accordingly. Notably, since the conditional distribution $p(\mathbf{Z}|\mathbf{V})$ is unknown, utilising the variational contrastive log-ratio upper bound of mutual information (CLUB) (Cheng et al., 2020) with samples $(\mathbf{z_n}, \mathbf{v_n})$, we can derive a tractable upper bound for $I(\mathbf{Z}; \mathbf{V})$:

$$I_{\text{VCLUB}_{\mathbf{Z};\mathbf{V}}} = \frac{1}{N} \sum_{n=1}^{N} \Big[ \log q(\mathbf{z_n}|\mathbf{v_n}) - \frac{1}{N} \sum_{m=1}^{N} \log q(\mathbf{z_m}|\mathbf{v_n}) \Big] \geq I(\mathbf{Z}; \mathbf{V}), \qquad (40)$$

for $q(\mathbf{Z}|\mathbf{V})$ is a variational distribution to approximate $p(\mathbf{Z}|\mathbf{V})$. The first term evaluates how well the variational approximation predicts the "positive" pairs $(\mathbf{z_n}, \mathbf{v_n})$, capturing the dependence between $\mathbf{Z}$ and $\mathbf{V}$. The second term penalises this by evaluating the model on "negative" pairs $(\mathbf{z_m}, \mathbf{v_n})$, approximating the behaviour when $\mathbf{Z}$ and $\mathbf{V}$ are independent, thereby providing an effective upper bound for $I(\mathbf{Z}; \mathbf{V})$. Similarly, we can derive the upper bounds for $I(\mathbf{Z}; \mathbf{Q})$ and $I(\mathbf{V}; \mathbf{Q})$, and obtain the final tractable objective $\mathcal{L}_{\text{VPMI}_{\mathbf{Z};\mathbf{V};\mathbf{Q}}}$ as:

$$\min \mathcal{L}_{\text{VPMI}_{\mathbf{Z};\mathbf{V};\mathbf{Q}}} = \alpha_1 I_{\text{VCLUB}_{\mathbf{Z};\mathbf{V}}} + \alpha_2 I_{\text{VCLUB}_{\mathbf{Z};\mathbf{Q}}} + \alpha_3 I_{\text{VCLUB}_{\mathbf{V};\mathbf{Q}}} \geq \mathcal{L}_{\text{PMI}_{\mathbf{Z};\mathbf{V};\mathbf{Q}}}. \qquad (41)$$

Hence, minimising this loss will ensure the pairwise MI independence between $\mathbf{Z}$, $\mathbf{V}$, and $\mathbf{Q}$. **Final tractable TRIBE objective:** Combining Eq. 38, 39, 41, the overall tractable objective for TRIBE is given by:

$$\max_{\theta_{\mathbf{Z}}, \theta_{\mathbf{V}}, \theta_{\mathbf{Q}}} \mathcal{L}_{\text{VIB}_{\mathbf{Z};\mathbf{V};\mathbf{Q}}} - \lambda_{\text{CInd}} \mathcal{L}_{\text{VCInd}_{\mathbf{X},\mathbf{A},\mathbf{Z}}} - \mathcal{L}_{\text{VPMI}_{\mathbf{Z},\mathbf{V},\mathbf{Q}}}. \qquad (42)$$

While the objective in Eq. 42 combines all losses into one expression, we optimise the networks individually using their respective losses.

The optimisation process for TRIBE is outlined in Algorithm 1 in the main text. In **Step 1**, the individual networks are updated to capture sufficient yet minimal joint-input, structure-only, and feature-only information for predicting the label $\mathbf{Y}$. **Step 2** enforces mutual conditional independence between $\mathbf{X}|\mathbf{Z}$ and $\mathbf{A}$ specifically for the GNN classifier. Finally, **Step 3** minimises pairwise mutual information, updating each network only with the loss relevant to its representation. This structured approach ensures efficient and targeted optimisation for TRIBE.

# F    NEURAL NETWORK PARAMETERISATION OF TRIBE

To optimise the intractable TRIBE objective in Eq. 42, we parameterise the variational approximations using GNNs and MLPs: **Main Encoder Networks.** Our framework uses three primary networks to encode the representations $\mathbf{Z}$, $\mathbf{V}$ and $\mathbf{Q}$ and for prediction (i.e., joint-input classifier, feature network, structure network) with parameters $\theta_{\mathbf{Z}}$, $\theta_{\mathbf{V}}$, and $\theta_{\mathbf{Q}}$ respectively:

$$\text{Joint-input Network:} \quad \text{GNN}_{\text{CLS}}(\mathbf{X}, \mathbf{A}) \leftarrow \mathbf{Z} \qquad (43)$$

$$\text{Feature Network:} \quad \text{MLP}_{\text{feat}}(\mathbf{X}) \leftarrow \mathbf{V} \qquad (44)$$

$$\text{Structure Network:} \quad \text{GNN}_{\text{struct}}(\mathbf{I}, \mathbf{A}) \leftarrow \mathbf{Q}, \qquad (45)$$

where $\mathbf{X} \in \mathbb{R}^{n \times d}$ is the node feature matrix, $\mathbf{A} \in \mathbb{R}^{n \times n}$ is the adjacency matrix, $\mathbf{I} \in \mathbb{R}^{n \times n}$ is the identity matrix (used as placeholder features), and $\mathbf{Z}, \mathbf{V}, \mathbf{Q}$ are the latent representations. To capture structural information, we leverage a GNN model that inherently provides robust structure encoding. Meanwhile, for the feature network - comprising solely feature embeddings - a lightweight MLP is more appropriate.

Each network contains the following elements respectively: **Variational Implementation.** Following common variational approximations (Alemi et al., 2017), we model the conditional distributions of our latent representations using Gaussian distributions parameterised by the respective encoders. We use a GNN/MLP encoder to extract representations $\boldsymbol{h}$ from the given inputs, and use linear MLP layers to encode the latent variables $\boldsymbol{\mu}$ and $\boldsymbol{\sigma}$, this produces Gaussian distributions over the latent space:

$$\boldsymbol{\mu}_Z = \text{MLP}_{\mu_Z}(\boldsymbol{h}_Z), \quad \boldsymbol{\sigma}_Z = \text{MLP}_{\sigma_Z}(\boldsymbol{h}_Z), \quad \boldsymbol{h}_Z = \text{GNN}_{\text{enc}}(\mathbf{X}, \mathbf{A}) \qquad (46)$$

$$\boldsymbol{\mu}_V = \text{MLP}_{\mu_V}(\boldsymbol{h}_V), \quad \boldsymbol{\sigma}_V = \text{MLP}_{\sigma_V}(\boldsymbol{h}_V), \quad \boldsymbol{h}_V = \text{MLP}_{\text{enc}}(\mathbf{X}) \qquad (47)$$

$$\boldsymbol{\mu}_Q = \text{MLP}_{\mu_Q}(\boldsymbol{h}_Q), \quad \boldsymbol{\sigma}_Q = \text{MLP}_{\sigma_Q}(\boldsymbol{h}_Q), \quad \boldsymbol{h}_Q = \text{GNN}_{\text{enc}}(\mathbf{I}, \mathbf{A}). \qquad (48)$$

Using these, we can sample the latent variables using the reparameterisation trick:

$$\mathbf{Z} = \boldsymbol{\mu}_Z + \boldsymbol{\sigma}_Z \odot \boldsymbol{\epsilon}_Z, \quad \boldsymbol{\epsilon}_Z \sim \mathcal{N}(0, \mathbf{I}) \tag{49}$$

$$\mathbf{V} = \boldsymbol{\mu}_V + \boldsymbol{\sigma}_V \odot \boldsymbol{\epsilon}_V, \quad \boldsymbol{\epsilon}_V \sim \mathcal{N}(0, \mathbf{I}) \tag{50}$$

$$\mathbf{Q} = \boldsymbol{\mu}_Q + \boldsymbol{\sigma}_Q \odot \boldsymbol{\epsilon}_Q, \quad \boldsymbol{\epsilon}_Q \sim \mathcal{N}(0, \mathbf{I}) \tag{51}$$

This enable us to obtain the representations $\mathbf{Z}, \mathbf{V}$ and $\mathbf{Q}$ for optimisation.

**Auxiliary Layers.** For predictions and calculation of the reconstruction loss, IB minimisation, and pairwise MI minimisation, we use the following auxiliary layers in the respective networks:

$$\text{Prediction Layers:} \quad \text{GNN}_{\text{pred-Z}}(\mathbf{Z}, \mathbf{A}) \rightarrow \hat{\mathbf{Y}}_Z \tag{52}$$

$$\text{MLP}_{\text{pred-V}}(\mathbf{V}) \rightarrow \hat{\mathbf{Y}}_V \tag{53}$$

$$\text{GNN}_{\text{pred-Q}}(\mathbf{Q}, \mathbf{A}) \rightarrow \hat{\mathbf{Y}}_Q \tag{54}$$

$$\text{Reconstruction Layer:} \quad \text{MLP}_{\text{recon}}(\mathbf{Z}) \rightarrow \hat{\mathbf{X}} \tag{55}$$

### F.1 Objective Components in Terms of Networks

**1) Information Bottleneck Terms.** For each representation $\mathbf{Z}, \mathbf{V}$, and $\mathbf{Q}$, we calculate the VIB loss Eq. 37. For example, for the joint representation $\mathbf{Z}$:

$$\text{VIB}_{\mathbf{Z}} = \underbrace{\mathbb{E}_{\boldsymbol{\epsilon}_Z}[\log \text{MLP}_{\text{pred-Z}}(\mathbf{Z} = \boldsymbol{\mu}_Z + \sigma_Z \odot \boldsymbol{\epsilon}_Z)(\mathbf{Y})]}_{\text{CE(Z,Y)}}$$

$$- \beta_Z \underbrace{\sum_i \left(1 + \log((\boldsymbol{\sigma}_{Z,i})^2) - (\boldsymbol{\mu}_{Z,i})^2 - (\boldsymbol{\sigma}_{Z,i})^2\right)}_{\text{KL divergence term (analytical form)}} \tag{56}$$

**2) Conditional Independence Term.** For the conditional independence loss Eq. 39, we use the reconstruction network:

$$\mathcal{L}_{\text{CInd}} = \underbrace{\mathbb{E}_{\boldsymbol{\epsilon}_Z}[-\|\mathbf{X} - \text{MLP}_{\text{recon}}(\mathbf{Z} = \boldsymbol{\mu}_Z + \boldsymbol{\sigma}_Z \odot \boldsymbol{\epsilon}_Z)\|_2^2]}_{\text{Reconstruction loss (encourages Z to encode X information)}} \tag{57}$$

**3) Pairwise Mutual Information Terms.** Following CLUB (Cheng et al., 2020), we estimate the pairwise mutual information terms using contrastive learning Eq. 41. For each pair of representations (i.e., $\mathbf{Z}, \mathbf{V}$), we compute:

$$\text{CLUB}(\mathbf{Z}, \mathbf{V}) = \frac{1}{N} \sum_{i=1}^N \log \frac{s(\mathbf{Z}_i, \mathbf{V}_i)}{\frac{1}{N} \sum_{j=1}^N s(\mathbf{Z}_i, \mathbf{V}_j)} \tag{58}$$

$$\tag{59}$$

where $s$ is a similarity function (e.g., dot product) between representations, N is the number of samples.

**Implementation Notes** In practice, we optimise this objective by:

- Forward pass through all encoders to get $\boldsymbol{\mu}_Z, \boldsymbol{\sigma}_Z, \boldsymbol{\mu}_V, \boldsymbol{\sigma}_V, \boldsymbol{\mu}_Q, \boldsymbol{\sigma}_Q$.
- Sample $\mathbf{Z}, \mathbf{V}, \mathbf{Q}$ using the reparameterisation trick.
- Compute all components of the loss using these samples.
- Backpropagate through the networks to update parameters according to Algorithm 1.

## G Extended Related Work

Out-of-distribution detection is a critical task in machine learning, extensively studied across various domains. A significant body of work focuses on methods that rely solely on ID data, employing

techniques such as softmax scores (Hendrycks & Gimpel, 2017; Liang et al., 2018), energy-based scoring (Liu et al., 2020; Wang et al., 2021; Yang et al., 2024b), and activation pruning (Djurisic et al., 2023; Sun & Li, 2022; Sun et al., 2021). Other strategies enhance model confidence (Hsu et al., 2020; Hein et al., 2019; Vyas et al., 2018), improve feature learning (Lin et al., 2021; Dong et al., 2022), or incorporate adversarial approaches (Bitterwolf et al., 2020; Chen et al., 2021; Choi & Chung, 2020). Beyond ID-based methods, OOD exposure leverages auxiliary OOD data during training to improve detection performance (Hendrycks et al., 2019; Liu et al., 2020; Park et al., 2023; Zhu et al., 2023; Zheng et al., 2023; Du et al., 2024; Wu et al., 2023b). Meanwhile, an emerging direction involves generating synthetic OOD data. GAN-based methods, such as ConfOOD (Lee et al., 2018a), train confidence classifiers alongside OOD data generation, while VOS (Du et al., 2022) synthesises outliers from low-probability Gaussian regions. More recently, diffusion models have been widely adopted, as seen in DFDD (Wu et al., 2023a) and Dream-OOD (Du et al., 2023).

### G.1 GRAPH OOD DETECTION

Recent advancements in OOD detection have expanded to graph-structured data, addressing both node-level and graph-level detection tasks. For node-level OOD detection, several innovative methods have been developed to improve detection accuracy and robustness. GNNSafe introduces an energy propagation schema that considers the inter-dependence of node instances, providing a more nuanced approach to identifying OOD nodes (Wu et al., 2023b). Building on this, NODESafe incorporates additional regularization terms to reduce and bound extreme energy scores, ensuring more stable and reliable detection (Yang et al., 2024b). Meanwhile, TopoOOD explores topological shifts in graph data and proposes a node-wise Dirichlet Energy metric to measure neighbourhood turbulence, which serves as a confidence score for OOD detection (Bao et al., 2024). Other approaches, such as GKDE, employ a multi-source uncertainty framework to estimate node-level Dirichlet distributions, which aids in identifying OOD instances (Zhao et al., 2020). Similarly, GPN leverages Bayesian posterior and density estimation to quantify uncertainty at the node level, further enhancing detection capabilities (Stadler et al., 2021). Moreover, DeGEM presents a novel energy-based model training framework involving a multi-hop graph encoder and energy head, targeting the detection on heterophilic graphs (Chen et al., 2025). Additionally, GOLD presents a data-synthesis-based framework that generates pseudo-OOD embeddings without relying on pre-trained generative models or auxiliary OOD datasets. At its core is an alternating optimisation framework, which effectively balances ID representation learning with divergence-enhanced pseudo-OOD generation (Wang et al., 2025a). More recently, some works explores OOD detection on text-attributed graph using Large Language Models for zero-shot detection or generating auxiliary pseudo-OOD samples Xu et al. (2025b;a); Wang et al. (2024).

At the graph level, OOD detection methods have focused on modelling distribution shifts, adopting data-centric perspectives, and utilising unsupervised learning techniques. GraphDE models distribution shifts through a graph generative process, deriving a posterior distribution to detect OOD graphs (Li et al., 2022b). In contrast, AAGOD takes a data-centric approach by learning structural patterns in graph data through a learnable amplifier matrix, improving detection performance (Guo et al., 2023). Another notable method, GOOD-D, applies unsupervised contrastive learning to enhance graph-level OOD detection without relying on labelled OOD data (Liu et al., 2023). These methods highlight the growing emphasis on leveraging graph structure and distributional properties to improve OOD detection.

### G.2 INFORMATION BOTTLENECK

The IB principle has been widely applied across various domains to enhance learned representations (Wu et al., 2020; Alemi et al., 2017; Tishby et al., 2000; Ahuja et al., 2021; Kawaguchi et al., 2023; Wang et al., 2023; Federici et al., 2020; Dai et al., 2023; Sun et al., 2022). Its objective is to learn a compressed representation that maximally retains label-relevant information while minimising label-irrelevant information from inputs (Tishby et al., 2000; Alemi et al., 2017). This concept has also been extended to graph-structured data, enabling robust representation learning from both node features and graph structure (Wu et al., 2020; Sun et al., 2022).

Notably, IB's potential for OOD detection has primarily been explored in Euclidean settings (Hu et al., 2024; Zhao & Cao, 2023; Li et al., 2023b; Sinha et al., 2021; Wu & Deng, 2024). For instance,

DRL introduces a dual representation learning framework that learns both a target-discriminative representation and an additional distribution-discriminative representation $\mathbf{C}$, capturing all information relevant to the target $\mathbf{Y}$ (Zhao & Cao, 2023). Meanwhile, Hu et al. (2024) examines information from the perspective of classification-relevant and classification-irrelevant detection, providing a theoretical analysis of the overconfidence of OOD samples in models trained on ID data using supervised learning. Alemi et al. (2018) empirically validates IB's effectiveness for OOD detection. In contrast to these studies, our work offers theoretical insights into the advantages of IB over standard supervised learning, particularly in improving prediction confidence. For graph-structured data, IS-GIB introduces I-GIB to mitigate irrelevant information by minimising the mutual information between the input graph and its embeddings. S-GIB was further leveraged to utilise structural relationships to discard irrelevant information, establishing an effective invariant learning framework for OOD generalisation (Yang et al., 2024a). Additionally, CSIB generates and refines causal subgraphs based on invariant causal prediction and the graph information bottleneck principle, preserving essential features while filtering out spurious correlations, thereby improving graph-based OOD generalisation (An et al., 2024). Moreover, IBPL introduces a graph-level OOD detection method that effectively tackles the issue of overlapping features between ID and OOD graphs, aiming to enhance detection performance. IBPL proposes a novel graph prompt that jointly optimises node features and graph structure, enabling the generation of more discriminative ID features. By leveraging the IB principle, IBPL maximises the MI between category labels and the prompt graph while minimising the MI between perturbed graphs and the prompt graph. This dual optimisation process allows for the extraction of robust ID features while significantly reducing the influence of overlapping features (Cao et al., 2025). Unlike these approaches, our work is driven by theoretical insights into IB and mutual decomposition, demonstrating their benefits for OOD detection. We validate our findings through extensive experiments, highlighting and validating the effectiveness of IB for graph OOD detection.

## H    DEFINING OOD SHIFTS

Typically, considering a message passing neural network (i.e., GNN) model, it is trained using ID data (i.e., $\mathbf{X}_{\text{ID}}^{tr}, \mathbf{A}_{\text{ID}}^{tr}$), and the test data consists of a combination of ID and OOD instances (i.e., $(\mathbf{X}_{\text{ID}}^{te}, \mathbf{A}_{\text{ID}}^{te}), (\mathbf{X}_{\text{OOD}}^{te}, \mathbf{A}_{\text{OOD}}^{te})$). The primary objective is to identify and detect the OOD samples from the ID instances while maintaining high ID classification performance.

Generally, we categorise OOD shifts into the following: $P_{train}(\mathbf{X}, \mathbf{A}) \neq P_{test}(\mathbf{X}, \mathbf{A})$ and the conditional distribution $P_{train}(\mathbf{Y}|\mathbf{X}, \mathbf{A}) \neq P_{test}(\mathbf{Y}|\mathbf{X}, \mathbf{A})$. To detect OOD data, the task is to formulate an OOD scoring function $F$, usually built upon the output from the classifier GNN, such that it outputs $F(\mathbf{X}, \mathbf{A}; \text{GNN}) = 0$ for data from in-distribution and $F(\mathbf{X}, \mathbf{A}; \text{GNN}) = 1$ for data from out-of-distribution. This definition, however, lacks the detail of the particular shift occurring alone on the feature $\mathbf{X}$ or among the structure $\mathbf{A}$. Thus, to understand when OOD detection is possible for such an ID-trained model, we adopt the following definitions and explore the impact of each of the distribution shifts (Han et al., 2024). **Feature Shift (X):** A feature shift occurs when the distribution of $\mathbf{X}$ and its relationship with $\mathbf{A}$ change between training and testing. Specifically, $P_{\text{train}}(\mathbf{X}) \neq P_{\text{test}}(\mathbf{X})$ and $P_{\text{train}}(\mathbf{X}|\mathbf{A}) \neq P_{\text{test}}(\mathbf{X}|\mathbf{A})$ or $P_{\text{train}}(\mathbf{A}|\mathbf{X}) \neq P_{\text{test}}(\mathbf{A}|\mathbf{X})$. Additionally, the conditional distribution of $\mathbf{Y}$ given $\mathbf{X}$ changes, such that $P_{\text{train}}(\mathbf{Y}|\mathbf{X}) \neq P_{\text{test}}(\mathbf{Y}|\mathbf{X})$. Consequently, the joint conditional distribution of $\mathbf{Y}$ given both $\mathbf{X}$ and $\mathbf{A}$ also shifts: $P_{\text{train}}(\mathbf{Y}|\mathbf{A}, \mathbf{X}) \neq P_{\text{test}}(\mathbf{Y}|\mathbf{A}, \mathbf{X})$. However, the conditional distribution of $\mathbf{Y}$ given $\mathbf{A}$ alone remains unchanged: $P_{\text{train}}(\mathbf{Y}|\mathbf{A}) = P_{\text{test}}(\mathbf{Y}|\mathbf{A})$. **Structural Shift (A):** A structural shift occurs when the distribution of $\mathbf{A}$ and dependencies with $\mathbf{X}$ shifts (i.e., $P_{\text{train}}(\mathbf{A}) \neq P_{\text{test}}(\mathbf{A})$ and $P_{\text{train}}(\mathbf{X}|\mathbf{A}) \neq P_{\text{test}}(\mathbf{X}|\mathbf{A})$ or $P_{\text{train}}(\mathbf{A}|\mathbf{X}) \neq P_{\text{test}}(\mathbf{A}|\mathbf{X})$). Additionally, the conditional distribution of $\mathbf{Y}$ given $\mathbf{A}$ changes, such that $P_{\text{train}}(\mathbf{Y}|\mathbf{A}) \neq P_{\text{test}}(\mathbf{Y}|\mathbf{A})$. Consequently, the joint conditional distribution of $\mathbf{Y}$ given both $\mathbf{X}$ and $\mathbf{A}$ also shifts: $P_{\text{train}}(\mathbf{Y}|\mathbf{A}, \mathbf{X}) \neq P_{\text{test}}(\mathbf{Y}|\mathbf{A}, \mathbf{X})$. However, the conditional distribution of $\mathbf{Y}$ given $\mathbf{X}$ remains invariant: $P_{\text{train}}(\mathbf{Y}|\mathbf{X}) = P_{\text{test}}(\mathbf{Y}|\mathbf{X})$. **Joint Shift (X, A):** More realistically, distributional shifts typically occur jointly on $\mathbf{X}, \mathbf{A}$, thus, we consider a joint distribution shift on both the feature and structure (i.e., $P_{\text{train}}(\mathbf{A}) \neq P_{\text{test}}(\mathbf{A})$, $P_{\text{train}}(\mathbf{X}) \neq P_{\text{test}}(\mathbf{X})$, and $P_{\text{train}}(\mathbf{X}|\mathbf{A}) \neq P_{\text{test}}(\mathbf{X}|\mathbf{A})$ or $P_{\text{train}}(\mathbf{A}|\mathbf{X}) \neq P_{\text{test}}(\mathbf{A}|\mathbf{X})$). The joint conditional distribution of $\mathbf{Y}$ given both $\mathbf{X}$ and $\mathbf{A}$ also shifts: $P_{\text{train}}(\mathbf{Y}|\mathbf{A}, \mathbf{X}) \neq P_{\text{test}}(\mathbf{Y}|\mathbf{A}, \mathbf{X})$. **Semantic Shift (Y):** Consider $\mathcal{Y}_{train}$ and $\mathcal{Y}_{test}$ as the label space of the train and test data respectively. OOD data with semantic shift is defined to consist of unknown labels $\mathbf{Y}$ that do not belong to any of the classes seen in the ID label space $\mathcal{Y}_{train}$, i.e., $\mathcal{Y}_{train} \subset \mathcal{Y}_{test}$ or $\mathcal{Y}_{train} \cap \mathcal{Y}_{test} = \emptyset$.

## I EVALUATION METRICS

To evaluate the performance of OOD detection, we followed common practices (Wu et al., 2023b; Liu et al., 2020; 2023; Yang et al., 2024b) and utilised three key metrics:

1. Area Under the Receiver Operating Characteristic curve (AUROC),

2. Area Under the Precision-Recall curve (AUPR),

3. False positive rate at 95% true positive rate (FPR95).

These metrics are independent of the threshold, avoiding the need to select $\tau$. AUROC captures the balance between the true positive rate (TPR) and false positive rate (FPR) across varying thresholds, offering an overall assessment of the model's ability to differentiate between ID and OOD samples. However, when OOD instances are rare in highly imbalanced datasets, AUROC can yield overly optimistic results. In contrast, AUPR accounts for both precision and recall, making it more suited for such imbalanced scenarios. Meanwhile, FPR95 emphasises performance under high-sensitivity conditions by measuring the rate at which ID samples are misclassified as OOD when the true positive rate is fixed at 95%. This metric highlights improvements in detection performance under stricter criteria, where a lower FPR95 indicates a more significant gain in the model's ability to differentiate between ID and OOD instances.

## J IMPLEMENTATION DETAILS

We follow (Wu et al., 2023b; Yang et al., 2024b) and use the publicly available benchmark datasets. The datasets were downloaded via Pytorch Geometric 2.0.3 and OGB 1.3.3 under the MIT license. Experiments were conducted using Python 3.9.19 and PyTorch 2.3.1 with Cuda 12.2 on a single NVIDIA RTX A6000 GPU with 48GB of memory. We follow (Wu et al., 2023b; Yang et al., 2024b) for the selection of baselines. We conducted experiments across three seeds for GNNSAFE and NODESAFE. Result inconsistencies with original reported statistics may stem from software environment, dataset differences, and unavailable hyperparameters. We reproduced results to the best of our ability, while results for other baselines were sourced from (Wu et al., 2023b; Yang et al., 2024b). The trade-off parameters $\beta$ for the individual IB losses are searched through the range $\beta \in \{0.0001, 0.001, 0.01, 0.1, 1\}$, with a default value of $\beta = 0.001$. A constant $\alpha_i \in \{0.0001, 0.001, 0.01, 0.1, 1\}$ is applied uniformly to all three mutual information terms in Eq. 11. The loss coefficient $\lambda_{\text{CInd}}$ for $\mathcal{L}_{\text{VCInd}}$ was tuned based on the dataset with values taking ranges of $\lambda_{\text{CInd}} \in \{0.0001, \ldots, 1\}$. Following (Wu et al., 2023b), the number of energy propagation iterations $k$ is set to 2, with the controlling parameter $\alpha$ fixed at 0.5, the OOD-exposure loss coefficient $\lambda_{\text{OE}}$ for $\mathcal{L}_{\text{EReg}}$ is set to 1. For OOD exposure experiments, we tuned the margins $t_{\text{ID}}$ and $t_{\text{OOD}}$ from the proposed ranges by (Wu et al., 2023b; Yang et al., 2024b), if not available, we tune with values from the range of $\{-9, \ldots, 0\}$ for $t_{\text{ID}} < t_{\text{OOD}}$ for different datasets. The Adam optimiser is used for training (Kingma & Ba, 2015). For simplicity, constant $\beta$ and $\alpha$ values were applied uniformly: $\alpha$ to all mutual information terms in PMI loss, and $\beta$ to IB terms for the GNN classifier, structure, and feature networks. Results show performance is sensitive to these weights. A suitable $\beta$ is crucial for balancing representation robustness with prediction ability, while an appropriate $\alpha$ encourages a compact joint representation, mitigating spurious impacts from structure and features. Tables 6, 7, and 8 presents the hyperparameter sensitivity analysis.

Table 6: Hyperparameter analysis for $\lambda_{\text{CInd}}$. Bold highlights the optimal parameter.

| $\lambda_{\text{CInd}}$ | Cora - S | | | | Citeseer - F | | | |
|---|---|---|---|---|---|---|---|---|
| | AUROC | AUPR | FPR | ID Acc | AUROC | AUPR | FPR | ID Acc |
| 0 | $94.79 \pm 0.22$ | $88.41 \pm 0.49$ | $26.22 \pm 2.55$ | 78.73 | $93.25 \pm 0.87$ | $82.94 \pm 2.15$ | $26.96 \pm 4.64$ | 67.83 |
| 0.0001 | $95.01 \pm 0.24$ | $88.93 \pm 0.51$ | $24.84 \pm 0.70$ | 78.90 | $\mathbf{93.96 \pm 0.73}$ | $\mathbf{84.67 \pm 2.35}$ | $\mathbf{23.13 \pm 3.55}$ | $\mathbf{68.40}$ |
| 0.001 | $94.97 \pm 0.19$ | $88.89 \pm 0.44$ | $25.55 \pm 1.86$ | 78.80 | $93.81 \pm 0.81$ | $84.51 \pm 2.42$ | $24.58 \pm 3.31$ | 68.37 |
| 0.01 | $94.93 \pm 0.32$ | $88.80 \pm 0.69$ | $25.39 \pm 0.70$ | 78.90 | $93.79 \pm 0.83$ | $84.53 \pm 2.44$ | $24.75 \pm 4.58$ | 68.33 |
| 0.1 | $94.92 \pm 0.26$ | $88.77 \pm 0.60$ | $25.65 \pm 1.80$ | 78.90 | $93.74 \pm 0.78$ | $84.25 \pm 2.67$ | $24.99 \pm 4.20$ | 68.37 |
| 1 | $\mathbf{95.15 \pm 0.37}$ | $\mathbf{89.33 \pm 0.57}$ | $\mathbf{23.31 \pm 1.04}$ | $\mathbf{78.97}$ | $93.90 \pm 0.75$ | $84.62 \pm 2.32$ | $24.05 \pm 3.94$ | 68.30 |

Table 7: Hyperparameter analysis for $\alpha$ – weight coefficients applied uniformly to all mutual information terms in PMI. Bold highlights the optimal parameter.

| $\alpha$ | Cora - F | | | | PubMed - S | | | |
|---|---|---|---|---|---|---|---|---|
| | AUROC | AUPR | FPR | ID Acc | AUROC | AUPR | FPR | ID Acc |
| 0 | $97.72 \pm 0.10$ | $94.26 \pm 0.52$ | $9.40 \pm 1.06$ | 77.47 | $96.45 \pm 0.64$ | $76.46 \pm 2.31$ | $15.66 \pm 4.88$ | 74.77 |
| 0.0001 | $97.68 \pm 0.13$ | $89.16 \pm 9.49$ | $8.80 \pm 0.79$ | 77.77 | $96.55 \pm 0.80$ | $77.26 \pm 2.72$ | $16.21 \pm 5.18$ | 74.47 |
| 0.001 | $97.66 \pm 0.10$ | $94.51 \pm 0.40$ | $8.63 \pm 0.94$ | 78.00 | $96.54 \pm 0.80$ | $77.21 \pm 2.71$ | $16.15 \pm 4.98$ | 74.47 |
| 0.01 | $\textbf{97.88} \pm \textbf{0.24}$ | $\textbf{94.67} \pm \textbf{0.56}$ | $\textbf{8.13} \pm \textbf{1.32}$ | **78.10** | $\textbf{97.25} \pm \textbf{0.58}$ | $\textbf{79.76} \pm \textbf{2.17}$ | $\textbf{12.97} \pm \textbf{3.87}$ | **75.93** |
| 0.1 | $78.49 \pm 7.47$ | $70.99 \pm 8.49$ | $91.89 \pm 6.77$ | 76.30 | $88.13 \pm 6.57$ | $66.19 \pm 11.57$ | $69.39 \pm 30.40$ | 75.70 |
| 1 | $47.82 \pm 4.92$ | $29.19 \pm 3.29$ | $97.89 \pm 1.31$ | 24.37 | $37.96 \pm 49.44$ | $31.37 \pm 49.70$ | $78.33 \pm 33.53$ | 40.77 |

Table 8: Hyperparameter analysis for $\beta$ – prediction-to-compression trade-off weight applied uniformly to the IB terms for the GNN classifier, structure, and feature networks. Bold highlights the optimal parameter.

| $\beta$ | Amazon - L | | | | Twitch | | | |
|---|---|---|---|---|---|---|---|---|
| | AUROC | AUPR | FPR | ID Acc | AUROC | AUPR | FPR | ID Acc |
| 0 | $96.92 \pm 0.33$ | $96.38 \pm 0.70$ | $9.29 \pm 1.15$ | 96.10 | $66.33 \pm 15.32$ | $72.59 \pm 13.44$ | $81.18 \pm 19.87$ | 70.75 |
| 0.0001 | $96.85 \pm 0.13$ | $95.82 \pm 0.11$ | $8.94 \pm 1.45$ | 95.89 | $\textbf{89.72} \pm \textbf{5.42}$ | $\textbf{91.92} \pm \textbf{3.85}$ | $\textbf{46.60} \pm \textbf{26.47}$ | **68.63** |
| 0.001 | $96.83 \pm 0.39$ | $95.78 \pm 0.42$ | $8.36 \pm 3.83$ | 96.04 | $89.39 \pm 5.57$ | $91.58 \pm 3.82$ | $47.75 \pm 27.79$ | 68.63 |
| 0.01 | $\textbf{96.86} \pm \textbf{0.35}$ | $\textbf{95.79} \pm \textbf{0.39}$ | $\textbf{8.13} \pm \textbf{3.55}$ | **96.07** | $89.04 \pm 6.61$ | $91.61 \pm 4.75$ | $48.89 \pm 33.03$ | 68.61 |
| 0.1 | $96.87 \pm 0.31$ | $95.80 \pm 0.38$ | $8.39 \pm 3.17$ | 94.87 | $82.14 \pm 13.18$ | $85.87 \pm 9.48$ | $51.07 \pm 33.40$ | 67.52 |
| 1 | $93.40 \pm 1.77$ | $90.85 \pm 1.20$ | $34.01 \pm 30.72$ | 82.32 | $57.70 \pm 24.03$ | $68.52 \pm 14.31$ | $68.30 \pm 25.88$ | 59.31 |

# K  DATASET DETAILS

Following the protocol outlined in (Wu et al., 2023b), we use publicly available graph benchmark datasets, sourced from the PyTorch Geometric (PyG) and the Open Graph Benchmark (OGB) [1] package[2] (Sen et al., 2008). We adhere to the provided splits and dataset generation process described in Wu et al. (2023b). **Cora** This dataset represents a citation network where nodes correspond to academic papers, and edges denote citation relationships (Sen et al., 2008). Each paper is classified into one of seven categories. Cora lacks explicit domain-based partitions for OOD evaluation, thus, we follow (Wu et al., 2023b) and generate OOD data synthetically as described in Section 6 (i.e., Structure shift, Feature shift, and label-leave-out).

Table 9: Cora dataset statistics

| | Structure (ID) | Structure (OOD) | Feature (ID) | Feature (OOD) | Label (ID) | Label (OOD) |
|---|---|---|---|---|---|---|
| Nodes | 2708 | 2708 | 2708 | 2708 | 904 | 986 |
| Edges | 10556 | 6696 | 10556 | 10556 | 10556 | 10556 |
| Feature Dim | 1433 | 1433 | 1433 | 1433 | 1433 | 1433 |
| Classes | 7 | 7 | 7 | 7 | 3 | 3 |

**Citeseer** This dataset is another citation network (Giles et al., 1998; Sen et al., 2008), where nodes represent scientific papers classified into one of six classes, and edges denote citation relationships. It contains slightly more papers with fewer edges than Cora, however, the feature dimension is larger. We follow Cora and generate three OOD data synthetically as described in Section 6 .

Table 10: Citeseer dataset statistics

| | Structure (ID) | Structure (OOD) | Feature (ID) | Feature (OOD) | Label (ID) | Label (OOD) |
|---|---|---|---|---|---|---|
| Nodes | 3327 | 3327 | 3327 | 3327 | 1104 | 1522 |
| Edges | 9104 | 5932 | 9104 | 9104 | 9104 | 9104 |
| Feature Dim | 3703 | 3703 | 3703 | 3703 | 3703 | 3703 |
| Classes | 6 | 6 | 6 | 6 | 2 | 3 |

**Pubmed** This dataset is a biomedical paper citation network (Sen et al., 2008), with each paper classified into one of three classes. The nodes represent academic papers, while edges are citation

---

[1] https://github.com/snap-stanford/ogb?tab=readme-ov-file
[2] https://pytorch-geometric.readthedocs.io/en/latest/modules/datasets.html

relationships, we follow (Yang et al., 2024b) and generate OOD data synthetically as described in Section 6. Due to the `Pubmed` dataset having only three classes, the NODESAFE approach designates two classes as OOD labels (i.e., OOD training and OOD testing), leaving just one class in the training set. This setup may pose challenges, as training with only one class can lead to significant imbalance and limitations in model evaluation. As a result, we exclude the label shift scenario for `Pubmed` from our analysis.

Table 11: `Pubmed` dataset statistics

|  | Structure (ID) | Structure (OOD) | Feature (ID) | Feature (OOD) |
|---|---|---|---|---|
| Nodes | 19717 | 19717 | 19717 | 19717 |
| Edges | 88648 | 74188 | 88648 | 88648 |
| Feature Dim | 500 | 500 | 500 | 500 |
| Classes | 3 | 3 | 3 | 3 |

**Amazon-Photo** This dataset models an item co-purchasing network, where nodes represent products, and edges indicate frequently co-purchased items (McAuley et al., 2015). Node features capture product descriptions, and labels correspond to product categories. Similar to `Cora`, we generate OOD data synthetically. **Coauthor-CS** This dataset represents a collaboration network, where

Table 12: `Amazon-Photo` dataset statistics

|  | Structure (ID) | Structure (OOD) | Feature (ID) | Feature (OOD) | Label (ID) | Label (OOD) |
|---|---|---|---|---|---|---|
| Nodes | 7650 | 7650 | 7650 | 7650 | 3095 | 3673 |
| Edges | 238162 | 149168 | 238162 | 238162 | 238162 | 238162 |
| Feature Dim | 745 | 745 | 745 | 745 | 745 | 745 |
| Classes | 8 | 8 | 8 | 8 | 3 | 4 |

nodes correspond to authors, and edges indicate co-authorships in computer science research (Sinha et al., 2015). The task involves classifying authors into their respective fields based on publication keywords. OOD graphs are generated following the synthetic protocol used for other datasets.

Table 13: `Coauthor-CS` dataset statistics

|  | Structure (ID) | Structure (OOD) | Feature (ID) | Feature (OOD) | Label (ID) | Label (OOD) |
|---|---|---|---|---|---|---|
| Nodes | 18333 | 18333 | 18333 | 18333 | 13290 | 3649 |
| Edges | 163788 | 92802 | 163788 | 163788 | 163788 | 163788 |
| Feature Dim | 6805 | 6805 | 6805 | 6805 | 6805 | 6805 |
| Classes | 15 | 15 | 15 | 15 | 10 | 4 |

**TwitchGamers - Explicit** This dataset comprises multiple social network subgraphs from different geographic regions (Rozemberczki & Sarkar, 2021). Nodes represent Twitch gamers, and edges indicate follower relationships. Node features include game-based embeddings, and the classification task focuses on predicting whether a user streams mature content. We use the DE subgraph as ID data and the ES, FR, and RU subgraphs as OOD test data. **OGBN-Arxiv** This

Table 14: `TwitchGamers - Explicit` dataset statistics

|  | DE (ID) | ES (OOD) | FR (OOD) | RU (OOD) |
|---|---|---|---|---|
| Nodes | 9498 | 4648 | 6551 | 4385 |
| Edges | 315774 | 123412 | 231883 | 78993 |
| Feature Dim | 128 | 128 | 128 | 128 |
| Classes | 2 | 2 | 2 | 2 |

dataset is a large-scale citation network spanning research papers from 1960 to 2020 (Hu et al., 2020). Nodes represent papers, categorised by subject area, and edges signify citation links. Node features are derived from word embeddings of paper titles and abstracts. Following (Wu et al., 2023b), we partition the dataset using publication timestamps-papers published before 2015 are used as ID data, while papers published after 2017 serve as OOD data.

Table 15: `OGBN-Arxiv` dataset statistics

|  | 2015 (ID) | 2018 (OOD) | 2019 (OOD) | 2020 (OOD) |
|---|---|---|---|---|
| Nodes | 53160 | 29799 | 39711 | 8892 |
| Edges | 152226 | 622466 | 1061197 | 1166243 |
| Feature Dim | 128 | 128 | 128 | 128 |
| Classes | 40 | 40 | 40 | 40 |

# L    EXTENDED EXPERIMENTAL RESULTS

In this section, we provide the extended results from the main paper. Table 16. Additionally, we provide the OOD detection performance for each subset of the OOD datasets in Tables 17 to 23, complementing Tables 1 and 3 in the main text. The scores reported in the subset tables are averaged across three runs, with variance reflecting performance deviation across the three seeds. Furthermore, we report an extended version of the ablation study in Tables 24 to 28, supplementing Table 5 in the main text.

Table 16: Overall Model performance comparison: out-of-distribution detection is measured by **AUROC** (↑) / **AUPR** (↑) / **FPR95** (↓) (%) and in-distribution classification results are measured by accuracy (**ID ACC**) (↑). OOD detection performance was prioritised, with the detection results of our TRIBE against Non- (Real-) OOD Exposure methods.

Table 17: `Cora:` Extended OOD detection performance with three types of OOD (**S**tructure manipulation, **F**eature interpolation, and **L**abel-leave-out). GS++ is short for GNNSAFE++ and NS++ is short for NODESAFE++. OE indicates OOD exposure.

Table 18: **Citeseer** Extended OOD detection performance with three types of OOD (**S**tructure manipulation, **F**eature interpolation, and **L**abel-leave-out). GS++ is short for GNNSAFE++ and NS++ is short for NODESAFE++. OE indicates OOD exposure.

| Dataset | Metrics | Non-OOD Exposure | | | | | | | | Real OOD Exposure | | | | TRIBE | |
|---|---|---|---|---|---|---|---|---|---|---|---|---|---|---|---|
| | | MSP | ODIN | Mahalanobis | Energy | GKDE | GPN | GNNSAFE | NODESAFE | OE | Energy FT | GS++ | NS++ | w/o OE | w/ OE |
| Citeseer-S | AUROC | 66.34 | 49.23 | 45.26 | 65.62 | 61.48 | 70.55 | 79.87 ± 0.56 | 87.70 ± 2.27 | 58.74 | 68.87 | 81.30 ± 0.43 | 77.32 ± 8.74 | 91.89 ± 0.36 | 91.90 ± 0.12 |
| | AUPR | 34.78 | 23.07 | 21.20 | 33.63 | 31.55 | 41.12 | 61.44 ± 1.50 | 76.81 ± 1.13 | 30.07 | 36.01 | 64.59 ± 0.73 | 59.74 ± 6.25 | 80.11 ± 0.89 | 79.86 ± 0.96 |
| | FPR95 | 85.03 | 100.00 | 99.13 | 87.59 | 93.71 | 78.26 | 74.45 ± 0.59 | 65.99 ± 12.52 | 95.37 | 76.44 | 71.33 ± 2.10 | 72.31 ± 10.55 | 38.41 ± 2.29 | 37.09 ± 3.07 |
| | ID ACC | 65.60 | 66.10 | 60.70 | 65.20 | 64.70 | 65.80 | 65.20 ± 0.50 | 69.47 ± 0.85 | 59.00 | 63.00 | 64.93 ± 1.29 | 52.17 ± 11.42 | 70.03 ± 0.78 | 69.63 ± 0.51 |
| Citeseer-F | AUROC | 78.32 | 49.86 | 49.92 | 79.19 | 74.69 | 78.46 | 84.07 ± 0.41 | 85.11 ± 11.68 | 72.06 | 79.23 | 84.38 ± 0.31 | 84.12 ± 0.41 | 93.96 ± 0.73 | 93.79 ± 0.16 |
| | AUPR | 55.48 | 23.11 | 31.20 | 55.94 | 50.25 | 53.21 | 68.22 ± 0.73 | 75.22 ± 11.74 | 48.80 | 55.69 | 68.77 ± 0.84 | 68.84 ± 0.76 | 84.67 ± 2.35 | 83.61 ± 0.42 |
| | FPR95 | 71.27 | 100.00 | 99.73 | 69.67 | 71.22 | 73.14 | 67.75 ± 1.69 | 60.72 ± 34.01 | 81.09 | 64.08 | 64.64 ± 0.32 | 65.69 ± 3.13 | 23.13 ± 3.55 | 23.32 ± 1.02 |
| | ID ACC | 66.20 | 65.80 | 53.30 | 64.50 | 64.20 | 63.20 | 64.70 ± 0.44 | 68.73 ± 0.47 | 60.50 | 64.40 | 65.03 ± 1.25 | 64.97 ± 1.35 | 68.40 ± 1.93 | 69.33 ± 0.21 |
| Citeseer-L | AUROC | 88.42 | 51.33 | 53.46 | 89.98 | 82.69 | 85.65 | 90.73 ± 0.17 | 90.59 ± 0.69 | 89.44 | 90.34 | 91.37 ± 0.29 | 91.44 ± 0.34 | 90.97 ± 0.23 | 90.77 ± 0.15 |
| | AUPR | 64.03 | 17.97 | 35.47 | 64.10 | 61.21 | 62.32 | 64.93 ± 0.58 | 65.01 ± 1.71 | 62.74 | 66.66 | 64.32 ± 1.55 | 66.00 ± 0.49 | 64.92 ± 1.49 | 64.24 ± 1.75 |
| | FPR95 | 51.97 | 100.00 | 86.32 | 38.76 | 50.61 | 41.37 | 40.36 ± 1.19 | 33.55 ± 5.29 | 45.99 | 31.60 | 28.32 ± 0.95 | 30.35 ± 2.55 | 30.84 ± 0.90 | 27.24 ± 2.76 |
| | ID ACC | 89.36 | 89.36 | 72.51 | 90.58 | 89.16 | 89.30 | 89.46 ± 0.76 | 89.97 ± 0.30 | 87.23 | 90.58 | 88.25 ± 0.17 | 88.66 ± 0.50 | 88.55 ± 0.63 | 88.45 ± 0.34 |

Table 19: **Pubmed** Extended OOD detection performance with two types of OOD (**S**tructure manipulation, **F**eature interpolation). GS++ is short for GNNSAFE++ and NS++ is short for NODESAFE++. Label-leave-out was left out as discussed in K. OE indicates OOD exposure.

| Dataset | Metrics | Non-OOD Exposure | | | | | | | | Real OOD Exposure | | | | TRIBE | |
|---|---|---|---|---|---|---|---|---|---|---|---|---|---|---|---|
| | | MSP | ODIN | Mahalanobis | Energy | GKDE | GPN | GNNSAFE | NODESAFE | OE | Energy FT | GS++ | NS++ | w/o OE | w/ OE |
| Pubmed-S | AUROC | 74.31 | 49.76 | 55.28 | 74.33 | 74.02 | 74.96 | 92.45 ± 1.15 | 93.17 ± 1.15 | 74.41 | 73.54 | 92.84 ± 0.28 | 95.32 ± 0.17 | 97.25 ± 0.58 | 98.46 ± 0.10 |
| | AUPR | 17.44 | 4.83 | 8.38 | 17.32 | 16.89 | 17.54 | 65.47 ± 2.38 | 69.31 ± 4.81 | 16.74 | 18.00 | 70.41 ± 0.33 | 68.07 ± 0.81 | 79.76 ± 2.17 | 85.51 ± 0.38 |
| | FPR95 | 84.08 | 100.00 | 97.59 | 78.90 | 81.52 | 80.33 | 47.81 ± 6.10 | 49.54 ± 15.93 | 83.52 | 92.04 | 47.67 ± 6.87 | 20.32 ± 1.83 | 12.97 ± 3.87 | 7.55 ± 0.57 |
| | ID ACC | 75.10 | 75.30 | 69.30 | 75.60 | 75.20 | 75.80 | 77.00 ± 0.44 | 77.30 ± 0.62 | 72.90 | 75.80 | 77.63 ± 0.31 | 74.20 ± 0.10 | 75.93 ± 0.40 | 76.80 ± 0.52 |
| Pubmed-F | AUROC | 83.28 | 49.67 | 69.12 | 84.16 | 82.25 | 82.56 | 95.18 ± 0.13 | 89.00 ± 11.30 | 82.34 | 78.94 | 94.70 ± 0.51 | 94.96 ± 0.50 | 97.44 ± 0.59 | 98.05 ± 0.25 |
| | AUPR | 39.29 | 4.83 | 39.10 | 39.10 | 32.41 | 39.75 | 74.41 ± 1.56 | 67.81 ± 19.39 | 38.60 | 37.21 | 77.71 ± 0.31 | 76.34 ± 2.06 | 81.70 ± 2.92 | 85.21 ± 0.85 |
| | FPR95 | 69.38 | 100.00 | 84.93 | 62.47 | 68.56 | 61.79 | 27.62 ± 2.55 | 50.80 ± 41.88 | 74.58 | 90.00 | 31.49 ± 5.20 | 29.43 ± 2.90 | 12.53 ± 1.63 | 10.12 ± 2.35 |
| | ID ACC | 75.00 | 75.30 | 73.00 | 75.50 | 74.10 | 74.50 | 76.57 ± 0.35 | 77.33 ± 1.04 | 73.10 | 75.30 | 78.13 ± 0.06 | 74.13 ± 1.16 | 76.40 ± 0.60 | 76.53 ± 0.38 |

Table 20: **Amazon-Photo:** Extended OOD detection performance with three types of OOD (**S**tructure manipulation, **F**eature interpolation, and **L**abel-leave-out).

| Dataset | Metrics | Non-OOD Exposure | | | | | | | | TRIBE |
|---|---|---|---|---|---|---|---|---|---|---|
| | | MSP | ODIN | Mahalanobis | Energy | GKDE | GPN | GNNSAFE | NODESAFE | |
| Amazon-S | AUROC | 98.27 | 93.24 | 71.69 | 98.51 | 76.39 | 97.17 | 98.60 ± 0.09 | 98.54 ± 0.05 | 98.97 ± 0.14 |
| | AUPR | 98.54 | 95.26 | 79.01 | 98.72 | 81.58 | 96.39 | 99.22 ± 0.05 | 99.18 ± 0.03 | 99.42 ± 0.08 |
| | FPR95 | 6.13 | 65.44 | 99.91 | 4.97 | 99.25 | 11.65 | 0.00 ± 0.00 | 0.00 ± 0.00 | 0.00 ± 0.00 |
| | ID ACC | 92.84 | 92.84 | 92.79 | 92.86 | 87.57 | 88.51 | 92.64 ± 0.44 | 91.36 ± 0.02 | 92.95 ± 0.38 |
| Amazon-F | AUROC | 97.31 | 81.15 | 76.50 | 97.87 | 58.96 | 87.91 | 98.46 ± 0.01 | 98.42 ± 0.03 | 98.65 ± 0.10 |
| | AUPR | 95.16 | 78.47 | 71.14 | 95.64 | 66.76 | 84.77 | 98.90 ± 0.15 | 98.77 ± 0.17 | 99.04 ± 0.22 |
| | FPR95 | 8.72 | 100.0 | 76.12 | 6.00 | 99.28 | 49.11 | 0.44 ± 0.28 | 0.30 ± 0.05 | 0.30 ± 0.06 |
| | ID ACC | 92.89 | 92.71 | 92.86 | 92.96 | 86.18 | 90.05 | 92.65 ± 0.55 | 91.55 ± 0.41 | 92.70 ± 0.33 |
| Amazon-L | AUROC | 93.97 | 65.97 | 73.25 | 93.81 | 65.58 | 92.72 | 96.92 ± 0.33 | 96.56 ± 0.35 | 96.86 ± 0.35 |
| | AUPR | 91.32 | 57.80 | 66.89 | 91.13 | 65.20 | 90.34 | 96.38 ± 0.70 | 95.35 ± 0.58 | 95.79 ± 0.39 |
| | FPR95 | 26.65 | 90.23 | 74.30 | 28.48 | 96.87 | 37.16 | 9.29 ± 1.15 | 10.78 ± 0.99 | 8.13 ± 3.55 |
| | ID ACC | 95.76 | 96.08 | 95.76 | 95.72 | 89.37 | 90.07 | 96.10 ± 0.29 | 95.20 ± 0.11 | 96.07 ± 0.20 |

Table 21: **Coauthor:** Extended OOD detection performance with three types of OOD (**S**tructure manipulation, **F**eature interpolation, and **L**abel-leave-out).

| Dataset | Metrics | Non-OOD Exposure | | | | | | | | TRIBE |
|---|---|---|---|---|---|---|---|---|---|---|
| | | MSP | ODIN | Mahalanobis | Energy | GKDE | GPN | GNNSAFE | NODESAFE | |
| Coauthor-S | AUROC | 95.30 | 52.14 | 80.46 | 96.18 | 65.87 | 34.67 | 99.80 ± 0.17 | 99.85 ± 0.07 | 99.95 ± 0.02 |
| | AUPR | 94.37 | 48.83 | 76.65 | 95.25 | 72.65 | 40.21 | 99.82 ± 0.11 | 99.85 ± 0.06 | 99.94 ± 0.02 |
| | FPR95 | 24.75 | 99.92 | 70.75 | 18.02 | 99.48 | 99.57 | 0.26 ± 0.02 | 0.23 ± 0.06 | 0.13 ± 0.03 |
| | ID ACC | 92.47 | 92.34 | 92.33 | 92.75 | 88.62 | 89.45 | 92.81 ± 0.10 | 92.11 ± 0.26 | 92.72 ± 0.11 |
| Coauthor-F | AUROC | 97.05 | 51.54 | 93.23 | 97.88 | 80.69 | 81.77 | 99.80 ± 0.14 | 99.86 ± 0.07 | 99.94 ± 0.02 |
| | AUPR | 96.93 | 45.50 | 90.88 | 97.69 | 86.47 | 80.56 | 99.78 ± 0.11 | 99.82 ± 0.07 | 99.92 ± 0.04 |
| | FPR95 | 15.55 | 100.0 | 28.10 | 9.75 | 96.57 | 74.46 | 0.39 ± 0.11 | 0.30 ± 0.03 | 0.19 ± 0.05 |
| | ID ACC | 92.45 | 92.39 | 92.34 | 92.75 | 84.72 | 87.05 | 95.38 ± 0.16 | 95.35 ± 0.15 | 95.47 ± 0.07 |
| Coauthor-L | AUROC | 94.88 | 51.44 | 85.36 | 95.87 | 61.15 | | 97.35 ± 0.26 | 97.34 ± 0.27 | 97.92 ± 0.04 |
| | AUPR | 97.99 | 74.79 | 93.61 | 98.34 | 81.39 | 97.55 | 99.03 ± 0.10 | 99.03 ± 0.10 | 99.24 ± 0.04 |
| | FPR95 | 23.81 | 100.0 | 45.41 | 18.69 | 94.60 | 34.78 | 12.22 ± 1.09 | 12.46 ± 1.22 | 9.60 ± 0.60 |
| | ID ACC | 95.18 | 95.15 | 95.19 | 95.20 | 89.05 | 91.68 | 95.38 ± 0.16 | 95.35 ± 0.15 | 95.47 ± 0.07 |

Table 22: **Twitch:** Extended OOD detection performance on OOD `Twitch` sub-graphs ES, FR and RU. GS++ is short for GNNSAFE++ and NS++ is short for NODESAFE++. OE indicates OOD exposure.

| Dataset | Metrics | Non-OOD Exposure | | | | | | | | Real OOD Exposure | | | | TRIBE | |
|---|---|---|---|---|---|---|---|---|---|---|---|---|---|---|---|
| | | MSP | ODIN | Mahalanobis | Energy | GKDE | GPN | GNNSAFE | NODESAFE | OE | Energy FT | GS++ | NS++ | w/o OE | w/ OE |
| Twitch-ES | AUROC | 37.72 | 83.83 | 45.66 | 38.80 | 48.70 | 53.00 | 70.20 ± 18.91 | 70.66 ± 18.90 | 55.97 | 80.73 | 94.53 ± 2.47 | 95.48 ± 1.20 | 95.97 ± 1.31 | 96.80 ± 1.33 |
| | AUPR | 53.08 | 80.43 | 58.82 | 54.26 | 61.05 | 64.24 | 75.88 ± 17.90 | 76.13 ± 17.89 | 69.49 | 87.56 | 97.14 ± 1.33 | 97.42 ± 0.47 | 96.30 ± 0.57 | 97.10 ± 2.47 |
| | FPR95 | 98.09 | 33.28 | 95.48 | 95.70 | 95.37 | 95.05 | 90.48 ± 4.55 | 89.56 ± 5.20 | 94.94 | 76.76 | 40.36 ± 22.32 | 27.99 ± 17.57 | 17.70 ± 13.41 | 7.38 ± 7.06 |
| | ID ACC | 68.72 | 70.79 | 70.51 | 70.40 | 67.44 | 68.09 | 70.75 ± 0.69 | 70.75 ± 0.69 | 70.73 | 70.52 | 70.36 ± 0.30 | 70.07 ± 0.33 | 68.63 ± 1.41 | 68.74 ± 0.79 |
| Twitch-FR | AUROC | 21.82 | 59.82 | 40.40 | 57.21 | 49.19 | 51.25 | 49.44 ± 28.72 | 49.38 ± 29.23 | 45.66 | 79.66 | 93.91 ± 1.73 | 37.28 ± 52.74 | 86.80 ± 12.59 | 84.22 ± 16.97 |
| | AUPR | 38.27 | 64.63 | 46.69 | 61.48 | 52.94 | 55.37 | 57.81 ± 20.56 | 57.90 ± 20.87 | 54.03 | 81.20 | 95.94 ± 0.99 | 56.13 ± 36.96 | 89.05 ± 11.22 | 88.71 ± 12.74 |
| | FPR95 | 99.25 | 92.57 | 95.54 | 91.57 | 95.04 | 93.92 | 94.69 ± 7.17 | 94.62 ± 7.25 | 95.48 | 76.39 | 45.82 ± 13.57 | 67.09 ± 55.37 | 52.43 ± 40.23 | 65.89 ± 27.51 |
| | ID ACC | 68.72 | 70.79 | 70.51 | 70.40 | 67.44 | 68.09 | 70.75 ± 0.69 | 70.75 ± 0.69 | 70.73 | 76.39 | 70.36 ± 0.30 | 70.07 ± 0.33 | 68.63 ± 1.41 | 68.74 ± 0.79 |
| Twitch-RU | AUROC | 41.23 | 58.67 | 55.68 | 57.72 | 46.48 | 50.89 | 79.34 ± 16.34 | 79.39 ± 16.59 | 55.72 | 93.12 | 98.79 ± 0.92 | 97.60 ± 0.57 | 86.38 ± 0.77 | 93.53 ± 0.79 |
| | AUPR | 56.06 | 72.58 | 66.42 | 66.68 | 62.11 | 65.14 | 84.07 ± 9.53 | 84.09 ± 9.73 | 70.18 | 95.36 | 99.28 ± 0.59 | 98.36 ± 0.64 | 90.41 ± 2.50 | 94.95 ± 4.94 |
| | FPR95 | 95.01 | 93.98 | 90.13 | 87.57 | 95.62 | 99.93 | 57.37 ± 34.19 | 57.67 ± 34.76 | 95.07 | 30.72 | 3.25 ± 3.97 | 8.29 ± 5.27 | 69.67 ± 7.40 | 26.29 ± 18.01 |
| | ID ACC | 68.72 | 70.79 | 70.51 | 70.40 | 67.44 | 68.09 | 70.75 ± 0.69 | 70.75 ± 0.69 | 70.73 | 70.52 | 70.36 ± 0.30 | 70.07 ± 0.33 | 68.63 ± 1.41 | 68.74 ± 0.79 |

Table 23: **Arxiv:** Extended OOD detection performance on OOD dataset of papers published in 2018, 2019, and 2020. GS++ is short for GNNSAFE++ and NS++ is short for NODESAFE++. OE indicates OOD exposure.

| Dataset | Metrics | Non-OOD Exposure | | | | | | | | Real OOD Exposure | | | | TRIBE | |
|---|---|---|---|---|---|---|---|---|---|---|---|---|---|---|---|
| | | MSP | ODIN | Mahalanobis | Energy | GKDE | GPN | GNNSAFE | NODESAFE | OE | Energy FT | GS++ | NS++ | w/o OE | w/ OE |
| Arxiv-2018 | AUROC | 61.66 | 53.49 | 57.08 | 61.75 | 56.29 | OOM | 65.94 ± 0.38 | 66.73 ± 0.16 | 67.72 | 69.58 | 69.47 ± 0.63 | 69.81 ± 0.58 | 67.97 ± 0.10 | 70.5 ± 0.20 |
| | AUPR | 70.63 | 63.06 | 65.09 | 70.41 | 66.78 | OOM | 74.37 ± 0.29 | 75.20 ± 0.27 | 75.74 | 76.31 | 77.63 ± 0.64 | 77.68 ± 0.48 | 76.42 ± 0.20 | 78.59 ± 0.21 |
| | FPR95 | 91.67 | 100.0 | 93.69 | 91.74 | 94.31 | OOM | 89.88 ± 0.67 | 88.80 ± 0.37 | 86.67 | 82.10 | 83.43 ± 1.24 | 81.51 ± 1.19 | 86.88 ± 0.48 | 81.57 ± 0.25 |
| | ID ACC | 53.78 | 51.39 | 51.59 | 53.36 | 50.76 | OOM | 53.21 ± 0.16 | 53.10 ± 0.79 | 52.39 | 53.26 | 53.28 ± 0.35 | 51.32 ± 0.21 | 52.44 ± 0.24 | 52.41 ± 0.29 |
| Arxiv-2019 | AUROC | 63.07 | 53.95 | 56.76 | 63.16 | 57.87 | OOM | 67.90 ± 0.37 | 68.76 ± 0.16 | 69.33 | 70.58 | 71.32 ± .63 | 71.87 ± 0.59 | 70.09 ± 0.21 | 72.30 ± 0.23 |
| | AUPR | 66.00 | 56.07 | 57.85 | 65.78 | 62.34 | OOM | 70.97 ± 0.29 | 71.98 ± .33 | 72.15 | 72.03 | 74.45 ± 0.77 | 74.74 ± 0.68 | 73.24 ± 0.24 | 75.56 ± 0.30 |
| | FPR95 | 90.82 | 100.0 | 94.01 | 90.96 | 93.97 | OOM | 88.83 ± 0.72 | 87.44 ± 0.23 | 85.52 | 81.30 | 81.80 ± 1.37 | 79.31 ± 1.13 | 85.10 ± 0.75 | 79.39 ± 0.38 |
| | ID ACC | 53.78 | 51.39 | 51.59 | 53.36 | 50.76 | OOM | 53.21 ± 0.16 | 53.10 ± 0.79 | 52.39 | 53.26 | 53.28 ± 0.35 | 51.32 ± 0.21 | 52.44 ± 0.24 | 52.41 ± 0.29 |
| Arxiv-2020 | AUROC | 67.00 | 55.78 | 56.92 | 67.70 | 60.79 | OOM | 77.90 ± 0.32 | 78.59 ± 0.10 | 72.35 | 74.53 | 81.15 ± 0.51 | 81.82 ± 0.59 | 80.10 ± 0.23 | 82.03 ± 0.23 |
| | AUPR | 90.92 | 87.41 | 85.95 | 91.15 | 88.74 | OOM | 94.64 ± 0.08 | 94.83 ± 0.03 | 92.57 | 93.08 | 95.43 ± .12 | 92.23 ± 5.66 | 95.19 ± 0.07 | 95.59 ± 0.06 |
| | FPR95 | 89.28 | 100.0 | 95.01 | 89.69 | 93.31 | OOM | 85.00 ± 0.98 | 83.11 ± 0.42 | 83.28 | 78.36 | 74.75 ± 1.99 | 71.78 ± 1.32 | 78.98 ± 0.56 | 71.15 ± 0.62 |
| | ID ACC | 53.78 | 51.39 | 51.59 | 53.36 | 50.76 | OOM | 53.21 ± 0.16 | 53.10 ± 0.79 | 52.39 | 53.26 | 53.28 ± 0.35 | 51.32 ± 0.21 | 52.44 ± 0.24 | 52.41 ± 0.29 |

Table 24: **Cora:** Extended ablation performance of TRIBE.

| Model | Cora-S | | | | Cora-F | | | | Cora-L | | | |
|---|---|---|---|---|---|---|---|---|---|---|---|---|
| | AUROC | AUPR | FPR95 | ID Acc | AUROC | AUPR | FPR95 | ID Acc | AUROC | AUPR | FPR95 | ID Acc |
| GNNSAFE | 87.64 ± 0.39 | 77.93 ± 0.59 | 73.49 ± 1.91 | 77.53 ± 0.38 | 92.76 ± 0.43 | 87.85 ± 1.07 | 48.56 ± 4.52 | 77.50 ± 0.26 | 93.19 ± 0.11 | 82.99 ± 0.45 | 29.55 ± 2.24 | 89.66 ± 0.66 |
| $\mathcal{L}_{\text{VIB}}$ | 94.79 ± 0.22 | 88.41 ± 0.49 | 26.22 ± 2.55 | 78.73 ± 0.50 | 97.69 ± 0.13 | 93.94 ± 0.36 | 9.45 ± 0.78 | 77.43 ± 0.58 | 93.06 ± 0.11 | 83.12 ± 0.11 | 30.93 ± 0.57 | 90.30 ± 0.37 |
| $\mathcal{L}_{\text{VIB}}$ & $\mathcal{L}_{\text{VCInd}}$ | 94.86 ± 0.29 | 88.48 ± 0.54 | 24.84 ± 0.65 | 78.60 ± 0.40 | 97.72 ± 0.10 | 94.26 ± 0.52 | 9.40 ± 1.06 | 77.47 ± 0.45 | 93.11 ± 0.16 | 83.20 ± 0.18 | 30.43 ± 0.11 | 90.30 ± 0.18 |
| TRIBE | 95.15 ± 0.37 | 89.33 ± 0.57 | 23.31 ± 1.04 | 78.97 ± 1.22 | 97.88 ± 0.24 | 94.67 ± 0.56 | 8.13 ± 1.32 | 78.10 ± 0.89 | 93.70 ± 0.04 | 84.03 ± 0.08 | 28.84 ± 0.36 | 90.61 ± 0.36 |

Table 25: **Citeseer:** Extended ablation performance of TRIBE.

| Model | Citeseer-S | | | | Citeseer-F | | | | Citeseer-L | | | |
|---|---|---|---|---|---|---|---|---|---|---|---|---|
| | AUROC | AUPR | FPR | ID Acc | AUROC | AUPR | FPR | ID Acc | AUROC | AUPR | FPR | ID Acc |
| GNNSAFE | 79.87 ± 0.56 | 61.44 ± 1.50 | 74.45 ± 0.59 | 65.20 ± 0.50 | 84.07 ± 0.41 | 68.22 ± 0.73 | 67.75 ± 1.69 | 64.70 ± 0.44 | 90.73 ± 0.17 | 64.93 ± 0.58 | 40.36 ± 1.19 | 89.46 ± 0.76 |
| $\mathcal{L}_{\text{VIB}}$ | 90.84 ± 0.57 | 78.76 ± 0.47 | 45.62 ± 3.06 | 69.17 ± 0.49 | 93.01 ± 0.64 | 82.83 ± 1.64 | 31.77 ± 5.79 | 69.27 ± 0.49 | 90.66 ± 0.10 | 64.06 ± 0.67 | 34.87 ± 3.12 | 87.74 ± 0.35 |
| $\mathcal{L}_{\text{VIB}}$ & $\mathcal{L}_{\text{VCInd}}$ | 91.85 ± 0.31 | 80.29 ± 0.69 | 39.61 ± 2.68 | 69.13 ± 0.21 | 93.25 ± 0.87 | 82.94 ± 2.15 | 26.96 ± 4.64 | 67.83 ± 1.42 | 90.88 ± 0.25 | 64.78 ± 1.62 | 31.14 ± 1.19 | 88.75 ± 0.31 |
| TRIBE | 91.89 ± 0.36 | 80.11 ± 0.89 | 38.41 ± 2.29 | 70.03 ± 0.78 | 93.96 ± 0.73 | 84.67 ± 2.35 | 23.13 ± 3.55 | 68.40 ± 1.93 | 90.97 ± 0.23 | 64.92 ± 1.49 | 30.84 ± 0.90 | 88.55 ± 0.63 |

Table 26: **Pubmed:** Extended ablation performance of TRIBE.

| Model | Pubmed-S | | | | Pubmed-F | | | |
|---|---|---|---|---|---|---|---|---|
| | AUROC | AUPR | FPR95 | ID Acc | AUROC | AUPR | FPR95 | ID Acc |
| GNNSAFE | 92.45 ± 1.15 | 65.47 ± 2.38 | 47.81 ± 6.10 | 77.00 ± 0.44 | 95.18 ± 0.13 | 74.41 ± 1.56 | 27.62 ± 2.55 | 76.57 ± 0.35 |
| $\mathcal{L}_{\text{VIB}}$ | 96.18 ± 0.44 | 75.74 ± 1.60 | 17.64 ± 3.69 | 74.60 ± 1.71 | 96.98 ± 0.41 | 78.94 ± 3.33 | 16.05 ± 1.64 | 75.30 ± 1.22 |
| $\mathcal{L}_{\text{VIB}}$ & $\mathcal{L}_{\text{VCInd}}$ | 96.45 ± 0.64 | 76.46 ± 2.31 | 15.66 ± 4.88 | 74.77 ± 1.86 | 97.42 ± 0.51 | 81.07 ± 3.23 | 13.08 ± 2.41 | 75.60 ± 0.46 |
| TRIBE | 97.25 ± 0.58 | 79.76 ± 2.17 | 12.97 ± 3.87 | 75.93 ± 0.40 | 97.44 ± 0.59 | 81.70 ± 2.92 | 12.53 ± 1.63 | 76.40 ± 0.60 |

Table 27: **Twitch:** Extended ablation performance of TRIBE.

| Model | Twitch-ES | | | | Twitch-FR | | | | Twitch-RU | | | |
|---|---|---|---|---|---|---|---|---|---|---|---|---|
| | AUROC | AUPR | FPR95 | ID Acc | AUROC | AUPR | FPR95 | ID Acc | AUROC | AUPR | FPR95 | ID Acc |
| GNNSAFE | 70.20 ± 18.91 | 75.88 ± 17.90 | 90.48 ± 4.55 | 70.75 ± 0.69 | 49.44 ± 28.72 | 57.81 ± 20.56 | 94.69 ± 7.17 | 70.75 ± 0.69 | 79.34 ± 16.34 | 84.07 ± 9.53 | 58.37 ± 34.19 | 70.75 ± 0.69 |
| $\mathcal{L}_{\text{VIB}}$ | 83.29 ± 11.53 | 86.36 ± 10.78 | 56.60 ± 33.50 | 70.36 ± 0.35 | 57.97 ± 14.37 | 58.80 ± 12.38 | 88.70 ± 13.46 | 70.36 ± 0.35 | 79.23 ± 13.76 | 84.86 ± 8.38 | 62.48 ± 36.82 | 70.36 ± 0.35 |
| $\mathcal{L}_{\text{VIB}}$ & $\mathcal{L}_{\text{VCInd}}$ | 95.04 ± 0.11 | 96.05 ± 0.13 | 19.35 ± 2.60 | 70.33 ± 0.63 | 75.14 ± 1.03 | 80.85 ± 0.58 | 90.69 ± 5.46 | 70.33 ± 0.63 | 92.67 ± 2.40 | 94.93 ± 0.93 | 39.68 ± 23.80 | 70.33 ± 0.63 |
| TRIBE | 95.97 ± 1.31 | 96.30 ± 0.57 | 17.70 ± 13.41 | 68.63 ± 1.41 | 86.80 ± 12.59 | 89.05 ± 11.22 | 52.43 ± 40.23 | 68.63 ± 1.41 | 86.38 ± 0.77 | 90.41 ± 2.50 | 69.67 ± 7.40 | 68.63 ± 1.41 |

Table 28: **ArXiv:** Extended ablation performance of TRIBE.

| Model | ArXiv-18 | | | | ArXiv-19 | | | | ArXiv-20 | | | |
|---|---|---|---|---|---|---|---|---|---|---|---|---|
| | AUROC | AUPR | FPR95 | ID Acc | AUROC | AUPR | FPR95 | ID Acc | AUROC | AUPR | FPR95 | ID Acc |
| GNNSAFE | $65.94 \pm 0.38$ | $74.37 \pm 0.29$ | $89.88 \pm 0.67$ | $53.21 \pm 0.16$ | $67.90 \pm 0.37$ | $70.97 \pm 0.29$ | $88.83 \pm 0.72$ | $53.21 \pm 0.16$ | $77.90 \pm 0.32$ | $94.64 \pm 0.08$ | $85.00 \pm 0.98$ | $53.21 \pm 0.16$ |
| $\mathcal{L}_{\text{VIB}}$ | $67.41 \pm 0.95$ | $75.84 \pm 1.02$ | $87.72 \pm 1.28$ | $52.46 \pm 0.24$ | $69.49 \pm 1.08$ | $72.51 \pm 1.09$ | $86.35 \pm 1.35$ | $52.46 \pm 0.24$ | $79.53 \pm 1.04$ | $95.05 \pm 0.27$ | $80.72 \pm 1.85$ | $52.46 \pm 0.24$ |
| $\mathcal{L}_{\text{VIB}}$ & $\mathcal{L}_{\text{VCInd}}$ | $67.95 \pm 0.11$ | $76.40 \pm 0.21$ | $86.97 \pm 0.46$ | $52.43 \pm 0.22$ | $70.03 \pm 0.25$ | $73.20 \pm 0.23$ | $85.43 \pm 0.89$ | $52.43 \pm 0.22$ | $80.07 \pm 0.25$ | $95.12 \pm 0.17$ | $79.18 \pm 0.65$ | $52.43 \pm 0.22$ |
| TRIBE | $67.97 \pm 0.10$ | $76.42 \pm 0.20$ | $86.88 \pm 0.48$ | $52.44 \pm 0.24$ | $70.09 \pm 0.21$ | $73.24 \pm 0.24$ | $85.10 \pm 0.75$ | $52.44 \pm 0.24$ | $80.10 \pm 0.23$ | $95.19 \pm 0.07$ | $78.98 \pm 0.56$ | $52.44 \pm 0.24$ |

# M COMPUTATIONAL COST

Table 29 presents a comparison of the computational cost of TRIBE against **GNNSafe** and NODE-SAFE, all evaluated using the same backbone. Evidently, **TRIBE consistently outperforms the baselines, achieving superior OOD detection performance at the cost of a 2-3x increase in training time, a marginal rise in memory usage, and a comparable inference time**. Despite the additional training cost, this **trade-off is deemed acceptable due to the significantly improved detection performance** and **competitive inference time**, as discussed in Appendix A. These results highlight the strong performance and practical applicability of TRIBE for node-level graph OOD detection.

Table 29: **Computation cost (single 48GB (49140MiB) NVIDIA RTX A6000 GPU) and OOD detection performance of TRIBE against SOTA Non-OOD exposure baselines.** The 'Train' column is the training convergence time in seconds. The 'In.' column is the inference time in seconds. The 'Mem.' column is the maximum memory usage in Mebibytes (MiB). The 'FPR95' column is the OOD detection performance in %, the lower the better.

| | Cora – Structure | | | | Citeseer – Structure | | | | Pubmed – Structure | | | | Twitch | | | | Arxiv | | | |
|---|---|---|---|---|---|---|---|---|---|---|---|---|---|---|---|---|---|---|---|---|
| | Train | In. | Mem. | FPR95($\downarrow$) | Train | In. | Mem. | FPR95($\downarrow$) | Train | In. | Mem. | FPR95($\downarrow$) | Train | In. | Mem. | FPR95($\downarrow$) | Train | In. | Mem. | FPR95($\downarrow$) |
| GNNSAFE | 2.05 | 0.01 | 625 | 73.49 | 2.25 | 0.02 | 693 | 74.45 | 4.22 | 0.02 | 827 | 47.81 | 3.55 | 0.03 | 749 | 81.18 | 19.72 | 0.11 | 3055 | 87.90 |
| NODESAFE | 0.76 | 0.02 | 625 | 38.06 | 1.45 | 0.02 | 693 | 65.99 | 1.69 | 0.02 | 827 | 49.54 | 1.35 | 0.03 | 749 | 80.62 | 16.79 | 0.11 | 3055 | 86.45 |
| TRIBE | 4.02 | 0.02 | 653 | 23.31 | 9.85 | 0.02 | 780 | 38.41 | 9.96 | 0.024 | 906 | 12.97 | 8.01 | 0.028 | 794 | 43.14 | 39.7 | 0.13 | 3490 | 83.65 |

