# OpenReview forum: "Tribe: Tri-Component Information Decomposition for Graph Out-of-Distribution Detection"
_ICLR.cc/2026/Conference — Submitted to ICLR 2026_

### Official Review · Reviewer_pQr9 · 2025-10-24

**Soundness:** 4
**Presentation:** 4
**Contribution:** 4
**Rating:** 8
**Confidence:** 5

**Summary:**

This paper approaches the problem from the perspective of the information bottleneck in information theory, aiming to train classifiers using only the information most relevant to the labels, thereby shielding the model from interference caused by redundant information and further improving its performance on the task it investigates: out-of-distribution node detection. In simple terms, the paper employs three networks: a backbone network to extract information most critical to the labels, and two branch networks designed to disentangle redundant information arising from features and structure, respectively. Overall, the paper has a natural and reasonable motivation, excellent writing, thorough experiments, and rigorous theoretical grounding, making it a high-quality contribution.

**Strengths:**

1. The writing is excellent and well-structured, from which I have learned a great deal. Thank you to the authors.
2. The motivation is natural and reasonable. Although the information bottleneck is not a novel theory, I believe its successful application to out-of-distribution detection brings significant innovation and ample room for further extension.
3. I have reviewed the theoretical part and did not find any major issues.
4. The experiments are comprehensive and thorough, making the results highly convincing.

**Weaknesses:**

I don't have major concerns; however, although the experiments are substantial, I feel that some recent baselines are missing (even though they have been presented in the related work). If possible and feasible, including comparisons with these latest baselines, either experimentally or in the textual discussion, would further strengthen the paper.

**Questions:**

See the weaknesses mentioned above.

**Details Of Ethics Concerns:**

No.

---

> ### Author Response · Authors · 2025-11-18
> **Response to Reviewer pQr9 (1/1)**
>
> We thank the reviewer for their strong recognition of our framework’s broad appeal, and we are grateful that you enjoyed reading the paper. We appreciate the positive assessment of our theoretical insights, empirical results, and extensive validation. We address your suggestion as follows.
>
> ---
> > **Re Experimental Completeness:**
>
> We agree that incorporating recent baselines provides useful additional context. Our core experiments focus on the **standard supervised-learning setting**, enabling us to directly assess the effect of our information-theoretic objective **without confounding factors** introduced by advanced or exposure-based paradigms. Recent approaches discussed in the related work, such as **diffusion-based pseudo-OOD generation** (GOLD) or **multi-hop energy modelling** specialised for heterophily (DeGEM), operate under assumptions distinct from our investigation of **feature-structure entanglement** and thus do not serve as clean baselines for our primary research question.
>
> **Additional experiments:** Nevertheless, in response to the reviewer’s recommendation, we have conducted **extended experiments** comparing TRIBE with these more advanced graph OOD detectors. To ensure a fairer alignment with our information-theoretic analysis, we additionally provide **"+TRIBE " variants** of GOLD and DeGEM by integrating our TRIBE objective into their training pipelines.
>
> These results demonstrate several key points:
>
> - Both baselines show strong performance, reflecting the strength of their respective modelling choices.
> - The **TRIBE-augmented variants** often improve detection performance relative to their original versions, supporting our theoretical prediction that **IB increases ID-OOD separability**.
> - While not exceeding these models in every setting, **TRIBE consistently matches or remains competitive**, highlighting the benefit of **explicitly decomposing joint, feature-only, and structure-only information** for stable and general graph OOD detection.
>
> We have updated the manuscript to include this discussion (Section 6.5, Table 4). We sincerely thank the reviewer for their encouraging feedback and thoughtful suggestion.
> | Dataset | Method | AUROC | AUPR | FPR95 | ID ACC |
> |-|-|-|-|-|-|
> | **Cora -Label** | GNNSafe | 93.19 | 82.99 | 29.55 | 89.66 |
> || TRIBE | 93.70 | 84.03 | 28.84 | 90.61 |
> || DeGEM | 92.24 | 78.80 | 31.34 | 91.77 |
> || DeGEM + TRIBE | 95.85 | 89.24 | 20.59 | 92.41 |
> || GOLD | 95.36 | 85.33 | 21.20 | 89.56 |
> || GOLD + TRIBE | 94.85 | 83.96 | 18.86 | 89.56 |
> | **Pubmed -S** | GNNSafe | 92.45 | 65.47 | 47.81 | 77.00 |
> || TRIBE | 97.25 | 79.76 | 12.97 | 75.93 |
> || DeGEM | 95.37 | 51.55 | 20.07 | 73.10 |
> || DeGEM + TRIBE | 97.20 | 55.60 | 8.44 | 78.40 |
> || GOLD | 88.01 | 97.84 | 11.57 | 75.30 |
> || GOLD + TRIBE | 91.13 | 98.72 | 5.52 | 74.60 |
> |**Pubmed -F** | GNNSafe | 95.18 | 74.41 | 27.62 | 76.57 |
> || TRIBE | 97.44 | 81.70 | 12.53 | 76.40 |
> || DeGEM | 99.63 | 93.40 | 1.70 | 79.00 |
> || DeGEM + TRIBE | 99.67 | 91.73 | 0.59 | 78.70 |
> || GOLD | 87.28 | 98.49 | 6.98 | 73.20 |
> || GOLD + TRIBE | 93.15 | 99.00 | 3.80 | 74.70 |
> | **Amazon-F** | GNNSafe | 98.46 | 98.90 | 0.44 | 92.65 |
> || TRIBE | 98.65 | 99.04 | 0.30 | 92.70 |
> || DeGEM | 97.34 | 96.70 | 3.16 | 91.65 |
> || DeGEM + TRIBE | 97.71 | 96.49 | 1.84 | 92.32 |
> || GOLD | 99.52 | 99.63 | 0.24 | 92.48 |
> || GOLD + TRIBE | 99.61 | 99.70 | 0.10 | 91.98 |
> | **Citeseer -S** | GNNSafe | 79.87 | 61.44 | 74.45 | 65.20 |
> || TRIBE | 91.89 | 80.11 | 38.41 | 70.03 |
> || DeGEM | 94.93 | 84.63 | 17.49 | 70.90 |
> || DeGEM + TRIBE | 96.92 | 85.63 | 7.24 | 69.70 |
> || GOLD | 78.09 | 82.12 | 65.98 | 69.10 |
> || GOLD + TRIBE | 78.61 | 82.90 | 44.97 | 68.70 |

---

> ### Comment · Reviewer_pQr9 · 2025-11-25
>
> Thank you to the authors for their response, which has helped make the paper more complete. I have no further questions. I will maintain my positive score and support the acceptance of this paper.

---

> > ### Author Response · Authors · 2025-11-25
> >
> > Dear Reviewer pQr9,
> >
> > Thank you once again for your recognition and the thoughtful feedback on our submission. We will ensure to integrate all clarifications and additional experiments in a future revision of the manuscript.

---

### Official Review · Reviewer_uf9e · 2025-10-31

**Soundness:** 2
**Presentation:** 3
**Contribution:** 2
**Rating:** 4
**Confidence:** 5

**Summary:**

Summary
This paper introduces TRIBE, a tri-component information decomposition framework for graph out-of-distribution (OOD) detection in node classification tasks. It addresses GNN vulnerabilities to feature, structure, or joint shifts by decomposing label information into feature-specific (V), structure-specific (Q), and joint (Z) components, filtering spurious individual-input correlations via an information bottleneck (IB) objective, conditional independence regularizer, and pairwise mutual information minimization. Theoretical analysis proves IB enhances ID confidence and entropy gap for better logit-based detection. Experiments on seven datasets show up to 34% FPR95 improvement over baselines like GNNSAFE, while preserving competitive ID accuracy.

**Strengths:**

1. Clear Structure: The paper follows a standard academic format (abstract, introduction, related work, preliminaries, method, theoretical insights), with smooth transitions between sections. This enhances readability and guides the reader through complex ideas, from problem motivation to theoretical proofs and implementation details.
2. Theoretical Rigor: Provides solid proofs (e.g., on IB's superiority over SL in ID confidence and entropy separation), offering clear insights into why the framework improves detection under shifts.

**Weaknesses:**

1. Unclear motivation.

The Abstract claim that “standard supervised learning (SL) objectives tend to capture spurious signals from either features and/or structure” lacks empirical evidence. Figure 1 in the Introduction suggests that SL representations mix feature-, structure-, and label-irrelevant components, but no diagnostic experiment supports this assumption. Similar claims have already been made by prior Information Bottleneck or invariant representation learning studies, so the novelty of this motivation is limited [1,2,3].

2. Intrinsic conflicts in the mutual-information optimization.

(a) Conflict between maximizing task relevance and minimizing redundancy:
The IB objectives encourage each component (Z, V, Q) to maximize its mutual information with the label Y, but the pairwise regularization (e.g., min I(Z; V)) penalizes their overlap.
When features and structure are strongly correlated, these goals may compete, reducing predictive strength and optimization stability.

(b) Conflict between compression and conditional independence:
The compression term min I(X, A; Z) aims to remove irrelevant noise, yet excessive compression may discard the necessary X–A interactions required for min I(A; X | Z)=0.
If X and A are intrinsically dependent, these objectives can become contradictory, leading to degenerate or unstable representations.

3. Experimental limitations.

(a) The paper omits comparisons with recent strong baselines such as DeGEM and GOLD, which weakens the claim of comprehensive SOTA superiority.

(b) In 6.5 energy gap, while the article defines energy-based OOD detection in Section 3 and uses energy scores for inference, the visualization shows "greater separation between ID and OOD energy scores" without explicitly explaining how this empirical energy gap maps to or supports the theoretical "entropy gap" in Section 5 (Proposition 5.3). The scores are derived from logits (related to entropy), but the lack of a clear connection between mutual information-based theory and energy visualization may seem inconsistent.

[1]Zhang, Ge, et al. "Enhancing graph neural networks for out-of-distribution graph detection." IEEE Transactions on Neural Networks and Learning Systems (2025).
[2] Ren, Lingfei, et al. "Heterophilic graph invariant learning for out-of-distribution of fraud detection." Proceedings of the 32nd ACM International Conference on Multimedia. 2024.
[3] Li, Zenan, et al. "Graphde: A generative framework for debiased learning and out-of-distribution detection on graphs." Advances in Neural Information Processing Systems 35 (2022): 30277-30290.

**Questions:**

1. Could the authors provide motivational experiments or additional evidence to justify why the tri-component decomposition is necessary, especially compared to existing IB-based or invariant learning methods that also aim to reduce spurious correlations?

2. In weakness (2), how do the authors address the potential conflict during optimization?

---

> ### Author Response · Authors · 2025-11-18
> **Response to Reviewer uf9e (1/5)**
>
> We greatly appreciate the time and effort dedicated to providing us with constructive comments. We have thoroughly addressed your concerns as follows.
>
> ---
> > **W1 Motivation clarification:**
>
> We appreciate the reviewer’s comment and agree that prior work in information bottleneck and invariant representation learning has shown that SL-trained models may rely on spurious correlations. We reference several examples of this phenomenon in the introduction. We clarify here that this is **precisely a major motivation for our work**, and our contribution is not to re-establish this general observation, but to **investigate how this issue manifests uniquely in the graph OOD problem** and to develop a framework tailored to addressing it for node-level OOD detection.
>
> The goal of TRIBE is **not** to reiterate a known limitation of SL, but to introduce a **new graph-specific solution**, where we:
>
> * **formally decompose** $I(X,A;Y)$ into joint, feature-specific, and structure-specific components following a information-theoretic perspective;
> * introduce **dedicated encoders** to realise this decomposition in practice, along with **conditional-independence and pairwise–mutual-information regularisation** to ensure separation; and
> * provide **new theory** showing that IB *increases the entropy gap* relevant for logit-based OOD detection, which has not been established for graph OOD.
>
> Together with the ablation study (Table 5) and representation visualisations (Figure 4), our experiments **demonstrate that this decomposition directly improves OOD separability**, verifying that the underlying motivation is both theoretically sound and empirically meaningful for graphs.
>
> Thus, while the high-level insight that SL models may capture spurious signals is shared with earlier IB and invariance studies, **our novelty lies in:**
> 1. showing how this issue specifically impacts graph OOD detection,
> 2. designing a **well-motivated and novel** information-theoretic based **tri-component decomposition** tailored to graph inputs, and
> 3. providing **new theoretical and empirical evidence** that this decomposition, when combined with IB, enhances **logit-based OOD detection** for graph data at the node-level.

---

> ### Author Response · Authors · 2025-11-18
> **Response to Reviewer uf9e (2/5)**
>
> > **W2 and Q2 Clarification of the mutual-information regularisers:**
>
> We thank the reviewer for these comments and clarify that the concerns arise from a **misinterpretation** of how TRIBE’s components interact. We clarify that the IB terms and the redundancy-reduction PMI terms do **not** compete; they operate on **different information pathways** and are complementary. Likewise, compression and conditional independence are **not contradictory**, but jointly enforce a stable and meaningful decomposition.
>
> ---
> #### **a) On the perceived conflict between maximising task relevance and minimising redundancy**
>
> This concern appears to treat the IB terms and the PMI regulariser as if they were pulling the **same representation** in opposing directions. In TRIBE, however, these objectives operate on **different aspects** of the three-branch system and are **not in contradiction**.
>
> The IB objective, $ \text{IB}_Z + \text{IB}_V + \text{IB}_Q = I(Z;Y) - \beta_Z I(X,A;Z) + I(V;Y) - \beta_V I(X;V) + I(Q;Y) - \beta_Q I(A;Q), $
> ensures that each encoder ($Z$, $V$, $Q$) **retains label-relevant information** from its own inputs while discarding label-irrelevant noise.
>
> The **pairwise MI** regulariser on the other hand, $ L_{\text{PMI}} = \alpha_1 I(Z;V) + \alpha_2 I(Z;Q) + \alpha_3 I(V;Q), $
> does **not** penalise label relevance. Instead, it penalises **redundant encoding of the same information** across the branches.
>
> Under the decomposition  $ I(X,A;Y) = I(Z;Y) + I(V;Y) + I(Q;Y), $
> the optimal solution under strong feature–structure correlation is that:
> - **$Z$** captures the shared, label-relevant interaction between $X$ and $A$,
> - **$V$** and **$Q$** capture the remaining feature-only and structure-only components.
>
> The PMI penalties therefore **do not suppress task-relevant information**. They regulate **where** information resides across the branches rather than deleting predictive content from $Z$. This is supported by the **Section 6.5 ablations**, where:
> - IB alone improves OOD detection over SL,
> - adding PMI yields **further improvements** without destabilising optimisation.
> ---
> #### **b) On the perceived conflict between compression and conditional independence**
>
> We clarify that the compression term and the conditional-independence regulariser are **complementary**, not conflicting, because **$Z$ is optimised to be label-relevant**.
>
> The concern seems to arise from interpreting minimising $I(X,A;Z)$ as discarding information required to enforce $I(X;A \mid Z)=0$. This is not the case.
>
> The IB objective for $Z$, $\max I(Z;Y) - \beta_Z I(X,A;Z),$ can be rewritten as: $ (1-\beta_Z)I(Z;Y) - \beta_Z I(X,A;Z \mid Y).$
> With moderate $\beta_Z$, this does **not** drive $I(X,A;Z)$ to zero. Instead, it encourages $Z$ to **retain the $X$-$A$ interactions that predict $Y$** while discarding label-irrelevant dependencies.
>
> The conditional-independence regulariser,
> $
> L_{\text{Cind}} = I(X;A \mid Z),
> $
> is then applied **after** $Z$ has captured this predictive joint signal. Any **remaining** dependence between $X$ and $A$ that is **not explained by $Z$** is treated as spurious and suppressed.
>
> Thus, the two objectives are consistent:
> - **IB** prevents overcompression and preserves **label-relevant** $X$-$A$ interactions,
> - **CInd** removes **label-irrelevant** residual correlations that do not contribute to $I(Z;Y)$.
>
> This behaviour aligns with:
> - the theoretical decomposition (Eq. 8–9),
> - the representation visualisations (Figure 4), and
> - the ablation study (Table 5), which shows that joint-input information remains well preserved even under strong $X$-$A$ correlations.
>
> Finally, the hyperparameter sensitivity analysis confirms that varying $\lambda_{\text{Cind}}$ and the $\alpha$ coefficients yields **stable improvements across a practical range**, supporting the intended behaviour of both regularisers.

---

> > ### Author Response · Authors · 2025-11-18
> > **Response to Reviewer uf9e (3/5)**
> >
> > > **W3 Experimental clarifications:**
> >
> > #### **(a) Experiment completeness clarification**
> >
> > Thank you for this question. We highlight that our study focuses on a **specific and clearly scoped research question**:
> > *How can graph OOD detection be addressed from an information-theoretic perspective, and in particular, how does an IB-based objective improve OOD performance relative to the standard supervised-learning paradigm?*
> >
> > Accordingly, our main comparisons rely on **widely adopted SL-trained detectors** such as GNNSAFE and NODESAFE (Lines 361–362). This setup isolates the effect of our IB analysis and tri-component decomposition **without confounding factors** introduced by orthogonal modelling directions.
> >
> > Methods such as **GOLD** and **DeGEM** adopt fundamentally different modelling assumptions:
> >
> > - **GOLD** uses **pseudo-OOD generation via latent diffusion**, placing it in the category of exposure-based or generative OOD detectors.
> > - **DeGEM** is tailored for **heterophily**, relying on multi-hop energy modelling, which is not the challenge we address (feature-structure entanglement).
> >
> > Because these approaches introduce **additional objectives** and **specialised architectures**, directly comparing them with TRIBE is not suitable to our central research question. However, **in response to the reviewer's question**, we implemented exploratory reproductions of these methods and **integrated our TRIBE objective into their pipelines**. Despite incomplete hyperparameter specifications (DeGEM) and potential implementation differences (seeds, data partitions), the results show:
> >
> > - **TRIBE is competitive with advanced OOD detectors**, and
> > - **TRIBE remains complementary**, yielding additional gains when combined with more complex architectures.
> >
> > These findings **strengthen** our main claim: TRIBE’s information-theoretic formulation improves OOD detection **across diverse frameworks**, not only under SL-trained baselines. The results and further discussions were included in the updated manuscript (Section 6.4).
> > | Dataset | Method | AUROC | AUPR | FPR95 | ID ACC |
> > |-|-|-|-|-|-|
> > | **Cora -Label** | GNNSafe | 93.19 | 82.99 | 29.55 | 89.66 |
> > || TRIBE | 93.70 | 84.03 | 28.84 | 90.61 |
> > || DeGEM | 92.24 | 78.80 | 31.34 | 91.77 |
> > || DeGEM + TRIBE | 95.85 | 89.24 | 20.59 | 92.41 |
> > || GOLD | 95.36 | 85.33 | 21.20 | 89.56 |
> > || GOLD + TRIBE | 94.85 | 83.96 | 18.86 | 89.56 |
> > | **Pubmed -S** | GNNSafe | 92.45 | 65.47 | 47.81 | 77.00 |
> > || TRIBE | 97.25 | 79.76 | 12.97 | 75.93 |
> > || DeGEM | 95.37 | 51.55 | 20.07 | 73.10 |
> > || DeGEM + TRIBE | 97.20 | 55.60 | 8.44 | 78.40 |
> > || GOLD | 88.01 | 97.84 | 11.57 | 75.30 |
> > || GOLD + TRIBE | 91.13 | 98.72 | 5.52 | 74.60 |
> > |**Pubmed -F** | GNNSafe | 95.18 | 74.41 | 27.62 | 76.57 |
> > || TRIBE | 97.44 | 81.70 | 12.53 | 76.40 |
> > || DeGEM | 99.63 | 93.40 | 1.70 | 79.00 |
> > || DeGEM + TRIBE | 99.67 | 91.73 | 0.59 | 78.70 |
> > || GOLD | 87.28 | 98.49 | 6.98 | 73.20 |
> > || GOLD + TRIBE | 93.15 | 99.00 | 3.80 | 74.70 |
> > | **Amazon-F** | GNNSafe | 98.46 | 98.90 | 0.44 | 92.65 |
> > || TRIBE | 98.65 | 99.04 | 0.30 | 92.70 |
> > || DeGEM | 97.34 | 96.70 | 3.16 | 91.65 |
> > || DeGEM + TRIBE | 97.71 | 96.49 | 1.84 | 92.32 |
> > || GOLD | 99.52 | 99.63 | 0.24 | 92.48 |
> > || GOLD + TRIBE | 99.61 | 99.70 | 0.10 | 91.98 |
> > | **Citeseer -S** | GNNSafe | 79.87 | 61.44 | 74.45 | 65.20 |
> > || TRIBE | 91.89 | 80.11 | 38.41 | 70.03 |
> > || DeGEM | 94.93 | 84.63 | 17.49 | 70.90 |
> > || DeGEM + TRIBE | 96.92 | 85.63 | 7.24 | 69.70 |
> > || GOLD | 78.09 | 82.12 | 65.98 | 69.10 |
> > || GOLD + TRIBE | 78.61 | 82.90 | 44.97 | 68.70 |

---

> > > ### Author Response · Authors · 2025-11-18
> > > **Response to Reviewer uf9e (4/5)**
> > >
> > > #### **(b) On the connection between the energy gap and the entropy gap:**
> > >
> > > Thank you for pointing this out. We agree that the earlier text did not explicitly connect the empirical **energy gap** with the theoretical **entropy gap** established in Proposition 5.3. We clarify the connection below.
> > >
> > > Our theoretical analysis in Sec. 5 is framed in terms of the predictive entropy of the softmax distribution $p_\theta(y \mid z) = \mathrm{Softmax}(\ell(z))$ (Eq. 21 in the Appendix). Lemma 5.1 and Theorem 5.2 show that the IB objective
> > > $\max I(Z;Y) - \beta I(X,A;Z), \beta > 0$
> > > leads to **higher ID prediction confidence** than standard supervised learning ($\beta=0$), and thus **lower predictive entropy** for ID samples.
> > >
> > > Proposition 5.3 further establishes that the learned representation $Z^*$ satisfies $H(Y \mid Z^{\text{id}}) \ll H(Y \mid Z^{\text{ood}}),$ demonstrating a strictly larger **entropy gap** under IB training.
> > >
> > > Here we note that the **energy score** used for OOD detection, $e(x) = -\log\sum_c \exp(\ell_c(x)),$ is the **negative log-partition term** of the **same logits** used in the softmax. Notably, expressing the entropy as: $
> > > H(Y \mid Z) = -\sum_c p_c \log p_c
> > > = -\sum_c p_c \ell_c + \log\sum_c \exp(\ell_c),
> > > $
> > > we see that the energy precisely corresponds to the log-partition component.
> > >
> > > Thus:
> > >
> > > - IB sharpening of ID posteriors (larger logit margins)
> > >   $\rightarrow$ **lower entropy** and **lower energy** for ID samples.
> > >
> > > - IB removal of label-irrelevant structure in OOD samples
> > >   $\rightarrow$  **higher entropy** and **higher energy** for OOD samples.
> > >
> > > Therefore, the **entropy gap** (theoretical) and the **energy gap** (empirical) measure **the same underlying change in logit geometry** induced by IB.
> > >
> > > ---
> > >
> > > #### **Visualisation validation**
> > >
> > > Figure 3 visualises this connection:
> > >
> > > - The first subplot shows TRIBE producing **more concentrated ID confidence** than GNNSAFE, consistent with the theoretical entropy gap.
> > > - The energy histograms show **clearer ID-OOD separation** for TRIBE, which is the empirical manifestation of the theoretical entropy gap through the energy formulation.
> > >
> > > We have added explicit clarifications in Sec. 6.6 (Lines 496-504) stating that the **energy gap is the log-partition realisation of the entropy gap in Proposition 5.3**. These clarifications ensure that the theoretical and empirical results are seen as two consistent perspectives on the same IB-induced effect.

---

> > > > ### Author Response · Authors · 2025-11-18
> > > > **Response to Reviewer uf9e (5/5)**
> > > >
> > > > > **Q1 Clarification on the necessity of Tri-component information decomposition**
> > > >
> > > > We thank the reviewer for the question. While existing IB-based and invariant-learning graph methods (InfoIGL, HGIF, CSIB, IBPL [1–4]) aim to reduce spurious correlations, they all operate on a **single mixed representation** $Z = f(X, A),$
> > > > typically trained at the **graph level**, and focus on graph generalisation rather than node-level OOD detection. These methods compress or regularise this *joint* embedding, or select invariant subgraphs, but they do **not** distinguish how much of $I(X,A;Y)$ originates from **feature-only**, **structure-only**, or **joint** signals. As a result, they cannot capture *which type* of information becomes spurious under feature shift, structure shift, or joint shift.
> > > >
> > > > In contrast, TRIBE is explicitly motivated by the information decomposition $I(X,A;Y) = I(Z;Y) + I(X;Y \mid Z) + I(A;Y \mid Z,X),$
> > > > introduced in Section 4.1. We operationalise this formulation with **three encoders** $(Z, V, Q)$ and their associated regularisers:
> > > >
> > > > - **$Z$** is trained to capture the *joint*, stable signal between $X$ and $A$ that is predictive of $Y$,
> > > > - **$V$** absorbs the **feature-only** residuals,
> > > > - **$Q$** absorbs the **structure-only** residuals.
> > > >
> > > > This design directly addresses the fact that *different components of $(X, A)$ become spurious under different OOD shifts*, something a single-encoder IB or invariant-learning method cannot identify or correct.
> > > >
> > > > The ablation study in **Table 5** shows that:
> > > >
> > > > - Starting from an IB backbone (which already plays a similar role to prior IB/invariant works),
> > > > - Adding the **conditional-independence term** produces consistent improvements,
> > > > - Adding the **full tri-component objective** yields **further gains** in AUROC, AUPR, and FPR95 across all datasets.
> > > >
> > > > Figures **3** and **4** reveal that TRIBE leads to:
> > > >
> > > > - **sharper ID confidence**,
> > > > - **larger ID-OOD energy gaps**,
> > > > - **clearer latent-space separation**,
> > > >
> > > > behaviours that do *not* emerge under SL or **single-encoder IB** methods. These empirical findings confirm that the decomposition is not only conceptually meaningful but leads to **measurable improvements in node-level OOD detection**.
> > > >
> > > > Therefore, compared to existing IB-based or invariant approaches, TRIBE is necessary because it does **not** simply compress a mixed graph representation. It **implements the full theoretical decomposition of $I(X,A;Y)$ into joint, feature-specific, and structure-specific components**, allowing finer-grained control over where information reside, an ability that proves crucial for effectively detecting OOD nodes under various graph shifts.
> > > >
> > > > Additionally, we provide further experiments comparing TRIBE with an **decoupled graph OOD method** (DeGEM) and the **graph invariant-learning**-based GraphIFE, which demonstrates the necessity of TRIBE for node OOD detection. **We kindly invite the reviewer to visit our response to Reviewer bK5E for detailed response.**
> > > >
> > > > We welcome any further suggestions for additional motivating experiments that could strengthen our findings.
> > > >
> > > > [1] Mao et al., *Invariant Graph Learning Meets Information Bottleneck for Out-of-Distribution Generalization*, FCS 2025.
> > > > [2] Ren et al., *Heterophilic Graph Invariant Learning for Out-of-Distribution Fraud Detection*, MM 2024.
> > > > [3] An et al., *Causal Subgraphs and Information Bottlenecks: Redefining OOD Robustness in Graph Neural Networks*, ECCV 2024.
> > > > [4] Cao et al., *IBPL: Information Bottleneck-based Prompt Learning for Graph Out-of-Distribution Detection*, Neural Networks 2025.

---

### Official Review · Reviewer_HnFV · 2025-11-01

**Soundness:** 2
**Presentation:** 2
**Contribution:** 2
**Rating:** 2
**Confidence:** 5

**Summary:**

This paper proposes TRIBE, a framework for graph OOD detection. The core idea is to decompose graph information into three parts — invariant, variant, and redundant components — with the goal of enhancing OOD detection by separating task-relevant and task-irrelevant information. The method leverages mutual information estimation to model the interaction among these components and integrates them into a graph representation learning framework. Experiments are conducted on several graph benchmarks to demonstrate the proposed method’s effectiveness compared with existing OOD detection baselines.

**Strengths:**

1. The idea of decomposing graph information into three components provides a new perspective for understanding and modeling graph representations.

2. The proposed framework attempts to connect information-theoretic principles with graph learning, which is conceptually interesting.

3. The paper includes some experimental evaluation across multiple datasets to demonstrate the general applicability of the method.

**Weaknesses:**

1. The core formulation of the tri-component decomposition is not clearly explained. It is unclear how the three components are defined, separated, or optimized in practice.

2. The technical novelty appears limited. The proposed framework largely combines existing concepts such as information decomposition and graph representation learning without a clear new algorithmic contribution.

3. The experiments do not provide convincing empirical support for the claimed benefits. Improvements are small or inconsistent, and there is no analysis showing that the decomposition itself enhances OOD detection.

4. The comparisons are incomplete. Recent and strong graph OOD detection methods  are missing, making it difficult to assess the true effectiveness of the proposed approach.

**Questions:**

Please refer to the comments listed in the Weaknesses section, which highlight points that would benefit from clarification or further empirical support.

---

> ### Author Response · Authors · 2025-11-18
> **Response to Reviewer HnFV (1/3)**
>
> Thank you for the comprehensive review and informative suggestions. We clarify your concerns below, using **bold font to clearly reaffirm our novelty**.
>
> ---
> > **W1 Clarification on how the three components are defined, separated, and practically optimised:**
>
> We thank the reviewer for raising this point. We highlight that the tri-component formulation is **formally defined, separated, and optimised in a fully specified manner** across Section 4 and the Appendix.
>
> - **Formally defined:** The decomposition is introduced in Section 4.1 through the mutual-information expansion in Eq. (8),
>   $$
>   I(X,A;Y) = I(Z;Y) + I(X;Y\mid Z) + I(A;Y\mid Z),
>   $$
>   which partitions label-relevant information into a joint term and two input-specific residuals. This formulation assigns the joint signal to **$Z$**, and the feature-only and structure-only signals to **$V$** and **$Q$**, giving each component a precise information-theoretic role.
>
> - **Separated:**  The three components are **separated architecturally** via encoders with separate inputs (Section 4.2):
>   - **$Z = f(X,A)$** captures the joint signal,
>   - **$V = g_X(X)$** receives features only,
>   - **$Q = g_A(A)$** receives structure only.
>  This architectural split ensures that each component extracts the input it is intended to represent, preventing mixing at the input level.
>
> - **Optimised:**  The components are **optimised by distinct objectives** specified in Eqs. (9)-(13).
>   - **$Z$** is optimised with its IB term, the conditional-independence constraint $I(X;A\mid Z)$, and PMI penalties, guiding it to encode stable joint interactions.
>   - **$V$** and **$Q$** are optimised with their respective IB losses and PMI penalties, driving them to encode the remaining input-specific information while staying disentangled from $Z$ and from each other.
>   Together, these losses realise the tri-component decomposition in practice.
>
> - **Practical optimisation:**
>   To facilitate practical implementation, Section 4.3 and the Appendix provide a **complete implementation-level description** (summarised in the **Reproducibility Statement**).
>   - Appendix E derives a tractable variational objective.
>   - Appendix F specifies the neural parameterisation of each encoder.
>   - **Algorithm 1** details the full optimisation loop (originally in the Appendix, now highlighted in Section 4.3).
>   Section 4.3 clarifies that **only $Z$ is used at inference**, while $V$ and $Q$ function solely as training-time regularisers.
>
> For clarity, we provide an explicit inference algorithm (Algorithm 2, Lines 299-308). Code will be released upon acceptance.
>
> ---
>
> > **W2 Novelty Clarification:**
>
> Thank you for the comment. We respectfully clarify that TRIBE is **not** a mix-and-match of existing components. It introduces a **new training paradigm and a new optimisation objective grounded in information-theoretic principles**, which no prior IB or invariant-learning method provides for graph OOD detection.
>
> - **Motivational novelty:**
>   Our starting point is simple: real world graphs suffer input-specific shifts in features $(X)$, structure $(A)$, and their joint interaction $(X,A)$. Standard GNNs entangle these modalities, making them vulnerable to spurious correlations. Thus, TRIBE offers a **graph-native solution**: it explicitly separates joint-input information from input-specific signals. We have **updated Figure 1** to better illustrate this motivation.**To our knowledge, no prior graph OOD method provides such an information-theoretic decomposition**.
>
> - **Technical novelty:**
>   TRIBE introduces a **dedicated three-way information-decomposition architecture** for graph OOD detection, strengthened by **two new regularisers** (CInd and PMI). Together with the TRIBE objective (Eq. 13, Algorithm 1), this constitutes a **new training paradigm**, not an IB variant or adaptation of existing IB-OOD approaches.
>   Prior IB/OOD methods (GIB, DRL, VIB [1-3])
>   - do not model graphs,
>   - do not handle entangled multi-modality inputs, and
>   - do not provide $(Z,V,Q)$ decomposition or CInd/PMI constraints.
>   These contributions cannot be replicated by simply combining existing components.
>
> - **Theoretical novelty:**
>   We provide, to the best of our knowledge, the **first theoretical connection showing that the IB objective improves logit-based OOD detection**.
>   - **Lemma 5.1** and **Theorem 5.2** show that IB increases ID prediction confidence beyond maximising $I(Z;Y)$ alone.
>   - **Proposition 5.3** proves that IB induces a strictly larger **entropy gap** between ID and OOD samples, leading to improved detection.
>   Prior IB-OOD work has not established this link, focusing instead on image data, class-conditional compression, or uncertainty heuristics.
>
> [1] Wu et.al., Graph Information Bottleneck. NIPS20.
>
> [2] Zhao et.al., Dual Representation Learning for Out-of-Distribution Detection. TMLR23.
>
> [3] Alemi et.al., Deep Variational Information Bottleneck. ICLR17.

---

> > ### Author Response · Authors · 2025-11-18
> > **Response to Reviewer HnFV (2/3)**
> >
> > > **W3 On the perceived marginal improvement and limited experimental analysis:**
> >
> > We thank the reviewer for the comment, we would like to invite the reviewer to suggest a clearer instruction on which evaluation scenario to further boost the performance of TRIBE. We highlight that the paper provides **consistent, dataset-wide improvements** and includes **multiple dedicated analyses** that isolate and validate the contributions of both the IB objective and the tri-component decomposition.
> >
> > - **Stable performance gain:**
> >   The improvements are **not marginal**. Across seven diverse datasets (Table 1), TRIBE reduces FPR95 by up to **34%** on Cora, Citeseer, Pubmed, and Amazon, substantial gains for graph OOD detection. On challenging real-world benchmarks (Twitch and Arxiv), TRIBE achieves **consistent improvements over the SL-based baselines**. These hold under both non-exposure and exposure settings (Table 3), demonstrating that the improvement is **consistent across datasets and traning setups**.
> >
> > - **Efficacy analysis:**
> >   Several analyses directly show **why the decomposition improves OOD detection**:
> >   • **Figure 4** shows that the joint encoder $Z$ learned under our decomposition yields **clearer ID-OOD separation** than a standard GCN baseline. The feature-only $V$ and structure-only $Q$ components display input-specific patterns consistent with their roles.
> >   • **Section 6.2** shows that IB alone creates **larger energy gaps** between ID and OOD samples compared to SL, even before adding decomposition regularisers.
> >   • The **ablation study (Table 5)** isolates the contribution of the decomposition. Removing the CInd and PMI constraints degrades FPR95, while adding them progressively improves OOD performance across all datasets. This demonstrates that the decomposition provides **predictive and detection benefits beyond the IB backbone**.
> >
> > - **Theory-to-practice alignment:**
> >   Our theoretical analysis (Lemma 5.1, Theorem 5.2, Proposition 5.3) predicts that IB enlarges the **entropy gap** between ID and OOD samples, improving logit and energy separation.
> >   **Section 6.5 and Figure 3** validate this empirically, showing that TRIBE produces **more confident ID predictions** and **better-separated OOD energies** than SL baselines.
> >   These results provide a **consistent and coherent theory-practice link**, confirming that the model behaves as intended and explaining the empirical gains.

---

> > > ### Author Response · Authors · 2025-11-18
> > > **Response to Reviewer HnFV (3/3)**
> > >
> > > > **W4 Experimental completeness clarification:**
> > >
> > > We appreciate the reviewer’s comment on the breadth of comparisons.
> > >
> > > **Experimental design rationale:**
> > >   We highlight that our experimental setup is aligned with the central research question of this work:
> > >   *how does an information-theoretic, IB-based, decomposition-aware training objective improve node-level OOD detection compared to standard supervised learning?*
> > >
> > >   For this reason, our primary comparisons focus on widely adopted SL-trained graph OOD detectors (GNNSAFE, NODESAFE). This allows a **clean, controlled evaluation** of the contributions introduced by our IB analysis and decomposition framework, supported through visualisations and ablation studies.
> > >
> > > **On the reviewer’s comment regarding additional baselines:** We would like to clarify that the current version already includes comparisons with several relevant and high-impact methods (e.g., Energy, GKDE, GNNSafe, NodeSafe), spanning NeurIPS 20,21, ICLR 23, and ICML 24. However, in response to this review, we have **further experimented with two additional graph OOD detection methods (DeGEM, GOLD)** from ICLR25, and report their results below. We also conducted **exploratory experiments** by implementing a TRIBE-variant with these advanced architectures. These results were reproduced to the best of our ability within this short turnaround. Deviations from the original reports may stem from differences in data partitions, random seeds, or unavailable hyperparameter settings, as the precise configurations were not publicly specified (DeGEM). The results demonstrate that **TRIBE is on par with or complementary to advanced detectors**, and that IB-based objectives can **enhance modern graph OOD pipelines**. Further discussion is included in the updated manuscript (Section 6.4).
> > >
> > > While our primary goal is to analyse the gap between SL and IB-based training, we are committed to ensuring broad and fair comparison. If the reviewer has specific additional methods in mind that they believe are essential, we would kindly appreciate the guidance and will be happy to include further evaluations.
> > > | Dataset | Method | AUROC | AUPR | FPR95 | ID ACC |
> > > |-|-|-|-|-|-|
> > > | **Cora -Label** | GNNSafe | 93.19 | 82.99 | 29.55 | 89.66 |
> > > || TRIBE | 93.70 | 84.03 | 28.84 | 90.61 |
> > > || DeGEM | 92.24 | 78.80 | 31.34 | 91.77 |
> > > || DeGEM + TRIBE | 95.85 | 89.24 | 20.59 | 92.41 |
> > > || GOLD | 95.36 | 85.33 | 21.20 | 89.56 |
> > > || GOLD + TRIBE | 94.85 | 83.96 | 18.86 | 89.56 |
> > > | **Pubmed -S** | GNNSafe | 92.45 | 65.47 | 47.81 | 77.00 |
> > > || TRIBE | 97.25 | 79.76 | 12.97 | 75.93 |
> > > || DeGEM | 95.37 | 51.55 | 20.07 | 73.10 |
> > > || DeGEM + TRIBE | 97.20 | 55.60 | 8.44 | 78.40 |
> > > || GOLD | 88.01 | 97.84 | 11.57 | 75.30 |
> > > || GOLD + TRIBE | 91.13 | 98.72 | 5.52 | 74.60 |
> > > |**Pubmed -F** | GNNSafe | 95.18 | 74.41 | 27.62 | 76.57 |
> > > || TRIBE | 97.44 | 81.70 | 12.53 | 76.40 |
> > > || DeGEM | 99.63 | 93.40 | 1.70 | 79.00 |
> > > || DeGEM + TRIBE | 99.67 | 91.73 | 0.59 | 78.70 |
> > > || GOLD | 87.28 | 98.49 | 6.98 | 73.20 |
> > > || GOLD + TRIBE | 93.15 | 99.00 | 3.80 | 74.70 |
> > > | **Amazon-F** | GNNSafe | 98.46 | 98.90 | 0.44 | 92.65 |
> > > || TRIBE | 98.65 | 99.04 | 0.30 | 92.70 |
> > > || DeGEM | 97.34 | 96.70 | 3.16 | 91.65 |
> > > || DeGEM + TRIBE | 97.71 | 96.49 | 1.84 | 92.32 |
> > > || GOLD | 99.52 | 99.63 | 0.24 | 92.48 |
> > > || GOLD + TRIBE | 99.61 | 99.70 | 0.10 | 91.98 |
> > > | **Citeseer -S** | GNNSafe | 79.87 | 61.44 | 74.45 | 65.20 |
> > > || TRIBE | 91.89 | 80.11 | 38.41 | 70.03 |
> > > || DeGEM | 94.93 | 84.63 | 17.49 | 70.90 |
> > > || DeGEM + TRIBE | 96.92 | 85.63 | 7.24 | 69.70 |
> > > || GOLD | 78.09 | 82.12 | 65.98 | 69.10 |
> > > || GOLD + TRIBE | 78.61 | 82.90 | 44.97 | 68.70 |

---

### Official Review · Reviewer_bK5E · 2025-11-02

**Soundness:** 3
**Presentation:** 3
**Contribution:** 2
**Rating:** 4
**Confidence:** 4

**Summary:**

This paper presents TRIBE, a novel framework for graph out-of-distribution (OOD) detection based on tri-component information decomposition. The authors identify that conventional supervised learning objectives in graph neural networks (GNNs) often entangle spurious correlations from node features and graph structure, resulting in poor generalization to OOD scenarios. To tackle this, TRIBE decomposes the mutual information between input (features & structure) and labels into three components: feature-specific, structure-specific, and joint-input signals. The framework incorporates an Information Bottleneck (IB) objective alongside pairwise mutual information regularization and conditional independence constraints, aiming to suppress spurious components while preserving only label-relevant joint information. Empirical evaluations across seven benchmark datasets show that TRIBE significantly improves OOD detection performance (up to 34% reduction in FPR95) while maintaining competitive in-distribution (ID) classification accuracy.

**Strengths:**

1. The paper introduces a principled and technically sound decomposition of predictive information into three components (feature, structure, joint). This is a non-trivial extension of IB to the graph domain, which is a valuable contribution.
2. Clear theoretical results are provided to justify why the IB objective is more suitable for OOD detection than standard supervised learning.
3. Experiments on diverse benchmarks demonstrate consistent performance gains over both non-OOD and OOD-exposed baselines.
4. The ablation study is thorough, showing the importance of each component to the overall performance.

**Weaknesses:**

1. While the method is conceptually interesting, the paper lacks a clear visual or algorithmic description of how all the modules (Z, V, Q networks) interact during training and inference. Figure 2 helps, but further schematic detail is needed to fully grasp the flow of gradients, loss contributions, and updates.
2. The paper relies heavily on mutual information estimators (e.g., CLUB), but does not provide sufficient detail or sensitivity analysis on the impact of approximation errors in these estimators.
3. The objective combines several components (IB terms, CI loss, PMI regularizers) with scalar weights (e.g., λ, α₁–α₃), but their selection strategy is not transparent.
4. While the paper compares against strong OOD baselines, it does not evaluate against decomposition-based or disentangled representation methods, which could be relevant comparators in the context of learning disentangled graph representations.

**Questions:**

1. Why is the feature network implemented as an MLP and the structure network as a GCN?

---

> ### Author Response · Authors · 2025-11-18
> **Response to Reviewer bK5E (1/2)**
>
> We thank the reviewer for recognising our decomposition framework, theoretical contributions, and strong empirical results in addressing the challenge of OOD detection for graph-structured data. We have carefully addressed your concerns as follows:
>
> ---
> > **W1 Module interactions, optimisation process, and algorithmic description:**
>
> Thank you for the thoughtful comment. We agree that a rigorous description of how the three modules ($Z, V, Q$) interact is essential, and **we emphasise that the paper already provides this information**, both conceptually in the main text and algorithmically in the appendix (We positioned this content within the main text of the revised manuscript). We highlight the key points below.
>
> - **Optimisation and module interaction:**
>   Section 4.2 defines the **complete optimisation structure**, where each encoder receives gradients **only from the losses associated with its intended information component**. Specifically, the $Z$ network is optimised by its IB term as well as the proposed regularisers, ensuring **it captures only the joint-input signal**. The $V$ and $Q$ networks, in contrast, receive gradients solely from their respective IB objectives and the pairwise MI penalties, allowing them **to isolate feature-/structure-specific residual signals**. These interactions are fully specified in Eq. (9) to (13), which collectively detail how each loss contributes to each module’s updates.
>
> - **Algorithmic description:**
>   Beyond the mathematical description, we provide a **complete algorithmic and implementation-level breakdown** in the appendix, as noted in Section 4.3 and the Reproducibility Statement in Lines. Appendix E describes how each information-theoretic term is transformed into a tractable variational objective. Appendix F provides the **full neural-network parameterisation** of $Z$, $V$, and $Q$. **Algorithm 1 outlines the full optimisation loop**, including the forward pass through all modules, computation of VIB, CLUB, and reconstruction terms, and joint gradient updates for all parameters.
>
> - **Inference behaviour:**
>   Inference-time behaviour is intentionally simple and is clearly specified in Section 4.3, where we state that **only the joint encoder $Z$ is used for classification and OOD scoring**. $V$ and $Q$ act solely as **training-time regularisers** and do not participate in inference. This design choice keeps the model lightweight at test time. For clarity, we provide **an explicit algorithmic description of inference** in Algorithm 2 in the **updated manuscript**.
>
> **We have updated the manuscript to include the Algorithms in the main text (Lines 280 -308).**
>
> We will also release the code upon acceptance.
>
> ---
>
> > **W2 Mutual information estimation clarification:**
>
> We thank the reviewer for pointing this out. Regarding sensitivity to approximation error, we emphasise that all MI-based quantities enter the objective as **regularisers**, not as hard constraints that determine the decision boundary. The dominant signal for the classifier still comes from the **label-likelihood term** $I(Z;Y)$ (implemented via cross-entropy) together with the **IB-style compression term**. The MI-based regularisers shape the representation geometry but **do not override label supervision**. Their effect on the classifier is controlled by the coefficients **$\beta$**'s (IB terms), **$\alpha$**’s (PMI terms), and **$\lambda$** for the conditional-independence loss.
>
> - **Empirical evidence:**
>   In Appendix J, we report a hyperparameter sensitivity study over several orders of magnitude for these coefficients (Tables 6-8). Across these sweeps, OOD metrics change smoothly, and **TRIBE maintains a consistent advantage over the baseline (GNNSAFE)**, indicating robustness to MI approximators. Since approximation errors in MI estimators manifests as effective rescaling of the corresponding terms, this provides indirect but practical evidence that **TRIBE’s performance does not rely on highly accurate MI values**.
>
> - **Additional experiment:**
>   Additionally, we replace **CLUB** with an alternative MI estimator (**InfoNCE**). Due to time constraints, we report results on a selection of datasets. The resulting changes in metrics are small, and the **relative advantage of TRIBE remains**. This indicates that TRIBE’s performance is driven by the **overall IB-decomposition design** rather than any specific MI estimator, and that optimisation remains stable under reasonable perturbations of these approximations.
>
> |Pubmed-S|AUROC|AUPR|FPR95|ACC|Cora-S|AUROC|AUPR|FPR95|ACC|Cora-F|AUROC|AUPR|FPR95|ACC|
> |-|-|-|-|-|-|-|-|-|-|-|-|-|-|-|
> |**w/CLUB**|97.25|79.76|12.97|75.93||95.15|89.33|23.31|78.97||97.88|94.67|8.13|78.10|
> |**w/InfoNCE**|97.33|77.61|11.69|74.30||94.95|88.86|24.78|78.20||97.94|94.38|6.94|78.10|
>
> |Citeseer-L|||||Coauthor-S|||||
> |-|-|-|-|-|-|-|-|-|-|
> |**w/CLUB**|90.97|64.92|30.84|88.55||99.95|99.94|0.13|92.72|
> |**w/InfoNCE**|90.69|63.28|31.34|87.54||99.94|99.92|0.13|92.66|

---

> ### Author Response · Authors · 2025-11-18
> **Response to Reviewer bK5E (2/2)**
>
> > **W3 Hyperparameters selection:**
>
> We appreciate the reviewer’s comment. As detailed in Appendix J, since these are objective weight scale factor, we follow the general practice of multi-objective combinations in deep learning to treat them as hyperparameters. And our hyperparameter sensitivity analysis in Tables 6-8 indicates the stability and robustness of our proposed TRIBE. Specifically, we conduct a grid search over a predefined, computationally feasible range for the scalar weights ($\lambda, \beta, \alpha$’s), and report a full hyperparameter sensitivity analysis in Tables 6-8. Because OOD data are not available during training or validation, all weights are **selected based on in-distribution performance** (e.g., validation loss). Although the grid search range is necessarily constrained for computational practicality, the sensitivity results confirm that **TRIBE is robust across these settings** and that the chosen values are effective. We also acknowledge in the Potential Limitations section that constant $\alpha$ and $\beta$ values were applied uniformly in the current experiments, and that **future work will explore finer-grained control of individual terms**.
>
> ---
> > **W4 Additional disentanglement-based methods:**
>
> Thank you for the suggestion. We include below results comparing TRIBE with two representative recent approaches: the **decoupled graph OOD detector** DeGEM and the **graph invariant-learning**-based GraphIFE [1].
>
> Notably, DeGEM is a specialised graph OOD detector with a tailored training design, so it naturally achieves competitive performance relative to TRIBE. To ensure a fair comparison of objectives, we additionally report an **TRIBE-enhanced variant** of DeGEM. This variant consistently improves its logit-based detection metrics, which aligns with our theoretical finding that the IB principle sharpens ID predictions and enlarges the ID-OOD separation.
>
> In contrast, GraphIFE, an invariant learning method, is **not designed for node-level OOD detection**. Using energy score for evaluation, both the **with and without energy propagation**-enhanced form show limited detection capability across all datasets, despite achieving reasonable ID accuracy. These results indicate that **typical invariant learning alone is insufficient for node-level OOD detection**, and they reinforce the importance of studying **OOD-specific information decomposition**, which TRIBE directly targets.
>
> |Dataset|Method|AUROC|AUPR|FPR95|IDAcc|
> |:---|:---|:---|:---|:---|:---|
> |**Cora–S**|TRIBE|95.15|89.33|23.31|78.97|
> ||DeGEM|92.02|70.24|18.91|79.80|
> ||DeGEM+TRIBE|96.43|90.18|15.03|77.80|
> ||GraphIFE|72.70|51.68|84.68|78.90|
> ||GraphIFE+Prop|83.03|75.67|77.70|-|
> |**Cora–F**|TRIBE|97.88|94.67|8.13|78.10|
> ||DeGEM|98.66|96.14|5.76|82.80|
> ||DeGEM+TRIBE|99.05|96.86|3.62|82.20|
> ||GraphIFE|80.45|70.74|78.84|78.75|
> ||GraphIFE+Prop|84.83|75.39|73.14|-|
> |**Pubmed–S**|TRIBE|97.25|79.76|12.97|75.93|
> ||DeGEM|95.37|51.55|20.07|73.10|
> ||DeGEM+TRIBE|97.20|55.60|8.44|78.40|
> ||GraphIFE|67.88|17.93|99.84|76.00|
> ||GraphIFE+Prop|75.35|47.39|97.55|-|
> |**Pubmed–F**|TRIBE|97.44|81.70|12.53|76.40|
> ||DeGEM|99.63|93.40|1.70|79.00|
> ||DeGEM+TRIBE|99.67|91.73|0.59|78.70|
> ||GraphIFE|71.02|31.68|99.97|75.95|
> ||GraphIFE+Prop|74.32|42.27|89.98|-|
>
> ---
> > **Q1 Auxiliary network implementation choices:**
>
> Thank you for raising this question. Our design choices directly follow the intended roles of the two auxiliary networks in TRIBE’s decomposition framework: **$V$ should capture feature-only information**, and **$Q$ should capture structure-only information**. The architectures are chosen to enforce this disentanglement rather than model preference.
>
> 1. **Feature network $V$ = MLP (captures feature-only information)**
>    The purpose of $V$ is to isolate **feature-specific** components that do not rely on graph propagation. We feed only the raw node attributes $X$ into an MLP to ensure that $V$ depends solely on features and does not inadvertently encode structural context. A GNN with identity adjacency could simulate "no propagation," but this would introduce unnecessary complexity. A simple MLP is therefore the appropriate choice.
>
> 2. **Structure network $Q$ = GCN (captures structure-only information)**
>    The purpose of $Q$ is to encode predictive information arising purely from graph topology. To enforce this, we use a GCN with **constant node features** (all-ones input), as noted in Sec. 6 (Line 371). This ensures that $Q$ encodes only structural variation from the adjacency matrix $A$. As discussed in Appendix A (Lines 829-833), TRIBE adopts a simple structural encoder without loss of generality. **More expressive alternatives** (learnable positional encodings, spectral embeddings, structure-only encoders) can be explored in future work.
>
> [1] Zeng et al., *GraphIFE: Rethinking Graph Imbalance Node Classification via Invariant Learning*, 2025.

---

> > ### Comment · Reviewer_bK5E · 2025-11-25
> >
> > I appreciate the authors’ detailed rebuttal, which has satisfactorily resolved most of my concerns. I encourage the authors to reflect these clarifications in the camera-ready version. I am updating my score accordingly.

---

> > > ### Author Response · Authors · 2025-11-25
> > >
> > > Dear Reviewer bK5E,
> > >
> > > Thank you very much for your thoughtful follow-up and for taking the time to re-evaluate our work. We truly appreciate your constructive feedback and are glad that our clarifications addressed your concerns. We once again thank you for recognising our valuable contribution to the challenging graph OOD problem.
> > >
> > > We will ensure to carefully incorporate these updates into a future revision of the manuscript.

---

### Meta-Review · Area_Chair_eHb3 · 2025-12-16

**Summary:**

The four reviewers' comments mainly fall into two categories.

Two were positive, believing that the paper proposes a rigorous and technically sound method for decomposing predictive information into three components (features, structure, and union). This is a nontrivial extension of information blocks (IBs) to the graph domain and a valuable contribution.

The other two were negative, stating that the core formula of the three-component decomposition is not clearly explained.  The improvements are small or unstable, and there is no analysis showing that the decomposition itself enhances OOD detection. The comparisons are not comprehensive enough. The lack of some state-of-the-art and effective graph OOD detection methods makes it difficult to assess the true effectiveness of the proposed method.

**Reviewer Concerns:**

Regarding the theoretical and technical innovation, motivation, and comparison with more advanced algorithms, the authors provided detailed responses and experimental reports. Although more experiments demonstrate the effectiveness of the method, the explanation of its theoretical motivation and technical solution remains unclear, failing to completely dispel all reviewers' doubts.

**Reviewer Scores:**

One reviewer gave a high score of 8.
One reviewer gave 4 and is willing to improve after the authors' response.
Two others maintained their scores of 4 and 2.
I believe the authors' current response is unlikely to completely dispel the doubts of the two reviewers who gave negative feedback.

---

### Decision · Program_Chairs · 2026-01-26

Reject